# Collaborative Linear Bandits with Adversarial Agents: Near-Optimal Regret Bounds

**Aritra Mitra**[*]     **Arman Adibi**[*]     **George J. Pappas**     **Hamed Hassani**
Department of Electrical and Systems Engineering
{amitra20, aadibi, hassani, pappasg}@seas.upenn.edu

## Abstract

We consider a linear stochastic bandit problem involving $M$ agents that can collaborate via a central server to minimize regret. A fraction $\alpha$ of these agents are adversarial and can act arbitrarily, leading to the following tension: while collaboration can potentially reduce regret, it can also disrupt the process of learning due to adversaries. In this work, we provide a fundamental understanding of this tension by designing new algorithms that balance the exploration-exploitation trade-off via carefully constructed robust confidence intervals. We also complement our algorithms with tight analyses. First, we develop a robust collaborative phased elimination algorithm that achieves $\tilde{O}\left(\alpha + 1/\sqrt{M}\right)\sqrt{dT}$ regret for each good agent; here, $d$ is the model-dimension and $T$ is the horizon. For small $\alpha$, our result thus reveals a clear benefit of collaboration despite adversaries. Using an information-theoretic argument, we then prove a matching lower bound, thereby providing the first set of tight, near-optimal regret bounds for collaborative linear bandits with adversaries. Furthermore, by leveraging recent advances in high-dimensional robust statistics, we significantly extend our algorithmic ideas and results to (i) the generalized linear bandit model that allows for non-linear observation maps; and (ii) the contextual bandit setting that allows for time-varying feature vectors.

## 1 Introduction

One of the primary challenges in modern large-scale computing systems is that of *security*. Given that the individual agents in such large systems are often vulnerable to attacks, it is important to understand how the overall system behaves in the face of adversarial corruptions. This observation has spurred a line of research dedicated to the design and analysis of distributed algorithms that are provably robust to a small fraction of adversarial agents; notably, motivated by emerging learning paradigms such as federated learning [1–3], this body of work has focused primarily on empirical risk minimization/stochastic optimization [4–14]. However, when it comes to multi-agent sequential decision-making problems under uncertainty (e.g., bandits and reinforcement learning), our understanding of analogous questions is quite limited. Our goal in this paper is to bridge the above gap by studying a collaborative linear bandit [15, 16] problem in the presence of adversaries.

In our model, $M$ agents interact with the same linear bandit characterized by a $d$-dimensional unknown parameter $\theta_*$, and a finite set of $K$ arms. These agents can collaborate via a central server to improve performance, as measured by cumulative regret. As examples, consider (i) a team of robots exploring actions (arms) in a common environment and interacting with a central controller; and (ii) a group of people exploring restaurants (arms) and writing reviews for a web-recommendation server. In the absence of adversaries, there is a clear reason to collaborate in either case: by exchanging information, each agent can reduce its uncertainty about the arms faster than it could when it acts

---

[*]The first two authors contributed equally.

alone, and thereby incur lesser regret. The situation becomes murkier and more delicate when certain agents *misbehave*: What if certain robots get attacked or certain people deliberately write spam reviews? More generally, the main question we ask in this paper is the following.

*In a multi-agent linear stochastic bandit problem, can we still hope for benefits of collaboration when a fraction $\alpha$ of the agents are adversarial? If so, what are the fundamental limits of such benefits?*

As far as we are aware, the answers to these questions have thus far remained elusive, motivating our current study. The main technical hurdle we must overcome is to delicately balance the exploration-exploitation trade-off in the presence of both statistical uncertainties due to the environment, and *worst-case* adversarial behavior. Importantly, the above trade-off - intrinsic to sequential decision-making - is absent in static optimization problems. Thus, the ideas used to guarantee robustness for distributed optimization do not apply to our problem, making our task quite non-trivial.

**Our Contributions.** In this paper, we contribute to a principled study of several canonical *structured* linear bandit settings with adversarial agents. Our specific contributions are as follows.

• **Robust Collaborative Linear Bandit Algorithm.** We propose RCLB - a phased elimination algorithm that relies on distributed exploration, and balances the exploration-exploitation dilemma in the presence of adversaries via carefully constructed *robust confidence intervals*. We prove that RCLB guarantees $\tilde{O}\left(\left(\alpha + 1/\sqrt{M}\right)\sqrt{dT}\right)$ regret for each good agent; see Theorem 1. This result is both novel, and significant in that it reveals a clear benefit of collaboration (despite adversaries) for small values of $\alpha$. In particular, when $\alpha = 0$, the regret bound of RCLB is minimax-optimal in all relevant parameters: the model-dimension $d$, the horizon $T$, and the number of agents $M$.

• **Fundamental Limits.** At this stage, one may ask: *Is the additive $\alpha\sqrt{T}$ term in Theorem 1 simply an artifact of our analysis?* In Theorem 2, we establish a fundamental lower bound, revealing that such a term is in fact unavoidable; it is the price one must pay due to adversarial corruptions. A key implication of this result is that *our work is the first to provide tight, near-optimal regret bounds for collaborative linear bandits with adversaries.* The proof of Theorem 2 relies on a novel connection between the information-theoretic arguments in [17], and ideas from the robust mean estimation literature [18, 19]. As such, our proof technique may be relevant for related settings.

In our next set of contributions, we significantly extend our results to more general bandit models.

• **Generalized Linear Bandit Setting.** In Theorem 3, we prove that one can achieve bounds akin to that in Theorem 1 for the generalized linear bandit model (GLM) [20, 21] that accounts for non-linear observation maps. To achieve this result, we propose a variant of RCLB that leverages very recently developed tools from high-dimensional robust Gaussian mean estimation [22]. Deriving robust confidence intervals for this setting requires some work: we exploit regularity properties of the non-linear observation model along with error bounds from [22] for this purpose. As far as we are aware, Theorem 3 is the first result to establish adversarial-robustness for GLMs, allowing our framework to be applicable to a broad class of problems (e.g., logistic and probit regression models).

• **Contextual Bandit Setting.** Finally, we turn our attention to the contextual bandit setting where the feature vectors of the arms can change over time. This setting is practically quite relevant as web-recommendation systems are often modeled as contextual bandits [23]. The main challenge here arises from the need to simultaneously contend with time-varying optimal arms and adversaries. To handle this scenario, we develop a robust variant of the SUPLINREL algorithm [24] that guarantees a near-optimal regret bound identical to that of Theorem 1; see Theorem 4.

Overall, via the proposal of new robust algorithms complemented with tight analyses, our work takes an important step towards multi-agent sequential decision-making in the presence of adversaries.

**Related Work.** There is a growing strand of literature that studies the effect of reward corruption in stochastic bandits (for a single-agent setting) where the adversary has a fixed corruption budget [25–32]. The techniques in these papers do not apply to our work as our setting is very different: the adversarial agents in our model can act *arbitrarily*, and have no budget constraints. Several papers study multi-agent bandit problems in the absence of adversaries [33–52]. A few very recent ones [53–56] also look at the effect of attacks, but for the simpler unstructured multi-armed bandit problem. Accounting for adversarial agents in the structured linear bandit setting we consider here requires significantly different techniques that we develop in this paper.

## 2   Problem Formulation

We consider a setting comprising of a central server and $M$ agents; the agents can communicate only via the server. Each agent $i \in [M]$ interacts with the same linear bandit model characterized by an unknown parameter $\theta_*$ that belongs to a known compact set $\Theta \subset \mathbb{R}^d$. We assume $\|\theta_*\| \leq 1$.[2] The set of actions $\mathcal{A}$ for each agent is given by $K$ distinct vectors in $\mathbb{R}^d$, i.e., $\mathcal{A} = \{a_1, \ldots, a_K\}$, where $K$ is a finite, positive integer. Based on all the information acquired by an agent $i$ up to time $t-1$, it takes an action $a_{i,t} \in \mathcal{A}$ at time $t$, and receives a reward $y_{i,t}$ given by the following observation model:

$$y_{i,t} = \langle \theta_*, a_{i,t} \rangle + \eta_{i,t}. \tag{1}$$

Here, $\{\eta_{i,t}\}$ is a sequence of independent Gaussian random variables with zero mean and unit variance. Thus far, we have essentially described a distributed/multi-agent linear stochastic bandit model. Departing from this standard model, we focus on a setting where a fraction $\alpha \in [0, 1/2)$ of the agents are adversarial; the adversarial set is denoted by $\mathcal{B}$, where $|\mathcal{B}| = \alpha M$. In particular, we consider a *worst-case* attack model, where each adversarial agent $i \in \mathcal{B}$ is assumed to have complete knowledge of the system, and is allowed to act arbitrarily. Under this attack model, an agent $i \in \mathcal{B}$ can transmit arbitrarily corrupted messages to the central server.

Our performance measure of interest is the following group regret metric $R_T$ defined w.r.t. the non-adversarial agents:

$$R_T = \mathbb{E}\left[ \sum_{i \in [M] \setminus \mathcal{B}} \sum_{t=1}^{T} \langle \theta_*, a_* - a_{i,t} \rangle \right], \tag{2}$$

where $a_* = \operatorname{argmax}_{a \in \mathcal{A}} \langle \theta_*, a \rangle$ is the optimal arm, and $T$ is the time horizon.[3] We will work under a regime where the horizon $T$ is large, satisfying $T \geq Md$. The goal of the good (non-adversarial) agents is to collaborate via the server and play a sequence of actions that minimize the group regret $R_T$. Let us now make a few key observations. In principle, each good agent can choose to act independently throughout (i.e., not talk to the server at all), and achieve $\tilde{O}(\sqrt{dT})$ regret by playing a standard bandit algorithm. Clearly, the group regret $R_T$ would scale as $\tilde{O}\left((1-\alpha)M\sqrt{dT}\right)$ in such a case. In the absence of adversaries however, one can achieve a significantly better group regret bound of $\tilde{O}\left(\sqrt{MdT}\right)$ via collaboration, i.e., the regret per good agent can be reduced by a factor of $\sqrt{M}$ relative to the case when it acts independently (see, for instance, [41] and [45]). Our specific interest in this paper is to investigate whether, and to what extent, one can retain the benefits of collaboration despite the worst-case attack model described above. Said differently, we ask: *Can we improve upon the trivial per agent regret bound of $\tilde{O}(\sqrt{dT})$ in the presence of adversaries?*

Throughout the rest of the paper, we will answer the above question in the affirmative by deriving novel robust algorithms for several canonical bandit models, and then establishing near-optimal regret bounds for each such model.

## 3   Robust Collaborative Phased Elimination Algorithm for Linear Bandits

In this section, we develop a robust phased elimination algorithm that achieves the near-optimal regret bound of $\tilde{O}\left(\left(\alpha + \sqrt{1/M}\right)\sqrt{dT}\right)$ per good agent. This is non-trivial as we must account for the worst-case attack model described in Section 2. To highlight the challenges that we need to overcome, consider the following scenario. During the initial stages of the learning process, when the arms in $\mathcal{A}$ have not been adequately sampled by the agents, even a good agent may have "poor estimates" of the true payoffs associated with each arm, i.e., the variance associated with such estimates may be large. This statistical uncertainty can be exploited by the adversarial agents to their benefit. In particular, we need to devise an approach that can distinguish between benign stochastic perturbations (due to the noise in our model) and deliberate adversarial behavior. In what follows, we describe our proposed algorithm - `Robust Collaborative Phased Elimination for Linear Bandits (RCLB)` - that precisely does so in a principled way.

---

[2] We will use $\|\cdot\|$ to represent the Euclidean norm, and $a'$ to denote the transpose of a vector $a$.

[3] For ease of exposition, we assume that there is an unique optimal arm.

---

**Algorithm 1** Robust Collaborative Phased Elimination for Linear Bandits (`RCLB`)

---

**Input:** Action set $\mathcal{A} = \{a_1, \ldots, a_K\}$, confidence parameter $\delta$, and corruption fraction $\alpha$.

**Initialize:** $\ell = 1$ and $\mathcal{A}_1 = \mathcal{A}$.

1: Let $V_\ell(\pi) \triangleq \sum_{a \in \mathcal{A}_\ell} \pi(a) aa'$ and $g_\ell(\pi) \triangleq \max_{a \in \mathcal{A}_\ell} \|a\|^2_{V_\ell(\pi)^{-1}}$. Server solves an approximate G-optimal design problem to compute a distribution $\pi_\ell$ over $\mathcal{A}_\ell$ such that $g_\ell(\pi_\ell) \leq 2d$ and $|\text{Supp}(\pi_\ell)| \leq 48d \log \log d$.

2: For each $a \in \mathcal{A}_\ell$, server computes $m_a^{(\ell)}$ via Eq. (5), and broadcasts $\{m_a^{(\ell)}\}_{a \in \mathcal{A}_\ell}$ to all agents.

3: **for** $i \in [M] \setminus \mathcal{B}$ **do**

4:     For each arm $a \in \mathcal{A}_\ell$, pull it $m_a^{(\ell)}$ times. Let $r_{i,a}^{(\ell)}$ be the average of the rewards observed by agent $i$ for arm $a$ during phase $\ell$.

5:     Compute local estimate $\hat{\theta}_i^{(\ell)}$ of $\theta_*$ as follows.[4]

$$\hat{\theta}_i^{(\ell)} = \tilde{V}_\ell^{-1} Y_{i,\ell}, \quad \text{where } \tilde{V}_\ell = \sum_{a \in \text{Supp}(\pi_\ell)} m_a^{(\ell)} aa' \; ; \; Y_{i,\ell} = \sum_{a \in \text{Supp}(\pi_\ell)} m_a^{(\ell)} r_{i,a}^{(\ell)} a. \tag{3}$$

6:     Transmit $\hat{\theta}_i^{(\ell)}$ to server. Adversarial agents can transmit arbitrary vectors at this stage.

7: **end for**

8: Server computes robust mean pay-offs for each active arm: for each $a \in \mathcal{A}_\ell$, estimate $\mu_a^{(\ell)}$ as

$$\mu_a^{(\ell)} = \texttt{Median}\left(\{\langle \hat{\theta}_i^{(\ell)}, a \rangle, i \in [M]\}\right).$$

9: Define robust confidence threshold $\gamma_\ell \triangleq \sqrt{2} C \left(1 + \alpha \sqrt{M}\right) \epsilon_\ell$, where $C$ is as in Lemma 1. Server performs phased elimination with the robust means and $\gamma_\ell$ to update active arm set:

$$\mathcal{A}_{\ell+1} = \{a \in \mathcal{A}_\ell : \max_{b \in \mathcal{A}_\ell} \mu_b^{(\ell)} - \mu_a^{(\ell)} \leq 2\gamma_\ell\}. \tag{4}$$

10: $\ell = \ell + 1$ and **Goto** line 1.

---

**Description of `RCLB` (Algorithm 1).** The `RCLB` algorithm we propose is inspired by the phased elimination algorithm in [57, Chapter 22], but features some key differences due to the distributed and adversarial nature of our problem. The algorithm proceeds in epochs/phases, and in each phase $\ell$, the server maintains an active candidate set $\mathcal{A}_\ell$ of potential optimal arms. The exploration of arms in $\mathcal{A}_\ell$ is distributed among the agents. Upon such exploration, each agent $i$ reports back a local estimate $\hat{\theta}_i^{(\ell)}$ of the unknown parameter $\theta_*$; adversarial agents can transmit arbitrarily corrupted messages at this stage. Using the local parameter estimates $\{\hat{\theta}_i^{(\ell)}\}_{i \in [M]}$, the server then constructs (i) a *robust* estimate of the true mean payoff $\langle \theta_*, a \rangle$ for each active arm $a \in \mathcal{A}_\ell$, and (ii) an *inflated* confidence interval that captures both statistical and adversarial uncertainties associated with such robust mean estimates. This step is crucial to our scheme and requires a lot of care as we explain shortly. With the robust mean payoffs and associated confidence intervals in hand, the server eliminates arms with rewards far away from that of the optimal arm $a_*$. We now elaborate on the above ideas.

• *Optimal Experimental Design.* To minimize the regret incurred in each phase $\ell$, we need to minimize the number of arm-pulls made to arms in $\mathcal{A}_\ell$. To that end, we appeal to a concept from statistics known as optimal experimental design. The concept is as follows. Let $\pi : \mathcal{A} \to [0, 1]$ be a distribution on $\mathcal{A}$ such that $\sum_{a \in \mathcal{A}} \pi(a) = 1$. Now define $V(\pi) \triangleq \sum_{a \in \mathcal{A}} \pi(a) aa'$; $g(\pi) \triangleq \max_{a \in \mathcal{A}} \|a\|^2_{V(\pi)^{-1}}$. The so-called *G-optimal design problem* seeks to find a distribution (design) $\pi_*$ that minimizes $g$. In essence, sampling each arm $a \in \mathcal{A}$ in proportion to $\pi_*(a)$ minimizes the number of samples/arm-pulls needed to achieve a desired level of precision in the estimates of the arm means $\langle \theta_*, a \rangle, a \in \mathcal{A}$. For our purpose, we only need to solve an approximate G-optimal design problem: using the Frank-Wolfe algorithm and an appropriate initialization, one can *efficiently* find an approximate optimal design $\pi$ such that $g(\pi) \leq 2d$, and $|\text{Supp}(\pi)| \leq 48d \log \log d$ [58, Chapter 3]. Accordingly, the server computes such an approximate optimal distribution $\pi_\ell$ over $\mathcal{A}_\ell$ in each epoch $\ell$ (line 1 of Algo. 1).

---

[4]Throughout, we assume that $\tilde{V}_\ell^{-1}$ is invertible in each epoch $\ell$; this is the case when $\mathcal{A}_\ell$ spans $\mathbb{R}^d$. If $\mathcal{A}_\ell$ does not span $\mathbb{R}^d$, we can consider the lower dimensional subspace given by span$(\mathcal{A}_\ell)$.

● *Construction of Robust Arm-Payoff Estimates and Confidence Intervals.* In each phase $\ell$, the server computes $T_a^{(\ell)}$ and $m_a^{(\ell)}$ for every $a \in \mathcal{A}_\ell$ as follows:

$$T_a^{(\ell)} = \left\lceil \frac{\pi_\ell(a) d}{\epsilon_\ell^2} \log\left(\frac{1}{\delta_\ell}\right) \right\rceil \quad \text{and} \quad m_a^{(\ell)} = \left\lceil \frac{T_a^{(\ell)}}{M} \right\rceil, \tag{5}$$

where $\pi_\ell(a)$ is obtained from the approximate $G$-optimal design problem, $\epsilon_\ell = 2^{-\ell}$, $\delta_\ell = \bar{\delta}/(K\ell^2)$, and $\bar{\delta}$ is a design variable to be chosen later. The idea is to explore an arm $a$ $T_a^{(\ell)}$ times to estimate $\langle \theta_*, a \rangle$ up to a precision that scales linearly with $\epsilon_\ell$; in later phases, we require progressively finer precision (hence, $\epsilon_\ell$ decays exponentially with $\ell$). The task of exploration is distributed among the agents, with each agent $i$ being assigned $m_a^{(\ell)}$ arm-pulls for every $a \in \mathcal{A}_\ell$. Using the rewards that it observes during phase $\ell$, each good agent $i \in [M] \setminus \mathcal{B}$ computes a local estimate $\hat{\theta}_i^{(\ell)}$ of $\theta_*$, and transmits it to the server (lines 3-7 of Algo. 1). The key question now is as follows: *How should the server use the local estimates* $\{\hat{\theta}_i^{(\ell)}\}_{i \in [M]}$? Let us consider two natural strategies.

**Candidate Strategies.** One option could be to use the local estimates $\{\hat{\theta}_i^{(\ell)}\}_{i \in [M]}$ along with a high-dimensional robust mean estimation algorithm to compute a robust estimate of $\theta_*$. Yet another strategy could be for the server to query the raw observations (i.e., the $y_{i,t}$'s from the agents, use an univariate robust mean estimation algorithm (e.g., trimmed mean or median) to generate a "clean" version of each observation, and then use these clean observations to compute a robust estimate of $\theta_*$. Although feasible, each of the above strategies can unfortunately lead to an additional $\sqrt{d}$ factor in the regret bound; we discuss this point in detail in the Appendix. The main message we want to convey here is that certain natural candidate solutions can lead to sub-optimal regret bounds.

**Main Ideas.** Our main insight is the following: to achieve near-optimal optimal regret bounds, one need not go through the route of first computing a robust estimate of $\theta_*$. As our analysis will soon reveal, it suffices to instead compute robust estimates of the arm pay-offs $\{\langle \theta_*, a \rangle\}_{a \in \mathcal{A}_\ell}$ *directly* by employing the estimator in line 8 of Algo. 1. The key statistical property that we exploit here is that for each $a \in \mathcal{A}_\ell$ and $i \in [M] \setminus \mathcal{B}$, the quantity $\langle \hat{\theta}_i^{(\ell)}, a \rangle$ is conditionally-Gaussian with mean $\langle \theta_*, a \rangle$. This observation informs the choice of the median operator in line 8 of Algo. 1. Our next task is to compute appropriate confidence intervals for the robust arm-mean-estimates in order to eliminate sub-optimal arms. This is a delicate task as such confidence intervals need to account for both statistical uncertainties *and* adversarial perturbations. Indeed, if the confidence intervals are too tight, then they can lead to elimination of the optimal arm $a_*$; if they are too loose, then they can lead to large regret. The confidence threshold $\gamma_\ell$ in line 9 of Algo. 1 strikes just the right balance; the choice of such intervals is justified in Lemma 1.

We now state and discuss our main result concerning the performance of RCLB.

**Theorem 1.** *(**Performance of** RCLB) Suppose $\alpha \in [0, 0.5)$. Given any $\delta \in (0, 1)$, set $\bar{\delta} = \delta/(10K)$. Then, RCLB guarantees that with probability at least $1 - \delta$, the following holds for each good agent $i \in [M] \setminus \mathcal{B}$:*

$$\sum_{t=1}^{T} \langle \theta_*, a_* - a_{i,t} \rangle = \tilde{O}\left( \left( \alpha + \sqrt{1/M} \right) \sqrt{dT} \right). \tag{6}$$

**Main Takeaways.** From Theorem 1, we note that RCLB guarantees sublinear regret despite the presence of adversarial agents. More importantly, the regret bound in Eq. (6) has optimal dependence on the model-dimension $d$, the horizon $T$, and also on the number of agents $M$ when $\alpha = 0$ (i.e., in the absence of adversaries). When $\alpha$ is small, Eq. (6) reveals that one can indeed retain the benefits of collaboration, and improve upon the trivial per agent regret of $\tilde{O}(\sqrt{dT})$. Interestingly, this result mirrors that of a similar flavor for distributed stochastic optimization under attacks: given $M$ machines, $\alpha$ fraction of which are corrupt, the authors in [6] showed that no algorithm can achieve statistical error lower than $\tilde{\Omega}\left( \alpha/\sqrt{T} + 1/\sqrt{MT} \right)$ for strongly convex loss functions; here, $T$ is the number of samples on each machine. *As far as we are aware, this is the first work to establish an analogous result for collaborative linear bandits.* Inspired by the lower bound in [6], one may ask: *Is the additive $\alpha\sqrt{T}$ term in Eq. (6) unavoidable?* We will provide a definitive answer in Section 4. Finally, we note that Theorem 1 immediately implies a bound of $\tilde{O}\left( \left( \alpha M + \sqrt{M} \right) \sqrt{dT} \right)$ on the group regret $R_T$. We provide a formal proof of this fact in Appendix C.

**Proof Outline of Theorem 1.** The first main step in our analysis of Theorem 1 is to provide guarantees on the estimates $\{\mu_a^{(\ell)}\}_{a \in \mathcal{A}_\ell}$ computed in line 8 of RCLB. This is achieved as follows.

**Lemma 1.** *(Robust Confidence Intervals) Fix any epoch $\ell$. There exists an universal constant $C > 0$ such that for each active arm $b \in \mathcal{A}_\ell$, the following holds with probability at least $1 - \delta_\ell$:*

$$|\mu_b^{(\ell)} - \langle \theta_*, b \rangle| \leq \gamma_\ell, \ \ where \ \ \gamma_\ell = \sqrt{2}C\left(1 + \alpha\sqrt{M}\right)\epsilon_\ell. \tag{7}$$

Equipped with the above result, we argue that with high-probability, (i) the optimal arm $a_*$ is never eliminated by RCLB (Lemma 4 in App. C); and (ii) in each epoch $\ell$, an active arm in $\mathcal{A}_\ell$ can contribute to at most $O(\gamma_\ell)$ per-time-step regret (Lemma 5 in App. C). Putting these pieces together in a careful manner yields the desired result; we defer a detailed proof of Theorem 1 to Appendix C.

In the next section, we derive a lower bound that provides fundamental insights into the impact of the adversarial agents for the multi-agent sequential decision-making problem considered in this paper.

## 4 Lower Bounds

In this section, we assess the optimality of the regret bound obtained in Theorem 1. To do so, we consider a slightly different attack model: we assume that each of the agents is adversarial with probability $\alpha$, independently of the other agents. Thus, the expected fraction of adversaries is $\alpha$. Next, to provide a clean argument, we will focus on a class of policies $\Pi$ where at each time-step $t$, the server assigns the *same* action to every agent, i.e., $a_{i,t} = a_t, \forall i \in [M]$. We note that all the algorithms developed in this paper adhere to policies in $\Pi$. Moreover, policies within the class $\Pi$ yield the minimax optimal per-agent regret bound of order $\tilde{O}(\sqrt{dT}/\sqrt{M})$ when $\alpha = 0$. Since the standard multi-armed bandit setting [59] is a special case of the structured linear bandit setting considered here, a lower bound for the former implies one for the latter. With this in mind, let us denote by $\mathcal{E}_\mathcal{N}^{(K)}(1)$ the class of multi-armed bandits with $K$ arms, where the reward distribution of each arm is Gaussian with unit variance. An instance $\nu_\mu \in \mathcal{E}_\mathcal{N}^{(K)}(1)$ is characterized by the mean vector $\mu \in \mathbb{R}^K$ associated with the $K$ arms. Finally, let $R_T^{(s)}(\nu_\mu)$ denote the expected cumulative regret of the server (which is the same as that of a good agent $i \in [M] \setminus \mathcal{B}$) when it interacts with the instance $\nu_\mu$. We can now state the following result which establishes a fundamental lower bound for our problem.

**Theorem 2.** *(Fundamental Lower Bound) Given any policy in $\Pi$, there exist two distinct instances $\nu_\mu, \nu_{\mu'} \in \mathcal{E}_\mathcal{N}^{(2)}(1)$, and an universal constant $c > 1$, such that*

$$\max\{R_T^{(s)}(\nu_\mu), R_T^{(s)}(\nu_{\mu'})\} \geq c\alpha\sqrt{T}, \tag{8}$$

*irrespective of the number of agents $M$.*

**Main Takeaways.** Observe from Eq. (6) that even when $M$ is arbitrarily large, the additive $\alpha\sqrt{T}$ term due to the adversaries remains unaffected - *Is this term truly unavoidable or just an artifact of our analysis?* Theorem 2 settles this question by revealing a fundamental performance limit: *every policy* in $\Pi$ has to suffer the additive $\alpha\sqrt{T}$ regret, regardless of the number of agents. Thus, taken together, *Theorems 1 and 2 provide the first set of tight, near-optimal regret guarantees for the setting considered in this paper. We consider this to be a significant contribution of our work.*

**Proof Idea for Theorem 2.** For our setting, the standard techniques to prove lower bounds for non-adversarial bandits do not directly apply. The lower-bound proofs used for reward-corruption models [60], and attacks with a fixed budget [31], are not applicable either. This motivates us to use a new proof technique that combines information-theoretic arguments in [17] with ideas from the robust mean estimation literature [18, 19]. Specifically, for a two-armed bandit setting, we carefully construct two instances and attack strategies such that the joint distribution of rewards seen by the server is *identical* for both instances. Moreover, the instances are constructed such that (i) the optimal arm in one instance is sub-optimal for the other; and (ii) the per-time-step regret for selecting a sub-optimal arm in either instance is $\Omega(\alpha/\sqrt{T})$. For a detailed proof, see Appendix D.

Having established tight bounds for the linear bandit model in Section 2, in the sequel, we will show how our algorithmic ideas and results can be significantly extended to more general settings.

# 5 Extension to Generalized Linear Models with Adversaries

In this section, we will show how to achieve a regret bound akin to that in Theorem 1 for the non-linear observation model shown below [20, 21], known as the generalized linear model (GLM):

$$y_{i,t} = \mu\left(\langle \theta_*, a_{i,t} \rangle\right) + \eta_{i,t}, \tag{9}$$

where $\mu : \mathbb{R} \to \mathbb{R}$ is a continuously differentiable function typically referred to as the (inverse) *link* function, and $\eta_{i,t} \sim \mathcal{N}(0,1)$ is as before. Our goal is to now control the following notion of regret:

$$R_T^{\text{GLM}} = \mathbb{E}\left[ \sum_{i \in [M] \setminus \mathcal{B}} \sum_{t=1}^{T} \left(\mu\left(\langle \theta_*, a_* \rangle\right) - \mu\left(\langle \theta_*, a_{i,t} \rangle\right)\right) \right], \tag{10}$$

where $a_* = \operatorname{argmax}_{a \in \mathcal{A}} \mu(\langle \theta_*, a \rangle)$. The main technical challenge relative to the setting considered in Section 2 pertains to the construction of the robust confidence intervals. In particular, the non-linearity of the map $\mu(\cdot)$ makes it hard to apply the technique adopted in line 8 of RCLB, necessitating a different approach that we describe next. We start with the following standard assumption [20, 21].

**Assumption 1.** *The function $\mu : \mathbb{R} \to \mathbb{R}$ is continuously differentiable, Lipschitz with constant $k_2 \geq 1$, and such that*

$$k_1 = \min\{1, \inf_{\theta \in \Theta, a \in \mathcal{A}} \dot{\mu}(\langle \theta, a \rangle)\} > 0.$$

*Here, $\dot{\mu}(\cdot)$ is used to represent the derivative of $\mu(\cdot)$.*

Next, for any $\theta \in \mathbb{R}^d$, we define $h_\ell(\theta) \triangleq \sum_{a \in \operatorname{Supp}(\pi_\ell)} m_a^{(\ell)} \mu(\langle \theta, a \rangle) a$. We now describe a variant of RCLB - dubbed RC-GLM - for generalized linear models. Notably, unlike the standard bandit algorithms [20, 21] for GLM's that build on LinUCB, RC-GLM is based on phased elimination.

**Description of RC-GLM.** Our algorithm uses as a sub-routine the recently proposed Iteratively Reweighted Mean Estimator for computing a robust estimate of the mean of high-dimensional Gaussian random variables with adversarial outliers [22]. Specifically, suppose we are given $M$ $d$-dimensional samples $x_1, \ldots, x_M$, such that $(1-\alpha)M$ of these samples are drawn i.i.d. from $\mathcal{N}(v, \Sigma)$, where $v \in \mathbb{R}^d$ is an unknown mean vector, and $\Sigma \in \mathbb{R}^{d \times d}$ is a known covariance matrix. The remaining $\alpha M$ samples are adversarial outliers and can be arbitrary. The estimator in [22] takes as input the $M$ samples, the corruption fraction $\alpha$, and the covariance matrix $\Sigma$. It then outputs an estimate $\hat{v}$ of $v$ such that with high probability, $\|\hat{v} - v\| = \tilde{O}\left(\|\Sigma\|_2^{1/2}\left(\sqrt{d/M} + \alpha\sqrt{\log(1/\alpha)}\right)\right)$. Importantly, the estimator in [22] runs in polynomial-time, and is minimax-rate-optimal. Let us now see how this estimator - described in Appendix E - can be applied to our setting.

We only describe the key differences of RC-GLM relative to RCLB here, and defer a detailed description of RC-GLM to Appendix E. To get around the difficulty posed by the non-linear link function, our main idea is to first compute a robust estimate $\hat{\theta}^{(\ell)}$ of $\theta_*$ at the server, and then use it to develop a phased elimination strategy. To that end, instead of computing a local estimate $\hat{\theta}_i^{(\ell)}$ as in RCLB, each good agent $i \in [M] \setminus \mathcal{B}$ transmits $Y_{i,\ell}$ to the server, and the server computes a vector $X_\ell$ as follows: $X_\ell = \text{ITW}(\{\tilde{V}_\ell^{-1/2} Y_{i,\ell}, i \in [M]\})$. Here, $\tilde{V}_\ell^{-1/2}$ and $Y_{i,\ell}$ are as in Eq. (3), and we used $\text{ITW}(\cdot)$ to denote the output of the robust estimator in [22]. Our key observation here is that for each good agent $i$, $\tilde{V}_\ell^{-1/2} Y_{i,\ell}$ is a $d$-dimensional Gaussian random variable with mean $\tilde{V}_\ell^{-1/2} h_\ell(\theta_*)$, and covariance matrix $\Sigma = I_d$, justifying the use of the robust Gaussian mean estimator in [22]. The fact that $\Sigma = I_d$ is crucial in our algorithm design as the error-bound in [22] scales with the 2-norm of $\Sigma$. Essentially, the above steps enable us to extract a statistic $X_\ell$ that captures information about the agents' observations during epoch $\ell$. Using this statistic, the server next computes an estimate $\hat{\theta}^{(\ell)}$ of $\theta_*$ by solving $h_\ell(\hat{\theta}^{(\ell)}) = \tilde{V}_\ell^{1/2} X_\ell$,[5] and employs the following phased elimination strategy:

$$\mathcal{A}_{\ell+1} = \{a \in \mathcal{A}_\ell : \max_{b \in \mathcal{A}_\ell} \mu(\langle \hat{\theta}^{(\ell)}, b \rangle) - \mu(\langle \hat{\theta}^{(\ell)}, a \rangle) \leq 2\bar{\gamma}_\ell\}; \quad \bar{\gamma}_\ell = \bar{C}(k_2/k_1)\left(\sqrt{d} + \alpha\sqrt{M\log(1/\alpha)}\right)\epsilon_\ell,$$

where $\bar{C}$ is an universal constant known to the server. Deriving an analogue of Lemma 1 to compute the robust confidence threshold $\bar{\gamma}_\ell$ requires some work. This is achieved by exploiting the regularity properties of the link function in tandem with the confidence bounds in [22]; see Appendix E for details and a proof of our main result for RC-GLM stated below.

---

[5]We argue in Appendix E that this equation admits a unique solution.

---

**Algorithm 2** Robust BaseLinUCB (at Server)

---

**Input:** Confidence parameter $\bar{\delta}$, corruption fraction $\alpha$, and index set $\psi_t \subseteq [t-1]$.

1: $A_t \leftarrow \frac{I_d}{M} + \sum_{\tau \in \psi_t} x_{\tau,a_\tau} x'_{\tau,a_\tau}$.
2: **for** $i \in [M]$ **do**            ▷ Compute local parameters for each agent
3:     $b_{i,t} \leftarrow \sum_{\tau \in \psi_t} r_{i,\tau} x_{\tau,a_\tau}; \hat{\theta}_{i,t} \leftarrow A_t^{-1} b_{i,t}$.
4: **end for**
5: **for** $a \in [K]$ **do**            ▷ Compute robust estimates for each feature vector
6:     $\hat{r}_{t,a} \leftarrow \texttt{Median}\left(\{\langle \hat{\theta}_{i,t}, x_{t,a}\rangle, i \in [M]\}\right); w_{t,a} \leftarrow \left(\alpha + 2C\sqrt{\frac{\log(\frac{1}{\bar{\delta}})}{M}}\right)\|x_{t,a}\|_{A_t^{-1}}$, where
    $C$ is as in Lemma 1.
7: **end for**

---

**Theorem 3.** (***Performance of Algorithm*** `RC-GLM`) *Suppose* $\alpha < (5 - \sqrt{5})/10$*, and* $M = \Omega(\log(KT))$. *Given any* $\delta \in (0,1)$, `RC-GLM` *guarantees that with probability at least* $1 - \delta$, *the following holds for each good agent* $i \in [M] \setminus \mathcal{B}$:

$$\sum_{t=1}^{T} (\mu(\langle \theta_*, a_*\rangle) - \mu(\langle \theta_*, a_{i,t}\rangle)) = \tilde{O}\left((k_2/k_1)\left(\alpha\sqrt{\log(1/\alpha)} + \sqrt{d/M}\right)\sqrt{dT}\right). \qquad (11)$$

**Main Takeaways.** Theorem 3 significantly generalizes Theorem 1 and shows that even for general non-linear observation maps, one can reap the benefits of collaboration in the presence of adversaries. The additional $\sqrt{d}$ factor in the bound of (11) relative to that in (6) is inherited from the error-rate guarantees in [22]; the task of tightening this bound is left as future work. Nonetheless, Theorem 3 is the only result we are aware of that provides adversarial-robustness guarantees for GLMs.

**Remark 1.** (***Communication Complexity of*** `RCLB` ***and*** `RC-GLM`) *It is not hard to see that the number of epochs/phases in* `RCLB` *and* `RC-GLM` *is* $O(\log(MT))$. *Since communication between the server and the agents occurs only once in every epoch, we note that the communication complexity of these algorithms scale logarithmically with the horizon* $T$. *Thus, our proposed algorithms not only lead to near-optimal regret bounds in the face of worst-case adversarial attacks, they are also communication-efficient by design. This is an important point to take note of as communication-efficiency is a key consideration in large-scale computing paradigms such as federated learning.*

## 6 Robust Collaborative Contextual Bandits with Adversaries

In this section, we will consider a collaborative contextual bandit setting where at each time-step $t \in [T]$, the server and the agents observe $K$ $d$-dimensional feature vectors, $\{x_{t,a}|a \in [K]\}$, with $\|x_{t,a}\| \leq 1, \forall a \in [K]$ and $\forall t \in [T]$. We assume that the adversarial agents have no control over the generation of the feature vectors. Associated with each arm $a \in [K]$, the stochastic reward observed by an agent $i \in [M]$ comes from the following observation model:

$$y_{i,t}(a) = \langle \theta_*, x_{t,a}\rangle + \eta_{i,t}(a), \qquad (12)$$

where $\{\eta_{i,t}(a)\}$ are drawn i.i.d. from $\mathcal{N}(0,1)$. At each time step $t$, a good agent $i \in [M] \setminus \mathcal{B}$ plays an action $a_{i,t} \in [K]$, and receives the corresponding reward $r_{i,t} \triangleq y_{i,t}(a_{i,t})$ based on the observation model in (12). The main difference of the setting considered here relative to the one in Section 2 is that the feature vectors for each arm can change over time. As a result, the optimal action $a_t^* = \text{argmax}_{a \in [K]} \langle \theta_*, x_{t,a}\rangle$ can change over time, making it particularly challenging to compete with a time-varying optimal action in the presence of adversaries. This dictates the need for a different algorithmic strategy compared to the one we developed in Section 1. Before we develop such a strategy, let us first formally define the performance metric of interest to us in this setting:

$$R_T^{\text{Context}} = \mathbb{E}\left[\sum_{i \in [M] \setminus \mathcal{B}} \sum_{t=1}^{T} \langle \theta_*, x_{t,a_t^*} - x_{t,a_{i,t}}\rangle\right]. \qquad (13)$$

The main question we ask is: *For the contextual bandit setting described above, can one continue to hope for benefits of collaboration in the presence of adversaries?* In what follows, we will answer this question in the affirmative by developing a variant of the `SUPLINREL` algorithm in [24].

**Algorithm 3** Robust Collaborative SupLinUCB for Contextual Bandits (at Server)

---

**Input:** Confidence parameter $\delta$, corruption fraction $\alpha$, and horizon $T$.

1: $S \leftarrow \lceil \ln T \rceil; \psi_1^{(s)} \leftarrow \emptyset, \forall s \in [S]$.
2: **for** $t \in [T]$ **do**
3:      $s \leftarrow 1$ and $\mathcal{A}_1 \leftarrow [K]$.
4:      **repeat**
5:          Use Algorithm 2 with $\bar{\delta} = \delta/(KST)$, and index set $\psi_t^{(s)}$ to compute robust estimates $\{\hat{r}_{t,a}^{(s)}, w_{t,a}^{(s)}\}$ of the means and variances of the payoffs associated with each arm $a \in \mathcal{A}_s$.
6:          If $w_{t,a}^{(s)} > 2^{-s}/\sqrt{M}$ for some $a \in \mathcal{A}_s$, then choose this arm, i.e., set $a_t = a$. Store the corresponding phase: $\psi_{t+1}^{(s)} = \psi_t^{(s)} \cup \{t\}, \psi_{t+1}^{(\ell)} = \psi_t^{(\ell)} \ \forall \ell \neq s$.
7:          Else if $w_{t,a}^{(s)} \leq 1/\sqrt{MT} \ \forall a \in \mathcal{A}_s$, then select the arm with the *highest robust upper confidence bound*: $a_t = \mathrm{argmax}_{a \in \mathcal{A}_s} \left( \hat{r}_{t,a}^{(s)} + w_{t,a}^{(s)} \right)$. Do not store this phase, i.e., $\psi_{t+1}^{(\ell)} = \psi_t^{(\ell)} \ \forall \ell \in [S]$.
8:          Else if $w_{t,a}^{(s)} \leq 2^{-s}/\sqrt{M} \ \forall a \in \mathcal{A}_s$, then update active arm-set as

$$\mathcal{A}_{s+1} = \{a \in \mathcal{A}_s | \max_{b \in \mathcal{A}_s} \left( \hat{r}_{t,b}^{(s)} + w_{t,b}^{(s)} \right) - \left( \hat{r}_{t,a}^{(s)} + w_{t,a}^{(s)} \right) \leq 2^{(1-s)}/\sqrt{M} \}.$$

9:          $s \leftarrow s + 1$.
10:      **until** an action $a_t$ is chosen.
11:      Broadcast the chosen action $a_t$ to every agent (i.e., $a_{i,t} = a_t, \forall i \in [M]$), and receive corresponding rewards $\{r_{i,t}\}_{i \in [M]}$. Adversarial agents can transmit arbitrary reward values.
12: **end for**

---

**Description of Algorithm 3.** At each time-step $t$, our proposed algorithm, namely Algorithm 3, scans through the set $\mathcal{A}$ of arms to determine a suitable action $a_t$. This scanning process (lines 4-10 of Algo. 3) is done at the server over $S$ phases. Corresponding to each phase $s \in [S]$, the server maintains a set $\psi_t^{(s)}$; the set $\psi_t^{(s)}$ stores all the time-steps in $[t-1]$ where an action is chosen in phase $s$ of the scanning process based on line 6 on Algo. 3. The scanning process itself relies on Algorithm 2 as a sub-routine. Specifically, in each phase $s$, the server first invokes Algorithm 2 to obtain a robust estimate $\hat{r}_{t,a}^{(s)}$ of $\langle \theta_*, x_{t,a} \rangle$ for each arm $a \in \mathcal{A}_s$, along with an associated *inflated* confidence width $w_{t,a}^{(s)}$ (line 5 of Algo. 3). If the confidence width is too large for a particular arm (as in line 6), then such an arm requires exploration and is accordingly chosen to be $a_t$. If, on the other hand, the confidence widths of all arms are sufficiently small (as in line 7), then $a_t$ is chosen to be the arm with the highest upper-confidence bound. Thus, we follow the principle of *optimism in the face of uncertainty here, while exercising caution to account for the presence of adversaries (via the use of inflated confidence intervals)*. If the conditions in lines 6 and 7 both fail, then the arms in $\mathcal{A}_s$ require further screening. Accordingly, we move to the next phase $s + 1$, retaining only those arms that are sufficiently close to the optimal arm $a_t^*$; see line 8 of Algo. 3. Our main innovation lies in (i) the construction of the robust arm-estimates in Algo. 2 that account for both statistical and adversarial behavior, and (ii) the careful use of such estimates in lines 6-8 of Algo. 3 to pick the action $a_t$. The next result reveals that the combination of these ideas yields near-optimal regret bounds.

**Theorem 4.** *(**Performance of Algo. 3**) Suppose $\alpha \in (0, 0.5)$. Given any $\delta \in (0, 1)$, Algo. 3 guarantees that with probability at least $1 - \delta$, the following holds for each good agent $i \in [M] \setminus \mathcal{B}$:*

$$\sum_{t=1}^{T} \langle \theta_*, x_{t,a_t^*} - x_{t,a_{i,t}} \rangle = \tilde{O} \left( \left( \alpha + \sqrt{1/M} \right) \sqrt{dT} \right). \tag{14}$$

**Main Takeaways.** For the contextual bandit setting considered here, the single-agent minimax optimal regret in the absence of adversaries is $\tilde{O}(\sqrt{dT})$ [24]. In light of the lower bound in Theorem 2, we see that Theorem 4 provides a near-optimal regret guarantee, just as Theorem 1.

**Remark 2.** *For ease of exposition, we have considered Gaussian noise (in the observation model) throughout the paper. However, both our algorithms and results can be extended with slight modifications to sub-Gaussian noise sequences. We elaborate on this point in Appendix B.*

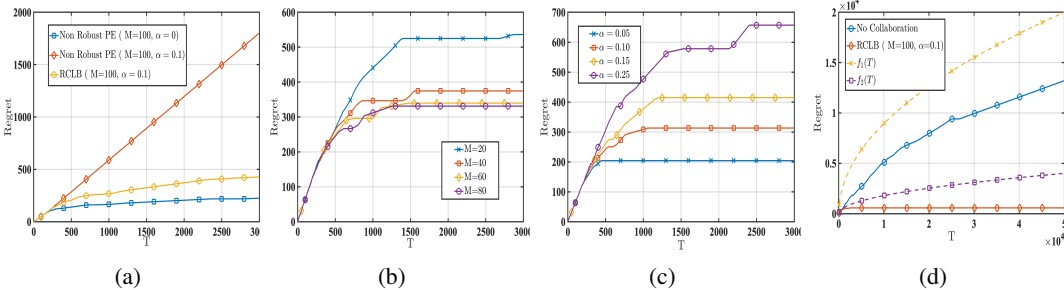

Figure 1: Plots of per-agent regret for the linear bandit experiment. (a) Comparison between RCLB and a vanilla non-robust phased elimination algorithm. (b) Performance of RCLB for varying number of agents $M$, with $\alpha = 0.1$. (c) Performance of RCLB for varying corruption fraction $\alpha$, with $M = 100$. For (d), we set $\alpha = 0.1$, $M = 100$, and compare RCLB to a phased elimination algorithm where the agents do not collaborate. We also plot theoretical upper-bounds: $f_1(T) = 40\sqrt{dT}$ and $f_2(T) = 40(\alpha + \sqrt{(1/M)})\sqrt{dT}$.

## 7 Simulation Results

We report simulation results on synthetic data to corroborate our developed theory. Additional simulations on contextual bandits and alternate attack models are presented in Appendix H.

**Experimental Setup.** We consider a setting with 50 actions in $\mathbb{R}^5$; we describe how these actions and $\theta^*$ are generated in Appendix H. The rewards are generated based on the observation model in Eq. (1). We now describe the attack model. To manipulate the server into selecting sub-optimal arms, each adversarial agent $i$ employs the simple strategy of reducing the rewards of the good arms and increasing the rewards of the bad arms. More precisely, in each epoch $\ell$, upon pulling an arm $a$ and observing the corresponding reward $y_a$, an adversarial agent $i$ does the following: if $y_a > p\langle\theta_*, a_*\rangle$, then this reward is corrupted to $\tilde{y}_a = y_a - \beta$; and if $y_a \leq p\langle\theta_*, a_*\rangle$, then the reward is corrupted to $\tilde{y}_a = y_a + \beta$. For this experiment, we fix $p = 0.6$ and $\beta = 5$. Agent $i \in \mathcal{B}$ then uses all the corrupted rewards in epoch $\ell$ to generate the local model estimate $\hat{\theta}_i^{(\ell)}$ that is transmitted to the server.

**Discussion of Simulation Results.** Fig. 1 summarizes our experimental results. In Fig. 1(a), we compare our proposed algorithm RCLB to a vanilla distributed phased elimination (PE) algorithm that does not account for adversarial agents. Specifically, the latter is designed by replacing the median operation in line 8 of Algorithm 1 with a mean operation, and setting the threshold $\gamma_\ell$ in line 9 to be $\epsilon_\ell$. Fig. 1(a) shows that even a small fraction $\alpha = 0.1$ of adversaries can cause the non-robust PE algorithm to incur linear regret. In contrast, RCLB continues to guarantee sub-linear regret bounds despite adversarial corruptions. Furthermore, the regret bound of RCLB in the presence of a small fraction of adversarial agents is close to that of the non-robust algorithm in the absence of adversaries. This goes on to establish the robustness of RCLB.

Fig. 1(b) depicts the performance of RCLB for varying values of the number of agents $M$, at a fixed corruption level $\alpha = 0.1$. We observe that increasing $M$ results in lower regret, indicating a clear benefit of collaboration despite the presence of adversaries. In Fig. 1(c), we vary the corruption fraction $\alpha$, keeping $M$ fixed at 100. As expected, increasing $\alpha$ leads to higher (albeit sub-linear) regret. Importantly, the trends observed in both Fig. 1(b) and Fig. 1(c) are consistent with the theoretical upper-bound of $O((\alpha + 1/\sqrt{M})\sqrt{dT})$ predicted by Theorem 1.

A trivial way to avoid adversarial corruption is for a good agent to not participate in any collaboration at all, and run a standard single-agent bandit algorithm. This would result in such an agent incurring $O(\sqrt{dT})$ regret. The purpose of Fig. 1(d) is to drive home the point that RCLB *can lead to significant improvements over a trivial non-collaborative strategy*. To make this point clear, we compare RCLB to a standard single-agent phased elimination algorithm that does not involve any collaboration, and observe that despite adversarial corruption, RCLB leads to considerably lower regret bounds as compared to the non-collaborative strategy. This highlights the importance of our approach.

**Acknowledgments and Disclosure of Funding**. This work was supported by NSF Award 1837253, NSF CAREER award CIF 1943064, the AFOSR-YIP award FA9550-20-1-0111, and EnCORE.

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
