## A Detailed Discussion of Related Work

Below, we provide a detailed discussion of relevant work.

• **Reward Corruption Attacks in Stochastic Bandits.** In the single-agent setting, there is a rich body of work that studies the effect of reward-corruption in stochastic bandits, both for the unstructured multi-armed bandit problem [25–28], and also for structured linear bandits [29–32]. In these works, an adversary can modify the true stochastic reward/feedback on certain rounds; a corruption budget $C$ captures the total corruption injected by the adversary over the horizon $T$. The attack model we study is fundamentally different: the adversaries in our setting can inject corruptions of *arbitrary* magnitude in *all* rounds, i.e., there are no budget constraints. As such, the algorithmic techniques in [25–31] do not apply to our model.

Continuing with this point, we note that in [28], the authors proved an *algorithm-independent* lower bound of $\Omega(C)$ on the regret. This lower bound suggests that for the reward-corruption attack model, when the attacker's budget $C$ scales linearly with the horizon $T$, there is no hope for achieving sub-linear regret. In [60], the authors studied a reward-corruption model closely related to those in [27–29], where in each round, with probability $\eta$ (independently of the other rounds), the attacker can bias the reward seen by the learner. Similar to the lower bound in [28], the authors in [60] proved a lower bound of $\Omega(\eta T)$ on the regret for their model. In sharp contrast to the fundamental limits established in [28, 60], for our setup, as long as the corruption fraction $\alpha$ is strictly less than half, we prove that with high-probability it is in fact possible to achieve sub-linear regret. The key is that for our setting, the server can leverage "clean" information from the good agents in every round; of course, the identities of such good agents are not known to the server. We finally note that beyond the task of minimizing cumulative regret, the impact of fixed-budget reward-contamination has also been explored for the problem of best-arm identification in [61].

• **Multi-Agent Bandits.** There is a growing literature that studies multi-agent multi-armed bandit problems in the absence of adversaries, both over peer-to-peer networks, and also for the server-client architecture model [33–52]. The main focus in these papers is the design of coordination protocols among the agents that balance communication-efficiency with performance. A few very recent works [53–56] also look at the effect of attacks, but for the simpler unstructured multi-armed bandit problem [59]. Accounting for adversarial agents in the structured linear bandit setting we consider here requires significantly different ideas that we develop in this paper. At this point, we should mention that the concurrent work of [62] looks at a contextual bandit model somewhat different from what we study; they identify fundamental performance limits and design efficient robust algorithms.

• **Security in Distributed Optimization and Federated Learning.** As we mentioned earlier, several papers have studied the problem of accounting for adversarial agents in the context of supervised learning [4–10, 63, 11–13, 64–66, 14, 67]. One of the primary applications of interest here is the emerging paradigm of federated learning [1–3]. Different from the sequential decision-making setting we investigate in our paper, the aforementioned works essentially abstract out the supervised learning task as a *static* distributed optimization problem, and then apply some form of secure aggregation on either gradient vectors or parameter estimates.

• **Robust Statistics.** The algorithms that we develop in this paper borrow tools from the literature on robust statistics, pioneered by Huber [68, 69]. We point the reader to [18, 19, 70–72, 22], and the references therein, to get a sense of some of the main results in this broad area of research. In a nutshell, given multiple samples of a random variable - with a small fraction of samples corrupted by an adversary - the essential goal of this line of work is to come up with statistically optimal and computationally efficient robust estimators of the mean of the random variable. Notably, unlike both the sequential bandit setting and the iterative optimization setting, the robust statistics literature focuses on one-shot estimation. In other words, the adversary gets to corrupt the batch of samples *only once*, and the effect of such corruption does not compound over time or iterations.

# B  Further Comments on our Algorithms

In what follows, we comment on certain key aspects of our proposed algorithms.

• **On the knowledge of the corruption fraction.** In practice, if we do not know the corruption fraction $\alpha$, but have access to an upper bound on $\alpha$, say $\tilde{\alpha}$, we can essentially use $\tilde{\alpha}$ as a proxy for $\alpha$, both in our algorithms and their analyses. In this case, all our results go through identically by simply replacing $\alpha$ with $\tilde{\alpha}$ in the bounds. However, if we have no idea at all about $\alpha$, then it is not clear whether one can design a robust algorithm that achieves minimax-optimal rates for the setting we consider in this paper. The main reason is as follows. Note that although computing a median does not require knowledge of $\alpha$, the robust confidence threshold we use in line 9 of Algorithm 1 critically relies on the knowledge of the corruption fraction $\alpha$ (or an upper-bound on it). Moreover, the correctness of our overall approach, and the fact that it is minimax-optimal, relies heavily on the tightness of the robust confidence thresholds in Lemma 1. Thus, at the moment, we do not know of a way that can achieve tight regret bounds without any knowledge whatsoever of $\alpha$; investigating this aspect further is an interesting topic of future research.

• **Beyond Gaussian Noise.** In what follows, we explain in detail that *both our algorithms and analysis apply, with slight modifications, to both sub-Gaussian noise, and more generally, noise with bounded variance. The reason why we chose Gaussian noise was primarily for ease of exposition.* Suppose the noise samples are i.i.d. with zero-mean (a standard assumption) and unit variance (the argument we present next trivially extends to bounded variance $\sigma^2$). To deal with such general noise distributions, we need to make two minor changes to Algorithm 1. First, we replace the Median operation in line 8 of Algorithm 1 with the scalar univariate trimmed mean estimator recently proposed in [72]. The estimator in [72] is *easily implementable and minimax-optimal*. Second, we set the robust threshold $\gamma_\ell$ in line 9 to be $\gamma_\ell = C(1 + \sqrt{\alpha M})\epsilon_\ell$, where $C$ is a suitably large universal constant; note that the only thing that has changed from before is the replacement of $\alpha$ by $\sqrt{\alpha}$ in $\gamma_\ell$. We will justify the choice of this threshold shortly, but before that, we mention the implications.

**Implications.** With the two minor modifications to Algorithm 1 described above, we can establish an analogue of Theorem 1 where for each good agent, with probability at least $1 - \delta$, the regret is bounded above by $\tilde{O}\left(\left(\sqrt{\alpha} + \sqrt{1/M}\right)\sqrt{dT}\right)$. Compared to the bound in Theorem 1, we note that our new bound is worsened by the replacement of $\alpha$ with $\sqrt{\alpha}$ - this is the price paid to account for general non-Gaussian distributions. For robust mean estimation with general noise distributions (with bounded fourth moments), the additive $\sqrt{\alpha}$ factor is *unavoidable*; see [72], for instance. Thus, we conjecture that the bound on regret we mentioned above is also minimax-optimal.

For sub-Gaussian distributions, we can achieve a tighter regret bound: by setting the robust confidence threshold to be $\gamma_\ell = C(1 + \alpha\sqrt{\log(1/\alpha)M})\epsilon_\ell$, we can achieve a per-agent regret bound of $\tilde{O}\left(\left(\alpha\sqrt{\log(1/\alpha)} + \sqrt{1/M}\right)\sqrt{dT}\right)$. *Hence, with sub-Gaussian noise, we essentially recover the same bounds as with Gaussian noise (up to logarithmic factors).* It is very likely that with sub-Gaussian noise, the median will continue to yield the bound above, i.e., we may not even need the univariate trimmed mean estimator from [72]. However, we do not have a concrete proof of this fact yet.

**Changes in Analysis.** We now go over the minor changes that need to be made to the analysis of Theorem 1 in view of replacing the median operator by the trimmed mean estimator in [72]. We first need an analog of Lemma 2 in Appendix C. This is supplied by Theorem 1 in [72] that provides guarantees on the univariate trimmed mean estimator. Suppose we are given $M$ data samples, where all the good samples are i.i.d. with mean $\mu$ and variance $\sigma^2$. Moreover, suppose the fraction of bad samples is at most $1/2$. With probability at least $1 - \delta$, we then have

$$|\hat{\mu} - \mu| \leq C\left(\sqrt{\alpha} + \sqrt{\frac{\log(1/\delta)}{M}}\right)\sigma,$$

where $\hat{\mu}$ is the output of the univariate trimmed mean estimator. Equipped with this result, one can follow the *exact same steps* as in the proof of Lemma 1 to justify the choice of the robust confidence threshold $\gamma_\ell = C(1 + \sqrt{\alpha M})\epsilon_\ell$. When the noise is additionally sub-Gaussian, the above bound can be tightened by replacing $\sqrt{\alpha}$ with $\alpha\sqrt{\log(1/\alpha)}$. The rest of the proof goes through *identically*.

To sum up, we have argued that with minor modifications to both our algorithm and analysis, our overall approach can handle both sub-Gaussian noise, and noise with bounded variance.

• **Is it possible to tolerate a corruption fraction** $\alpha > 0.5$? The key question here is: *Who is the learner? The agent or the server?* As we explain shortly, the answer to this question has significant implications for whether or not we can tolerate the $\alpha > 0.5$ case. Suppose the learners are the agents, in that each good agent is capable of taking their own decisions (actions). If it is known ahead of time that $\alpha > 0.5$, then acting alone is the most natural thing to do. This is because there is nothing to be gained by collaborating: no robust aggregation scheme can provide any guarantees when $\alpha > 0.5$; indeed, $\alpha = 0.5$ is a *fundamental breakdown point* when one considers an *arbitrary* corruption model. On the other hand, when $\alpha < 0.5$, one can significantly improve upon the trivial per-agent regret of $O(\sqrt{dT})$ by using the approach developed in our paper. This is revealed not only by our theory, but the simulations that we report in Section 7.

Now suppose the learner is the server, i.e., *the actions are decided by the server*, and the agents interact with an unknown environment and report certain relevant statistics to the server. As a concrete example, consider a web-advertising example where the server displays ads (actions) to a group of people (agents), assesses their (potentially corrupted) feedback, and then decides upon subsequent ad displays to maximize click-through rates. Crucially, unlike the previous setting (where the learners were the agents), the only way the server can acquire feedback about the environment is by interacting with the agents (some of whom might be adversarial). *In other words, the server does not have at its disposal the trivial option of not interacting with the agents, and acting alone.* Since in this setting, the server is forced to interact with potentially corrupted agents, $\alpha < 0.5$ is the only scenario that can lead to meaningful regret bounds. More precisely, when $\alpha > 0.5$, there is no hope of achieving sub-linear regret, let alone benefiting from collaboration.

# C   Analysis of RCLB: Proof of Theorem 1

In this section, we will prove Theorem 1. We start with a standard result from robust statistics on the guarantees afforded by the median operator for robust mean estimation of univariate Gaussian random variables; see, for instance, [19].

**Lemma 2.** *Consider a set $\mathcal{S} = \{x_1, \ldots, x_M\}$ of $M$ samples partitioned as $\mathcal{S} = \mathcal{S}_g \cup \mathcal{S}_b$, such that (i) all the samples in $\mathcal{S}_g$ are drawn i.i.d. from $\mathcal{N}(\mu, \sigma^2)$, where $\mu, \sigma^2 \in \mathbb{R}$; (ii) the samples in $\mathcal{S}_b$ are chosen by an adversary, and can be arbitrary; and (iii) $|\mathcal{S}_b| < \alpha|\mathcal{S}|$, where $\alpha < 1/2$. Let $\hat{\mu} = \mathtt{Median}\left(\{x_i\}, i \in [M]\right)$. Given any $\delta \in (0, 1)$, we then have that with probability at least $1 - \delta$,*

$$|\hat{\mu} - \mu| \leq C \left( \alpha + \sqrt{\frac{\log(\frac{1}{\delta})}{M}} \right) \sigma, \tag{15}$$

*where $C$ is a suitably large universal constant.*

The next key lemma - a restatement of Lemma 1 in the main body of the paper - informs us about the quality of the robust mean payoffs computed in line 8 of Algorithm 1. Before proceeding to prove this result, we define by $\mathcal{F}_\ell$ the $\sigma$-algebra generated by all the actions and rewards up to the beginning of epoch $\ell$.

**Lemma 3.** *(**Robust Confidence Intervals**) Fix any epoch $\ell$. For each active arm $b \in \mathcal{A}_\ell$, the following holds with probability at least $1 - \delta_\ell$:*

$$|\mu_b^{(\ell)} - \langle \theta_*, b \rangle| \leq \gamma_\ell, \quad where \quad \gamma_\ell = \sqrt{2}C \left( 1 + \alpha\sqrt{M} \right) \epsilon_\ell, \tag{16}$$

*where $C$ is as in Lemma 2.*

*Proof.* Fix an epoch $\ell$, an active arm $b \in \mathcal{A}_\ell$, and a good agent $i \in [M] \setminus \mathcal{B}$. We start by analyzing the statistics of the quantity $\langle \hat{\theta}_i^{(\ell)}, b \rangle$. From the definition of $\hat{\theta}_i^{(\ell)}$ and $\tilde{V}_\ell$ in Eq.(3), we have

$$
\begin{aligned}
\hat{\theta}_i^{(\ell)} &= \tilde{V}_\ell^{-1} Y_{i,\ell} \\
&= \tilde{V}_\ell^{-1} \left( \sum_{a \in \mathrm{Supp}(\pi_\ell)} m_a^{(\ell)} r_{i,a}^{(\ell)} a \right) \\
&= \tilde{V}_\ell^{-1} \left( \sum_{a \in \mathrm{Supp}(\pi_\ell)} m_a^{(\ell)} \left( \langle \theta_*, a \rangle + \bar{\eta}_{i,a}^{(\ell)} \right) a \right) \\
&= \theta_* + \tilde{V}_\ell^{-1} \left( \sum_{a \in \mathrm{Supp}(\pi_\ell)} m_a^{(\ell)} \bar{\eta}_{i,a}^{(\ell)} a \right).
\end{aligned} \tag{17}
$$

For the third equality above, we used the observation model (1), and denoted by $\bar{\eta}_{i,a}^{(\ell)}$ the average of the noise terms associated with the rewards observed by agent $i$ during phase $\ell$ for arm $a$. From (17), we then have

$$\langle \hat{\theta}_i^{(\ell)}, b \rangle = \langle \theta_*, b \rangle + \sum_{a \in \mathrm{Supp}(\pi_\ell)} m_a^{(\ell)} \bar{\eta}_{i,a}^{(\ell)} \langle \tilde{V}_\ell^{-1} a, b \rangle. \tag{18}$$

Now conditioned on $\mathcal{F}_\ell$, the only randomness in the above equation corresponds to the noise terms $\{\bar{\eta}_{i,a}^{(\ell)}\}_{a \in \mathrm{Supp}(\pi_\ell)}$. Furthermore, based on our noise model, it is clear that $\bar{\eta}_{i,a}^{(\ell)} \sim \mathcal{N}(0, 1/m_a^{(\ell)})$ for each $a \in \mathrm{Supp}(\pi_\ell)$. It then follows that

$$\mathbb{E}\left[ \langle \hat{\theta}_i^{(\ell)}, b \rangle | \mathcal{F}_\ell \right] = \langle \theta_*, b \rangle.$$

We also have

$$
\begin{aligned}
\mathbb{E}\left[\left(\langle\hat{\theta}_i^{(\ell)}-\theta_*,b\rangle\right)^2|\mathcal{F}_\ell\right] &= \sum_{a\in\mathrm{Supp}(\pi_\ell)}m_a^{(\ell)}\left(\langle\tilde{V}_\ell^{-1}a,b\rangle\right)^2 \\
&= b'\tilde{V}_\ell^{-1}\left(\sum_{a\in\mathrm{Supp}(\pi_\ell)}m_a^{(\ell)}aa'\right)\tilde{V}_\ell^{-1}b \\
&= \|b\|_{\tilde{V}_\ell^{-1}}^2,
\end{aligned}
\tag{19}
$$

where we used the fact that the noise terms are independent across arms. We conclude that conditioned on $\mathcal{F}_\ell$,

$$
\langle\hat{\theta}_i^{(\ell)},b\rangle\sim\mathcal{N}\left(\langle\theta_*,b\rangle,\|b\|_{\tilde{V}_\ell^{-1}}^2\right).
$$

In each epoch $\ell$, the server has access to a set $\mathcal{S}_b^{(\ell)}=\{\langle\hat{\theta}_i^{(\ell)},b\rangle\}_{i\in[M]}$, where the samples corresponding to agents in $[M]\setminus\mathcal{B}$ are independent and identically distributed as per the distribution above. Moreover, at most $\alpha\in[0,1/2)$ fraction of the samples in $\mathcal{S}_b^{(\ell)}$ are corrupted. Recalling that $\mu_b^{(\ell)}=\texttt{Median}\left(\{\langle\hat{\theta}_i^{(\ell)},b\rangle,i\in[M]\}\right)$, and using Lemma 2, we immediately observe that conditioned on $\mathcal{F}_\ell$, with probability at least $1-\delta_\ell$,

$$
|\mu_b^{(\ell)}-\langle\theta_*,b\rangle|\le C\left(\alpha+\sqrt{\frac{\log(\frac{1}{\delta_\ell})}{M}}\right)\|b\|_{\tilde{V}_\ell^{-1}}.
\tag{20}
$$

We now proceed to bound the term $\|b\|_{\tilde{V}_\ell^{-1}}$. To that end, let us start by noting that

$$
\begin{aligned}
\tilde{V}_\ell &= \sum_{a\in\mathrm{Supp}(\pi_\ell)}m_a^{(\ell)}aa' \\
&= \sum_{a\in\mathrm{Supp}(\pi_\ell)}\left\lceil\frac{T_a^{(\ell)}}{M}\right\rceil aa' \\
&\succcurlyeq \frac{1}{M}\sum_{a\in\mathrm{Supp}(\pi_\ell)}T_a^{(\ell)}aa' \\
&\succcurlyeq \frac{d}{M\epsilon_\ell^2}\log\left(\frac{1}{\delta_\ell}\right)\sum_{a\in\mathrm{Supp}(\pi_\ell)}\pi_\ell(a)aa' \\
&= \frac{d}{M\epsilon_\ell^2}\log\left(\frac{1}{\delta_\ell}\right)\sum_{a\in\mathcal{A}_\ell}\pi_\ell(a)aa' \\
&= \frac{d}{M\epsilon_\ell^2}\log\left(\frac{1}{\delta_\ell}\right)V_\ell(\pi_\ell).
\end{aligned}
\tag{21}
$$

Thus, we have

$$
\tilde{V}_\ell^{-1}\preccurlyeq\frac{M\epsilon_\ell^2}{d\log\left(\frac{1}{\delta_\ell}\right)}V_\ell^{-1}(\pi_\ell).
$$

Using the above bound, we proceed as follows.

$$\|b\|_{\tilde{V}_\ell^{-1}} = \sqrt{b'\tilde{V}_\ell^{-1}b}$$

$$\leq \epsilon_\ell \sqrt{\frac{M}{d\log\left(\frac{1}{\delta_\ell}\right)}} \sqrt{b'V_\ell^{-1}(\pi_\ell)b}$$

$$\leq \epsilon_\ell \sqrt{\frac{M}{d\log\left(\frac{1}{\delta_\ell}\right)}} \sqrt{\max_{a\in\mathcal{A}_\ell}\|a\|^2_{V_\ell^{-1}(\pi_\ell)}} \tag{22}$$

$$\overset{(a)}{=} \epsilon_\ell \sqrt{\frac{M}{d\log\left(\frac{1}{\delta_\ell}\right)}} \sqrt{g_\ell(\pi_\ell)}$$

$$\overset{(b)}{\leq} \epsilon_\ell \sqrt{\frac{2M}{\log\left(\frac{1}{\delta_\ell}\right)}}.$$

In the above steps, we used the definition of $g_\ell(\pi_\ell)$ for (a); for (b), we used the fact that based on the approximate G-optimal design problem solved by the server in line 1 of RCLB, $g_\ell(\pi_\ell) \leq 2d$. Plugging the bound from (22) into (20), and using the fact that $\log\left(\frac{1}{\delta_\ell}\right) \geq 1$, we have that

$$\mathbb{P}\left(|\mu_b^{(\ell)} - \langle\theta_*, b\rangle| \geq \gamma_\ell|\mathcal{F}_\ell\right) \leq \delta_\ell. \tag{23}$$

Consider the following event $\mathcal{E}_\ell \triangleq \{|\mu_b^{(\ell)} - \langle\theta_*, b\rangle| \geq \gamma_\ell\}$. Now observe that

$$\begin{aligned}
\mathbb{P}(\mathcal{E}_\ell) &= \mathbb{E}[\mathbf{1}_{\mathcal{E}_\ell}] \\
&= \mathbb{E}\left[\mathbb{E}[\mathbf{1}_{\mathcal{E}_\ell}|\mathcal{F}_\ell]\right] \\
&= \mathbb{E}\left[\mathbb{P}(\mathcal{E}_\ell|\mathcal{F}_\ell)\right] \\
&\leq \delta_\ell,
\end{aligned} \tag{24}$$

where we used $\mathbf{1}_{\mathcal{E}_\ell}$ to denote an indicator random variable associated with the event $\mathcal{E}_\ell$; also, for the last line, we used (23). $\qquad\square$

In the following two results, we use the robust confidence intervals from Lemma 3 to construct clean events that hold with high probability on which (i) the optimal arm $a_*$ is never eliminated (Lemma 4); and (ii) any arm retained in epoch $\ell$ contributes at most $O(\gamma_\ell)$ regret in each time-step within epoch $\ell$ (Lemma 5). To proceed, for each $a \in \mathcal{A}$, define the arm-gap $\Delta_a = \langle\theta_*, a_* - a\rangle$.

**Lemma 4.** *Define the event* $\mathcal{G}_1 \triangleq \{a_* \in \mathcal{A}_\ell, \forall \ell \in [L]\}$, *where $L$ is the total number of epochs. It then holds that* $\mathbb{P}(\mathcal{G}_1) \geq 1 - 4\delta$.

*Proof.* Based on the arm-elimination criterion in line 9 of Algorithm 1, it follows that $\{a_* \in \mathcal{A}_\ell, a_* \notin \mathcal{A}_{\ell+1}\} \implies \{\exists b \in \mathcal{A}_\ell : \mu_b^{(\ell)} - \mu_{a_*}^{(\ell)} > 2\gamma_\ell\}$. Now for any fixed $b \in \mathcal{A}_\ell$, we have

$$\begin{aligned}
&\mu_b^{(\ell)} - \mu_{a_*}^{(\ell)} > 2\gamma_\ell \\
&\implies \left(\mu_b^{(\ell)} - \langle\theta_*, b\rangle\right) + \left(\langle\theta_*, a_*\rangle - \mu_{a_*}^{(\ell)}\right) > 2\gamma_\ell + \Delta_b \tag{25} \\
&\implies \left(\mu_b^{(\ell)} - \langle\theta_*, b\rangle\right) + \left(\langle\theta_*, a_*\rangle - \mu_{a_*}^{(\ell)}\right) > 2\gamma_\ell,
\end{aligned}$$

where for the second step, we used the fact that $\Delta_b \geq 0$. Thus, the event $\{\mu_b^{(\ell)} - \mu_{a_*}^{(\ell)} > 2\gamma_\ell\}$ implies the occurrence of either $\{\mu_b^{(\ell)} - \langle\theta_*, b\rangle > \gamma_\ell\}$ or $\{\langle\theta_*, a_*\rangle - \mu_{a_*}^{(\ell)} > \gamma_\ell\}$. From Lemma 3, we further know that the probability of each of these latter events is at most $\delta_\ell$. Putting these pieces together,

and using an union bound, we have

$$
\begin{aligned}
\mathbb{P}(\mathcal{G}_1^c) &\leq \sum_{\ell \in [L]} \mathbb{P}(a^* \in \mathcal{A}_\ell, a^* \notin \mathcal{A}_{\ell+1}) \\
&\leq \sum_{\ell \in [L]} \mathbb{P}(\exists b \in \mathcal{A}_\ell : \mu_b^{(\ell)} - \mu_{a_*}^{(\ell)} > 2\gamma_\ell) \\
&\leq 2K \sum_{\ell \in [L]} \delta_\ell \\
&= 2K \sum_{\ell \in [L]} \frac{\bar{\delta}}{K\ell^2} \\
&\leq 2 \sum_{\ell=1}^{\infty} \frac{\bar{\delta}}{\ell^2} \\
&\leq 2\bar{\delta} \int_{x=1}^{\infty} \frac{1}{x^2} \, dx \leq 4\bar{\delta}.
\end{aligned}
\tag{26}
$$

This completes the proof. $\qquad\square$

In our next result, we work towards bounding the regret incurred from playing each active arm in a given epoch.

**Lemma 5.** *Consider any arm $a \in \mathcal{A} \setminus \{a_*\}$. Let $\ell_a$ be defined as $\ell_a \triangleq \min\{\ell : \gamma_\ell < \frac{\Delta_a}{4}\}$. It then holds that $\mathbb{P}(a \in \mathcal{A}_{\ell_a+1}) \leq 6\bar{\delta}$.*

*Proof.* Let us start by observing that

$$
\begin{aligned}
\mathbb{P}(a \in \mathcal{A}_{\ell_a+1}) &= \mathbb{P}(a \in \mathcal{A}_{\ell_a+1}, a^* \in \mathcal{A}_{\ell_a}) + \mathbb{P}(a \in \mathcal{A}_{\ell_a+1}, a^* \notin \mathcal{A}_{\ell_a}) \\
&\leq \mathbb{P}(a \in \mathcal{A}_{\ell_a+1}, a^* \in \mathcal{A}_{\ell_a}) + \mathbb{P}(a^* \notin \mathcal{A}_{\ell_a}) \\
&\leq \mathbb{P}(a \in \mathcal{A}_{\ell_a}, a \in \mathcal{A}_{\ell_a+1}, a^* \in \mathcal{A}_{\ell_a}) + 4\bar{\delta},
\end{aligned}
\tag{27}
$$

where for the last step, we used the fact that $\{a \in \mathcal{A}_{\ell_a+1}\} \implies \{a \in \mathcal{A}_{\ell_a}\}$, and Lemma 4. Now, to bound $\mathbb{P}(a \in \mathcal{A}_{\ell_a}, a \in \mathcal{A}_{\ell_a+1}, a^* \in \mathcal{A}_{\ell_a})$, we note based on line 9 of RCLB that

$$
\begin{aligned}
\mathbb{P}(a \in \mathcal{A}_{\ell_a}, a \in \mathcal{A}_{\ell_a+1}, a^* \in \mathcal{A}_{\ell_a}) &\leq \mathbb{P}\left( \max_{b \in \mathcal{A}_{\ell_a}} \mu_b^{(\ell_a)} - \mu_a^{(\ell_a)} \leq 2\gamma_{\ell_a} \right) \\
&\leq \mathbb{P}\left( \mu_{a_*}^{(\ell_a)} - \mu_a^{(\ell_a)} \leq 2\gamma_{\ell_a} \right) \\
&\leq \mathbb{P}\left( \left( \mu_a^{(\ell_a)} - \langle \theta_*, a \rangle \right) + \left( \langle \theta_*, a_* \rangle - \mu_{a_*}^{(\ell_a)} \right) > \Delta_a - 2\gamma_{\ell_a} \right) \\
&\leq \mathbb{P}\left( \mu_a^{(\ell_a)} - \langle \theta_*, a \rangle > \gamma_{\ell_a} \right) + \mathbb{P}\left( \langle \theta_*, a_* \rangle - \mu_{a_*}^{(\ell_a)} > \gamma_{\ell_a} \right) \\
&\leq 2\delta_{\ell_a},
\end{aligned}
\tag{28}
$$

where we used $\Delta_a > 4\gamma_{\ell_a}$ for the second last step, and Lemma 3 for the last step. Noting that $\delta_{\ell_a} \leq \bar{\delta}$, and combining the bounds in equations (27) and (28) leads to the claim of the lemma. $\quad\square$

We are now in place to prove Theorem 1.

*Proof.* (**Proof of Theorem 1**) We start by constructing an appropriate clean event $\mathcal{E}$ for our subsequent analysis. Accordingly, let us define:

$$
\mathcal{E} = \{a_* \in \mathcal{A}_\ell, \forall \ell \in [L]\} \bigcap \{\cap_{a \in \mathcal{A} \setminus \{a_*\}} \{a \notin \mathcal{A}_{\ell_a+1}\}\}.
\tag{29}
$$

Based on Lemmas 4 and 5, we then have

$$
\begin{aligned}
\mathbb{P}(\mathcal{E}^c) &\leq 4\bar{\delta} + \sum_{a \in \mathcal{A} \setminus \{a_*\}} \mathbb{P}(a \in \mathcal{A}_{\ell_a+1}) \\
&\leq 4\bar{\delta} + 6\bar{\delta}K \\
&\leq 10\bar{\delta}K = \delta,
\end{aligned}
\tag{30}
$$

as per the choice of $\bar{\delta}$ in Theorem 1. Thus, $\mathbb{P}(\mathcal{E}) \geq 1 - \delta$. Throughout the rest of the proof, we will condition on the clean event $\mathcal{E}$. Based on the definition of the event $\mathcal{E}$, it is easy to see that for any epoch $\ell \in [L]$, $a \in \mathcal{A}_\ell \implies \Delta_a \leq 8\gamma_\ell$. Using this key fact, we now proceed to bound the regret of any good agent $i \in [M] \setminus \mathcal{B}$.

$$
\sum_{t=1}^{T} \langle \theta_*, a_* - a_{i,t} \rangle = \sum_{\ell=1}^{L} \sum_{a \in \mathrm{Supp}(\pi_\ell)} m_a^{(\ell)} \langle \theta_*, a_* - a \rangle
$$

$$
= \sum_{\ell=1}^{L} \sum_{a \in \mathrm{Supp}(\pi_\ell)} \left\lceil \frac{T_a^{(\ell)}}{M} \right\rceil \Delta_a \tag{31}
$$

$$
\leq \underbrace{\sum_{\ell=1}^{L} \sum_{a \in \mathrm{Supp}(\pi_\ell)} \frac{T_a^{(\ell)}}{M} \Delta_a}_{T_1} + \underbrace{\sum_{\ell=1}^{L} \sum_{a \in \mathrm{Supp}(\pi_\ell)} \Delta_a}_{T_2}.
$$

We now bound $T_1$ and $T_2$ separately. For bounding $T_1$, we have:

$$
\begin{aligned}
T_1 &= \sum_{\ell=1}^{L} \sum_{a \in \mathrm{Supp}(\pi_\ell)} \frac{T_a^{(\ell)}}{M} \Delta_a \\
&= \frac{d}{M} \sum_{\ell=1}^{L} \frac{1}{\epsilon_\ell^2} \log\left(\frac{1}{\delta_\ell}\right) \sum_{a \in \mathrm{Supp}(\pi_\ell)} \pi_\ell(a) \Delta_a \\
&\leq \frac{8d}{M} \sum_{\ell=1}^{L} \frac{1}{\epsilon_\ell^2} \log\left(\frac{1}{\delta_\ell}\right) \gamma_\ell \\
&= \frac{8\sqrt{2}C(1 + \alpha\sqrt{M})d}{M} \sum_{\ell=1}^{L} \frac{1}{\epsilon_\ell} \log\left(\frac{10K^2\ell^2}{\delta}\right) \\
&\leq \frac{8\sqrt{2}C(1 + \alpha\sqrt{M})d}{M} \log\left(\frac{10K^2 L^2}{\delta}\right) \sum_{\ell=1}^{L} 2^\ell \\
&= O\left(\frac{(1 + \alpha\sqrt{M})d}{M} \log\left(\frac{10K^2 L^2}{\delta}\right) 2^L\right).
\end{aligned} \tag{32}
$$

In the third step above, we used $\sum_{a \in \mathrm{Supp}(\pi_\ell)} \pi_\ell(a) = 1$. We now need an upper-bound on the term $2^L$. To that end, notice that the length of the horizon $T$ is bounded below by the length of the last epoch, i.e., the $L$-th epoch. Moreover, the duration of the $L$-th epoch corresponds to the number of arm-pulls made by any single good agent during the $L$-th epoch. We thus have:

$$
\begin{aligned}
T &\geq \sum_{a \in \mathrm{Supp}(\pi_L)} \frac{T_a^{(L)}}{M} \\
&\geq \frac{4^L d}{M} \sum_{a \in \mathrm{Supp}(\pi_L)} \pi_L(a) \log\left(\frac{1}{\delta_L}\right) \\
&= \frac{4^L d}{M} \log\left(\frac{10K^2 L^2}{\delta}\right) \sum_{a \in \mathrm{Supp}(\pi_L)} \pi_L(a) \\
&= \frac{4^L d}{M} \log\left(\frac{10K^2 L^2}{\delta}\right).
\end{aligned} \tag{33}
$$

Thus, $2^L \leq \sqrt{MT/(d \log(10K^2 L^2/\delta))}$. Plugging this bound in (32), we obtain

$$
T_1 = O\left(\left(\alpha + \frac{1}{\sqrt{M}}\right) \sqrt{\log\left(\frac{KT}{\delta}\right) dT}\right) = \tilde{O}\left(\left(\alpha + \frac{1}{\sqrt{M}}\right) \sqrt{dT}\right).
$$

As for the term $T_2$, we have

$$
\begin{aligned}
T_2 &= \sum_{\ell=1}^{L} \sum_{a \in \mathrm{Supp}(\pi_\ell)} \Delta_a \\
&\leq 8 \sum_{\ell=1}^{L} \gamma_\ell |\mathrm{Supp}(\pi_\ell)| \\
&\overset{(a)}{\leq} 384\sqrt{2} C d \log\log d \left(1 + \alpha\sqrt{M}\right) \sum_{\ell=1}^{L} 2^{-\ell} \\
&= O\left(d \log\log d \left(1 + \alpha\sqrt{M}\right)\right) \\
&\overset{(b)}{=} \tilde{O}\left(\left(\alpha + \frac{1}{\sqrt{M}}\right)\sqrt{dT}\right).
\end{aligned}
\tag{34}
$$

In the above steps, for (a), recall from line 1 of `RCLB` that $|\mathrm{Supp}(\pi_\ell)| \leq 48d \log\log d$ based on the approximate G-optimal design computation. For (b), we used the fact that by assumption, $T \geq Md$. Combining the bounds on $T_1$ and $T_2$, and recalling that $\mathbb{P}(\mathcal{E}) \geq 1 - \delta$, we have that with probability at least $1 - \delta$,

$$
\sum_{t=1}^{T} \langle \theta_*, a_* - a_{i,t} \rangle = O\left(\left(\alpha + \frac{1}{\sqrt{M}}\right)\sqrt{\log\left(\frac{KT}{\delta}\right)dT}\right) = \tilde{O}\left(\left(\alpha + \frac{1}{\sqrt{M}}\right)\sqrt{dT}\right). \tag{35}
$$

This concludes the proof. $\qquad\square$

The following is an immediate corollary of Theorem 1 on the group regret $R_T$.

**Corollary 1.** *(**Bound on Group Regret**) Under the conditions of Theorem 1, we have:*

$$
R_T = \tilde{O}\left(\left(\alpha M + \sqrt{M}\right)\sqrt{dT}\right). \tag{36}
$$

*Proof.* (**Proof of Corollary 1**) Recall from the proof of Theorem 1 that there exists a clean event $\mathcal{E}$ of measure at least $1 - \delta$ on which the regret of every good agent is bounded above as per Eq. (35). Let $\tilde{C}$ be an upper bound on the maximum instantaneous regret, i.e.,

$$
\max_{a \in \mathcal{A}} \langle \theta_*, a_* - a \rangle \leq \tilde{C}.
$$

Now set $\delta = \frac{1}{MT}$ and observe that:

$$
\begin{aligned}
R_T &= \mathbb{E}\left[\sum_{i \in [M]\backslash\mathcal{B}} \sum_{t=1}^{T} \langle \theta_*, a_* - a_{i,t} \rangle\right] \\
&= \mathbb{E}\left[\sum_{i \in [M]\backslash\mathcal{B}} \sum_{t=1}^{T} \langle \theta_*, a_* - a_{i,t} \rangle \Big| \mathcal{E}\right] \mathbb{P}(\mathcal{E}) + \mathbb{E}\left[\sum_{i \in [M]\backslash\mathcal{B}} \sum_{t=1}^{T} \langle \theta_*, a_* - a_{i,t} \rangle \Big| \mathcal{E}^c\right] \mathbb{P}(\mathcal{E}^c) \\
&\leq C_1 \left(\alpha M + \sqrt{M}\right)\sqrt{\log\left(KMT\right)dT} + \tilde{C}MT \times \frac{1}{MT} \\
&= O\left(\left(\alpha M + \sqrt{M}\right)\sqrt{\log\left(KMT\right)dT}\right) \\
&= \tilde{O}\left(\left(\alpha M + \sqrt{M}\right)\sqrt{dT}\right).
\end{aligned}
\tag{37}
$$

In the above steps, $C_1$ is a suitably large universal constant. $\qquad\square$

# D  Lower Bound Analysis: Proof of Theorem 2

In this section, we will prove Theorem 2. Before diving into the technical details, we remind the reader that we consider a slightly different adversarial model from the one considered throughout the paper. In this modified model, with probability $\alpha$, each agent is adversarial independently of the other agents. We will consider a class of policies $\Pi$ where at each time-step, the same action (decided by the server) is played by every agent. Thus, the regret incurred by any individual agent is the same as the regret incurred by the server. Finally, to prove the lower bound, we will focus on a class of 2-armed bandits where the reward distribution of each arm is Gaussian with unit variance; such a class of bandits is succinctly denoted by $\mathcal{E}_{\mathcal{N}}^{(2)}(1)$.

We will have occasion to use the following result [57, Theorem 14.2].

**Lemma 6.** *(Bretagnolle-Huber Inequality) Let $P$ and $Q$ be two probability measures on the same measurable space $(\Omega, \mathcal{F})$, and let $A \in \mathcal{F}$ be any arbitrary event. Then,*

$$P(A) + Q(A^c) \geq \frac{1}{2} \exp\left(-KL(P, Q)\right),$$

*where $A^c$ is the complement of the event $A$, and $KL(P, Q)$ is the Kullback-Leibler distance between $P$ and $Q$.*

Our proof comprises of two main steps. First, we construct two *distinct* bandit instances within the class $\mathcal{E}_{\mathcal{N}}^{(2)}(1)$ such that the two instances - although different - appear *identical* to the server. Second, we devise an attack strategy and argue that regardless of the policy played by the server, it will end up suffering a regret of $\Omega(\alpha\sqrt{T})$ upon interacting with at least one of the two instances; here, $T$ is the horizon for our problem. We now elaborate on these two steps.

• **Step 1. Construction of the two bandit instances.** We first take a detour and describe an idea that is typically used to prove lower bounds for the robust mean estimation literature [18, 19]. It will soon be apparent how such an idea can be exploited to construct the two bandit instances for our problem. We show that there are two univariate Gaussian distributions $P_1 = \mathcal{N}(\mu_1, 1)$, $P_2 = \mathcal{N}(\mu_2, 1)$, and two $T$-dimensional distributions $Q_1, Q_2$, such that $\|\mu_1 - \mu_2\|_2 = \Omega(\alpha/\sqrt{T})$, and:

$$(1-\alpha)P_1^T + \alpha Q_1 = (1-\alpha)P_2^T + \alpha Q_2, \tag{38}$$

where $P_1^T$ (resp., $P_2^T$) is the joint distribution of $T$ i.i.d. samples drawn from $P_1$ (resp., $P_2$). Clearly, $P_1^T$ (resp., $P_2^T$) is equivalent to a $T$-dimensional Gaussian distribution $\mathcal{N}(\boldsymbol{\mu}_1, I_T)$ (resp., $\mathcal{N}(\boldsymbol{\mu}_2, I_T)$), where $\boldsymbol{\mu}_1$ (resp., $\boldsymbol{\mu}_2$) is a $T$-dimensional vector with each entry equal to $\mu_1$ (resp., $\mu_2$). Let $\phi_1$ be the p.d.f. of $P_1^T$ and $\phi_2$ be the p.d.f. of $P_2^T$. Next, let $\mu_1$ and $\mu_2$ be chosen such that the total variation distance $\delta(P_1^T, P_2^T)$ between $P_1^T$ and $P_2^T$ is

$$\frac{1}{2} \int \|\phi_1 - \phi_2\|_1 dx = \frac{\alpha}{1-\alpha}.$$

Let $Q_1$ be the distribution with p.d.f. $\frac{1-\alpha}{\alpha}(\phi_2 - \phi_1)\mathbf{1}_{\phi_2 \geq \phi_1}$, and $Q_2$ be the distribution with p.d.f. $\frac{1-\alpha}{\alpha}(\phi_1 - \phi_2)\mathbf{1}_{\phi_1 \geq \phi_2}$. With such a construction of $Q_1$ and $Q_2$, one can verify that the equality in Eq. (38) is satisfied; see, for instance, the arguments in Appendix E of [18]. Now from Pinsker's inequality, we know that:

$$\sqrt{\frac{1}{2}KL(P_1^T, P_2^T)} \geq \delta(P_1^T, P_2^T) = \frac{\alpha}{1-\alpha},$$

where we used $KL(P_1^T, P_2^T)$ to denote the Kullback-Leibler distance between $P_1^T$ and $P_2^T$. We conclude that:

$$\frac{1}{2}\|\boldsymbol{\mu}_1 - \boldsymbol{\mu}_2\|_2^2 = KL(P_1^T, P_2^T) \geq 2\left(\frac{\alpha}{1-\alpha}\right)^2.$$

This, in turn, implies

$$\sqrt{T}\left(|\mu_1 - \mu_2|\right) = \|\boldsymbol{\mu}_1 - \boldsymbol{\mu}_2\|_2 \geq \frac{2\alpha}{1-\alpha} \geq 2\alpha.$$

We have thus shown that there exist $\mu_1, \mu_2 \in \mathbb{R}$, satisfying $|\mu_1 - \mu_2| \geq 2\alpha/\sqrt{T}$, such that with $P_1 = \mathcal{N}(\mu_1, 1)$, $P_2 = \mathcal{N}(\mu_2, 1)$, one can satisfy Eq. (38) with appropriately chosen $T$-dimensional distributions $Q_1$ and $Q_2$. With these ideas in place, we now return to our bandit problem.

Without loss of generality, suppose $\mu_2 > \mu_1$, where $\mu_2$ and $\mu_1$ are as in the construction described above. Let us construct two bandit instances $\nu$ and $\nu'$, each involving two arms, i.e., $\mathcal{A} = \{a_1, a_2\}$. Let $r_{a_k}^{(\nu)}$ and $r_{a_k}^{(\nu')}$ denote the reward distribution of arm $a_k, k \in \{1, 2\}$, in instance $\nu$ and instance $\nu'$, respectively. These reward distributions are chosen as follows.

$$\text{Instance } \nu : r_{a_1}^{(\nu)} \sim \mathcal{N}\left(\frac{\mu_1 + \mu_2}{2}, 1\right); \quad r_{a_2}^{(\nu)} \sim \mathcal{N}(\mu_1, 1)$$

$$\text{Instance } \nu' : r_{a_1}^{(\nu')} \sim \mathcal{N}\left(\frac{\mu_1 + \mu_2}{2}, 1\right); \quad r_{a_2}^{(\nu')} \sim \mathcal{N}(\mu_2, 1). \tag{39}$$

Thus, the distribution of the first arm is the same in both instances. However, as $\mu_2 > \mu_1$, the first arm is the best arm in the first instance while the opposite is true for the second instance. The attack strategy for the adversarial agents will be dictated by the distributions $Q_1$ and $Q_2$ in a manner to be described shortly.

• **Step 2. The attack strategy and regret analysis.** Inspired by the argument in the proof of [17, Theorem 5], we consider a full information setting where the server has access to $t$ reward samples from *each* arm from *each* agent at time-step $t$. Since this full information setting is simpler than the bandit setting, a lower bound for the former implies one for the latter.

Here is the attack strategy. Suppose an agent is adversarial (which happens with probability $\alpha$). In either instance, for arm 1, it reports samples from the true distribution for arm 1 corresponding to that instance. In other words, the reward samples for arm 1 are not corrupted by the agent. As for arm 2, in instance $\nu$ (resp., $\nu'$), the first $t$ reward samples (where $t \in [T]$) corresponding to $a_2$ are generated from $Q_1^t$ (resp., $Q_2^t$) by the adversarial agent. Here, $Q_1^t$ (resp., $Q_2^t$) is the marginal of the $T$-dimensional distribution $Q_1$ (resp., $Q_2$) corresponding to the first $t$ components. To sum up, in instance $\nu$, the joint distribution $D_{a_1}^{(\nu)}$ of rewards for $a_1$ over the horizon $T$, as seen by the server from any given agent, is the $T$-dimensional Gaussian distribution $\mathcal{N}(\bar{\boldsymbol{\mu}}, I_T)$, where $\bar{\boldsymbol{\mu}}$ is a $T$-dimensional vector with each component equal to $\frac{\mu_1 + \mu_2}{2}$. Based on our discussion above, $D_{a_1}^{(\nu)} = D_{a_1}^{(\nu')}$. Let $D_{a_2}^{(\nu)}$ and $D_{a_2}^{(\nu')}$ have analogous meanings for arm $a_2$. Then, we have:

$$D_{a_2}^{(\nu)} = (1-\alpha)P_1^T + \alpha Q_1; \quad D_{a_2}^{(\nu')} = (1-\alpha)P_2^T + \alpha Q_2. \tag{40}$$

In light of Eq. (38), however, we have $D_{a_2}^{(\nu)} = D_{a_2}^{(\nu')}$. Essentially, what we have established is the following: *the joint distribution of rewards for each arm over the horizon $T$, as seen by the server from each agent, is identical for both instances.*

In what follows, given any two distributions $D_1$ and $D_2$, let $D_1 \otimes D_2$ represent their product distribution. Since the rewards across arms are independent, the joint distribution of rewards for both arms is given by $D^{(\nu)} \triangleq D_{a_1}^{(\nu)} \otimes D_{a_2}^{(\nu)}$ in instance $\nu$, and $D^{(\nu')} \triangleq D_{a_1}^{(\nu')} \otimes D_{a_2}^{(\nu')}$ in instance $\nu'$. Since rewards across agents are independent, the joint distributions of rewards from *all* agents, as seen by the server in each of the two instances, are given by:

$$D_M^{(\nu)} \triangleq \underbrace{D^{(\nu)} \otimes D^{(\nu)} \otimes \cdots \otimes D^{(\nu)}}_{M\text{-fold product distribution}}; \quad D_M^{(\nu')} \triangleq \underbrace{D^{(\nu')} \otimes D^{(\nu')} \otimes \cdots \otimes D^{(\nu')}}_{M\text{-fold product distribution}}.$$

Let $\mathbb{E}_\nu[\cdot]$ (resp., $\mathbb{E}_{\nu'}[\cdot]$) represent the expectation operation w.r.t. the measure $D_M^{(\nu)}$ (resp., $D_M^{(\nu')}$). Let us use $\mathbb{P}_\nu$ as a shorthand for $D_M^{(\nu)}$, and $\mathbb{P}_{\nu'}$ as a shorthand for $D_M^{(\nu')}$. Furthermore, let $n_k(T)$ be the random variable representing the total number of times arm $a_k, k \in \{1, 2\}$, is chosen by the server over the horizon $T$. Finally, recall that $R_T^{(s)}(\nu)$ (resp., $R_T^{(s)}(\nu')$) is the regret incurred by the server upon interaction with instance $\nu$ (resp., instance $\nu'$).

In instance $\nu$, each time arm 2 is chosen by the server, it incurs an instantaneous regret of $(\mu_2 - \mu_1)/2$. We thus have:

$$R_T^{(s)}(\nu) = \left(\frac{\mu_2 - \mu_1}{2}\right)(T - \mathbb{E}_\nu[n_1(T)]) \geq \frac{(\mu_2 - \mu_1)T}{4}\mathbb{P}_\nu\left(n_1(T) \leq \frac{T}{2}\right). \tag{41}$$

To see why the latter inequality is true, observe:

$$
\begin{aligned}
\mathbb{E}_\nu[n_1(T)] &= \mathbb{E}_\nu\left[n_1(T)|n_1(T) \le \frac{T}{2}\right]\mathbb{P}_\nu\left(n_1(T) \le \frac{T}{2}\right) + \mathbb{E}_\nu\left[n_1(T)|n_1(T) > \frac{T}{2}\right]\mathbb{P}_\nu\left(n_1(T) > \frac{T}{2}\right) \\
&\le \frac{T}{2}\mathbb{P}_\nu\left(n_1(T) \le \frac{T}{2}\right) + T\left(1 - \mathbb{P}_\nu\left(n_1(T) \le \frac{T}{2}\right)\right) \\
&= T - \frac{T}{2}\mathbb{P}_\nu\left(n_1(T) \le \frac{T}{2}\right).
\end{aligned}
\tag{42}
$$

In instance $\nu'$, each time arm 1 is chosen by the server, it incurs an instantaneous regret of $(\mu_2 - \mu_1)/2$. We thus have:

$$
R_T^{(s)}(\nu') = \left(\frac{\mu_2 - \mu_1}{2}\right)\mathbb{E}_{\nu'}[n_1(T)] \ge \frac{(\mu_2 - \mu_1)T}{4}\mathbb{P}_{\nu'}\left(n_1(T) > \frac{T}{2}\right).
\tag{43}
$$

Combining Eq. (41) and Eq. (43) yields:

$$
\begin{aligned}
\max\{R_T^{(s)}(\nu), R_T^{(s)}(\nu')\} &\ge \frac{1}{2}\left(R_T^{(s)}(\nu) + R_T^{(s)}(\nu')\right) \\
&\ge \frac{(\mu_2 - \mu_1)T}{8}\left(\mathbb{P}_\nu\left(n_1(T) \le \frac{T}{2}\right) + \mathbb{P}_{\nu'}\left(n_1(T) > \frac{T}{2}\right)\right) \\
&\overset{(a)}{\ge} \frac{(\mu_2 - \mu_1)T}{16}\exp\left(-\mathrm{KL}\left(\mathbb{P}_\nu, \mathbb{P}_{\nu'}\right)\right) \\
&\overset{(b)}{\ge} \frac{\alpha\sqrt{T}}{8}\exp\left(-\mathrm{KL}\left(\mathbb{P}_\nu, \mathbb{P}_{\nu'}\right)\right) \\
&\overset{(c)}{=} \frac{\alpha\sqrt{T}}{8}.
\end{aligned}
\tag{44}
$$

In the above steps, we used the Bretagnolle-Huber inequality (namely, Lemma 6) for (a); for (b), we used the fact that $\mu_2$ and $\mu_1$ were chosen in Step 1 to satisfy $\mu_2 - \mu_1 \ge 2\alpha/\sqrt{T}$; and for (c), we used $\mathrm{KL}\left(\mathbb{P}_\nu, \mathbb{P}_{\nu'}\right) = 0$. To see why $\mathrm{KL}\left(\mathbb{P}_\nu, \mathbb{P}_{\nu'}\right) = 0$, we use the chain-rule for relative entropies to obtain:

$$
\begin{aligned}
\mathrm{KL}\left(\mathbb{P}_\nu, \mathbb{P}_{\nu'}\right) &= \mathrm{KL}\left(D_M^{(\nu)}, D_M^{(\nu')}\right) \\
&= M\left(\mathrm{KL}\left(D^{(\nu)}, D^{(\nu')}\right)\right) \\
&= M\left(\mathrm{KL}\left(D_{a_1}^{(\nu)}, D_{a_1}^{(\nu')}\right) + \mathrm{KL}\left(D_{a_2}^{(\nu)}, D_{a_2}^{(\nu')}\right)\right) \\
&= 0,
\end{aligned}
\tag{45}
$$

where the last step is a consequence of the fact that $D_{a_1}^{(\nu)} = D_{a_1}^{(\nu')}$, and $D_{a_2}^{(\nu)} = D_{a_2}^{(\nu')}$. The claim of Theorem 2 follows from noting that the resulting lower bound in Eq. (44) holds regardless of the number of agents $M$.

∎

**Remark 3.** *The proof of Theorem 2 above reveals a constructive attack strategy for the adversarial agents to follow. As future work, it would be interesting to see if similar ideas can be extended to more general reinforcement learning problems.*

**Algorithm 4** Robust Collaborative Phased Elimination for Generalized Linear Bandits (RC-GLM)

---

**Input:** Action set $\mathcal{A} = \{a_1, \ldots, a_K\}$, confidence parameter $\delta$, and corruption fraction $\alpha$.

**Initialize:** $\ell = 1$ and $\mathcal{A}_1 = \mathcal{A}$.

1: Let $V_\ell(\pi) \triangleq \sum_{a \in \mathcal{A}_\ell} \pi(a) a a'$ and $g_\ell(\pi) \triangleq \max_{a \in \mathcal{A}_\ell} \|a\|^2_{V_\ell(\pi)^{-1}}$. Server solves an approximate G-optimal design problem to compute a distribution $\pi_\ell$ over $\mathcal{A}_\ell$ such that $g_\ell(\pi_\ell) \leq 2d$ and $|\text{Supp}(\pi_\ell)| \leq 48d \log \log d$.

2: For each $a \in \mathcal{A}_\ell$, server computes $m_a^{(\ell)}$ via Eq. (5), and broadcasts $\{m_a^{(\ell)}\}_{a \in \mathcal{A}_\ell}$ to all agents.

3: **for** $i \in [M] \setminus \mathcal{B}$ **do**

4:     For each arm $a \in \mathcal{A}_\ell$, pull it $m_a^{(\ell)}$ times. Let $r_{i,a}^{(\ell)}$ be the average of the rewards observed by agent $i$ for arm $a$ during phase $\ell$.

5:     Compute $\tilde{V}_\ell$ and $Y_{i,\ell}$ as follows.

$$\tilde{V}_\ell = \sum_{a \in \text{Supp}(\pi_\ell)} m_a^{(\ell)} a a' \; ; \; Y_{i,\ell} = \sum_{a \in \text{Supp}(\pi_\ell)} m_a^{(\ell)} r_{i,a}^{(\ell)} a.$$

6:     Transmit $Y_{i,\ell}$ to server. Adversarial agents can transmit arbitrary vectors at this stage.

7: **end for**

8: Server computes a statistic $X_\ell$ as follows:

$$X_\ell = \text{ITW}(\{\tilde{V}_\ell^{-1/2} Y_{i,\ell}, i \in [M]\}),$$

where $\text{ITW}(\cdot)$ is the output of the Iteratively Reweighted Mean Estimator from [22].

9: Server computes a robust estimate $\hat{\theta}^{(\ell)}$ of $\theta_*$ by solving:

$$h_\ell(\hat{\theta}^{(\ell)}) = \tilde{V}_\ell^{1/2} X_\ell.$$

10: Define robust confidence threshold $\bar{\gamma}_\ell = 4C_1(k_2/k_1)\left(\sqrt{d} + \alpha\sqrt{M \log(1/\alpha)}\right)\epsilon_\ell$, where $\epsilon_\ell = 2^{-\ell}$, and $C_1$ is as in Lemma 8. Server performs phased elimination with the estimate $\hat{\theta}^{(\ell)}$ and confidence threshold $\bar{\gamma}_\ell$ to update active arm set:

$$\mathcal{A}_{\ell+1} = \{a \in \mathcal{A}_\ell : \max_{b \in \mathcal{A}_\ell} \mu(\langle \hat{\theta}^{(\ell)}, b \rangle) - \mu(\langle \hat{\theta}^{(\ell)}, a \rangle) \leq 2\bar{\gamma}_\ell\}. \tag{46}$$

11: $\ell = \ell + 1$ and **Goto** line 1.

---

## E   Algorithms and Analysis for the Generalized Linear Bandit Model

In this section, we first provide a detailed outline of the RC-GLM algorithm introduced in Section 5; see Algorithm 4. We then proceed to analyze RC-GLM, and provide a proof for Theorem 3. Finally, since RC-GLM uses the iteratively reweighted mean estimator from [22] as a sub-routine, we also provide a description of this estimator to keep the paper self-contained; this description, however, is deferred to the end of the section. We start by reminding the reader that the non-linear observation model of interest to us in this section is as follows:

$$y_{i,t} = \mu\left(\langle \theta_*, a_{i,t} \rangle\right) + \eta_{i,t}, \tag{47}$$

where $\mu : \mathbb{R} \to \mathbb{R}$ is the link function. We also recall the definition of $h_\ell(\theta)$:

$$h_\ell(\theta) \triangleq \sum_{a \in \text{Supp}(\pi_\ell)} m_a^{(\ell)} \mu(\langle \theta, a \rangle) a, \forall \theta \in \Theta.$$

From comparing RC-GLM (Algorithm 4) to RCLB (Algorithm 1), we note that while both algorithms share the same general structure, the key difference between the two stems from the manner in which the robust confidence thresholds are computed. In particular, to tackle the difficulty posed by the non-linearity of the observation map, we first compute a robust estimate $\hat{\theta}^{(\ell)}$ of $\theta_*$ at the server - a route that we avoided in RCLB - and then use such an estimate to devise a phased elimination rule.

### E.1 Proof of Theorem 3

In this section, we prove Theorem 3. The crux of the analysis lies in deriving a robust confidence bound akin to that in Lemma 1. To work towards such a result, we need to first go through a few intermediate steps; these are as follows.

**Step 1.** Prove that conditioned on $\mathcal{F}_\ell$, for each $i \in [M] \setminus \mathcal{B}$, $\tilde{V}_\ell^{-1/2} Y_{i,\ell}$ is a $d$-dimensional Gaussian random variable with mean $\tilde{V}_\ell^{-1/2} h_\ell(\theta_*)$, and covariance matrix $\Sigma = I_d$.[6]

**Step 2.** Use the result from Step 1, along with the error-bounds of the iteratively reweighted Gaussian mean estimator from [22], to derive a high-probability error-bound on $\|X_\ell - \tilde{V}_\ell^{-1/2} h_\ell(\theta_*)\|$.

**Step 3.** Exploit regularity properties of the link function in tandem with the bounds from Step 2, to derive high-probability error bounds on $|\mu(\langle \hat{\theta}^{(\ell)}, a \rangle) - \mu(\langle \theta_*, a \rangle)|$, for each $a \in \mathcal{A}_\ell$.

We now proceed to formally establish each of the above steps, starting with step 1.

**Lemma 7.** *For each epoch $\ell$, and each good agent $i \in [M] \setminus \mathcal{B}$, it holds that:*

$$\mathbb{E}\left[\tilde{V}_\ell^{-1/2} Y_{i,\ell} | \mathcal{F}_\ell\right] = \tilde{V}_\ell^{-1/2} h_\ell(\theta_*); \quad and \tag{48}$$

$$\mathbb{E}\left[\left(\tilde{V}_\ell^{-1/2} Y_{i,\ell} - \tilde{V}_\ell^{-1/2} h_\ell(\theta_*)\right)\left(\tilde{V}_\ell^{-1/2} Y_{i,\ell} - \tilde{V}_\ell^{-1/2} h_\ell(\theta_*)\right)' | \mathcal{F}_\ell\right] = I_d. \tag{49}$$

*Proof.* Fix an epoch $\ell$, and a good agent $i \in [M] \setminus \mathcal{B}$. We start by observing that:

$$\begin{aligned}
Y_{i,\ell} &= \sum_{a \in \text{Supp}(\pi_\ell)} m_a^{(\ell)} r_{i,a}^{(\ell)} a \\
&= \sum_{a \in \text{Supp}(\pi_\ell)} m_a^{(\ell)} \left(\mu\left(\langle \theta_*, a \rangle\right) + \bar{\eta}_{i,a}^{(\ell)}\right) a \\
&= h_\ell(\theta_*) + \sum_{a \in \text{Supp}(\pi_\ell)} m_a^{(\ell)} \bar{\eta}_{i,a}^{(\ell)} a,
\end{aligned} \tag{50}$$

where for the last step, we used the definition of $h_\ell(\theta_*)$. Just as in the proof of Theorem 1, we have used $\bar{\eta}_{i,a}^{(\ell)}$ to denote the average of the noise terms associated with the rewards observed by agent $i$ during phase $\ell$ for arm $a$. We thus have:

$$\tilde{V}_\ell^{-1/2} Y_{i,\ell} = \tilde{V}_\ell^{-1/2} h_\ell(\theta_*) + \tilde{V}_\ell^{-1/2}\left(\sum_{a \in \text{Supp}(\pi_\ell)} m_a^{(\ell)} \bar{\eta}_{i,a}^{(\ell)} a\right).$$

Now conditioned on $\mathcal{F}_\ell$, the only randomness in the above equation stems from the noise terms $\{\bar{\eta}_{i,a}^{(\ell)}\}, a \in \text{Supp}(\pi_\ell)$, that are each zero-mean. The claim in Eq. (48) thus follows.

Based on Eq. (50), we have:

$$\begin{aligned}
\mathbb{E}\left[(Y_{i,\ell} - h_\ell(\theta_*))(Y_{i,\ell} - h_\ell(\theta_*))' | \mathcal{F}_\ell\right] &= \mathbb{E}\left[\left(\sum_{a \in \text{Supp}(\pi_\ell)} m_a^{(\ell)} \bar{\eta}_{i,a}^{(\ell)} a\right)\left(\sum_{a \in \text{Supp}(\pi_\ell)} m_a^{(\ell)} \bar{\eta}_{i,a}^{(\ell)} a\right)' \Big| \mathcal{F}_\ell\right] \\
&\overset{(a)}{=} \sum_{a \in \text{Supp}(\pi_\ell)} \left(m_a^{(\ell)}\right)^2 \mathbb{E}\left[\left(\bar{\eta}_{i,a}^{(\ell)}\right)^2\right] aa' \\
&\overset{(b)}{=} \sum_{a \in \text{Supp}(\pi_\ell)} m_a^{(\ell)} aa' \\
&\overset{(c)}{=} \tilde{V}_\ell.
\end{aligned} \tag{51}$$

---

[6]Recall that $\mathcal{F}_\ell$ is the $\sigma$-algebra generated by all the actions and rewards up to the beginning of epoch $\ell$.

In the above steps, (a) follows by observing that the noise terms are independent across different arms; hence, the expectation of each of the cross-terms vanish. For (b), we used the fact that $\bar{\eta}_{i,a}^{(\ell)}$ is the average of $m_a^{(\ell)}$ independent Gaussian noise terms, each with zero-mean and unit variance; hence, $\mathbb{E}\left[\left(\bar{\eta}_{i,a}^{(\ell)}\right)^2\right] = 1/(m_a^{(\ell)})$. For (c), we simply used the definition of $\tilde{V}_\ell$. In light of Eq. (51), it is easy to see why Eq. (49) holds. $\qquad\square$

We now state - adapted to our notation - one of the main convergence guarantees from [22] for the iteratively reweighted mean estimator.

**Lemma 8.** *Suppose we are given $M$ $d$-dimensional samples $x_1, \ldots, x_M$, such that $(1-\alpha)M$ of these samples are drawn i.i.d. from $\mathcal{N}(v, \Sigma)$, where $v \in \mathbb{R}^d$ is an unknown mean vector, and $\Sigma \in \mathbb{R}^{d \times d}$ is a known covariance matrix. The remaining $\alpha M$ samples can be arbitrary. Let the corruption fraction $\alpha$ satisfy $\alpha < (5 - \sqrt{5})/10$, and let $\delta \in (16\exp(-M), 1)$ be a given tolerance level. Then, with probability at least $1 - \delta$, we have*

$$\|\hat{v} - v\|_2 \leq C_1 \|\Sigma\|_2^{1/2} \left( \sqrt{\frac{d + \log(16/\delta)}{M}} + \alpha\sqrt{\log\frac{1}{\alpha}} \right), \tag{52}$$

*where $C_1$ is a suitably large universal constant, and $\hat{v}$ is the output of the Iteratively Reweighted Mean Estimator, namely Algorithm 1 in [22], when it takes as input the $M$ samples, the covariance matrix $\Sigma$, and the corruption fraction $\alpha$.*

Let us now see how the above bound can assist in our cause. Fix any epoch $\ell$, and recall that $\delta_\ell = \bar{\delta}/(K\ell^2)$, where $\bar{\delta} = \delta/(10K)$, and $\delta$ is the given confidence parameter. Suppose we want to derive an error-bound based on Lemma 8 that holds with probability at least $1 - \delta_\ell$. For this to happen, we need $\delta_\ell > 16\exp(-M)$. Since $\ell \leq T$, one can verify that the aforementioned condition is satisfied as long as $M$ is large enough in the following sense:

$$M > \log\left(\frac{160K^2T^2}{\delta}\right). \tag{53}$$

From now on, we assume that the above condition holds. Next, recall that

$$X_\ell = \mathtt{ITW}(\{\tilde{V}_\ell^{-1/2}Y_{i,\ell}, i \in [M]\}).$$

Based on Lemma 7, Lemma 8, and the same line of reasoning as used to arrive at Eq. (24), we have that with probability at least $1 - \delta_\ell$:

$$\|X_\ell - \tilde{V}_\ell^{-1/2}h_\ell(\theta_*)\| \leq C_1 \left( \sqrt{\frac{d + \log(16/\delta_\ell)}{M}} + \alpha\sqrt{\log\frac{1}{\alpha}} \right). \tag{54}$$

We will call upon the above bound later in our analysis. For now, this ends Step 2. As for Step 3, we start with the following result.

**Lemma 9.** *Consider any $\theta_1, \theta_2 \in \Theta$, and any epoch $\ell$. There exists a symmetric positive definite matrix $G_\ell(\theta_1; \theta_2)$ satisfying $k_1\tilde{V}_\ell \preccurlyeq G_\ell(\theta_1; \theta_2) \preccurlyeq k_2\tilde{V}_\ell$, such that:*
$$h_\ell(\theta_1) - h_\ell(\theta_2) = G_\ell(\theta_1; \theta_2)(\theta_1 - \theta_2). \tag{55}$$

*Proof.* For any $\theta \in \Theta$, let us denote by $\nabla h_\ell(\theta)$ the Jacobian matrix of $h_\ell(\cdot)$ at $\theta$. Such a matrix exists based on Assumption 1. Now based on the mean value theorem, $\exists \alpha \in (0, 1)$ such that
$$h_\ell(\theta_1) - h_\ell(\theta_2) = (\nabla h_\ell(\alpha\theta_1 + (1-\alpha)\theta_2))(\theta_1 - \theta_2).$$
Let $\bar{\theta} = \alpha\theta_1 + (1-\alpha)\theta_2$, and $G_\ell(\theta_1; \theta_2) = \nabla h_\ell(\bar{\theta})$. To complete the proof, we need to show that the matrix $G_\ell(\theta_1; \theta_2)$ defined above is symmetric, positive definite, and bounded above and below (in the Loewner sense) by scalar multiples of $\tilde{V}_\ell$. To that end, observe that:

$$\begin{aligned} \nabla h_\ell(\bar{\theta}) &\overset{(a)}{=} \sum_{a \in \mathrm{Supp}(\pi_\ell)} m_a^{(\ell)}\dot{\mu}(\langle\bar{\theta}, a\rangle)aa' \\ &\overset{(b)}{\succcurlyeq} k_1\left( \sum_{a \in \mathrm{Supp}(\pi_\ell)} m_a^{(\ell)}aa' \right) \\ &= k_1\tilde{V}_\ell. \end{aligned} \tag{56}$$

In the above steps, we used the definition of $h_\ell(\cdot)$ for (a); and for (b), we used Assumption 1. From the above steps, it is clear that $G_\ell(\theta_1; \theta_2)$ is symmetric. That it is also positive definite follows from the fact that $\tilde{V}_\ell \succ 0$. Finally, using the fact that $\mu(\cdot)$ is $k_2$-Lipschitz, and a similar line of reasoning, one can show that $G_\ell(\theta_1; \theta_2) \preccurlyeq k_2 \tilde{V}_\ell$. This concludes the proof. $\qquad\square$

We now have all the pieces required to establish an analogue of Lemma 1.

**Lemma 10.** (**_Robust Confidence Intervals for_** `RC-GLM`) *Suppose $M$ satisfies the condition in Eq. (53), and $\alpha < (5 - \sqrt{5})/10$. Fix any epoch $\ell$. For each arm $a \in \mathcal{A}_\ell$, with probability at least $1 - \delta_\ell$, it holds that:*

$$|\mu(\langle \hat{\theta}^{(\ell)}, a \rangle) - \mu(\langle \theta_*, a \rangle)| \leq \bar{\gamma}_\ell, \ \ where \ \ \bar{\gamma}_\ell = 4C_1 \frac{k_2}{k_1} \left( \sqrt{d} + \alpha \sqrt{M \log(1/\alpha)} \right) \epsilon_\ell, \qquad (57)$$

*and $C_1$ is as in Lemma 8.*

*Proof.* Let us start by conditioning on the event of measure at least $1 - \delta_\ell$ on which Eq. (54) holds. Invoking Lemma 9, we know that there exists a symmetrix positive definite matrix $G_\ell$ such that $k_1 \tilde{V}_\ell \preccurlyeq G_\ell \preccurlyeq k_2 \tilde{V}_\ell$, and:[7]

$$\begin{aligned} G_\ell \left( \hat{\theta}^{(\ell)} - \theta_* \right) &= h_\ell(\hat{\theta}^{(\ell)}) - h_\ell(\theta_*) \\ &= \tilde{V}_\ell^{1/2} X_\ell - h_\ell(\theta_*) \\ &= \tilde{V}_\ell^{1/2} \left( X_\ell - \tilde{V}_\ell^{-1/2} h_\ell(\theta_*) \right). \end{aligned} \qquad (58)$$

For the second step above, we used the fact that based on line 9 of `RC-GLM`, $h_\ell(\hat{\theta}^{(\ell)}) = \tilde{V}_\ell^{1/2} X_\ell$. Now fix any arm $a \in \mathcal{A}_\ell$, and observe that:

$$\langle \hat{\theta}^{(\ell)} - \theta_*, a \rangle = \left\langle G_\ell^{-1} \tilde{V}_\ell^{1/2} \left( X_\ell - \tilde{V}_\ell^{-1/2} h_\ell(\theta_*) \right), a \right\rangle.$$

This, in turn, implies:

$$|\langle \hat{\theta}^{(\ell)} - \theta_*, a \rangle| \leq \underbrace{\left\| G_\ell^{-1} \tilde{V}_\ell^{1/2} \left( X_\ell - \tilde{V}_\ell^{-1/2} h_\ell(\theta_*) \right) \right\|_{\tilde{V}_\ell}}_{T_1} \|a\|_{\tilde{V}_\ell^{-1}}. \qquad (59)$$

We bound $T_1$ as follows.

$$\begin{aligned} T_1 &= \sqrt{\left( X_\ell - \tilde{V}_\ell^{-1/2} h_\ell(\theta_*) \right)' \tilde{V}_\ell^{1/2} G_\ell^{-1} \tilde{V}_\ell G_\ell^{-1} \tilde{V}_\ell^{1/2} \left( X_\ell - \tilde{V}_\ell^{-1/2} h_\ell(\theta_*) \right)} \\ &\overset{(a)}{\leq} \frac{1}{\sqrt{k_1}} \sqrt{\left( X_\ell - \tilde{V}_\ell^{-1/2} h_\ell(\theta_*) \right)' \tilde{V}_\ell^{1/2} G_\ell^{-1} \tilde{V}_\ell^{1/2} \left( X_\ell - \tilde{V}_\ell^{-1/2} h_\ell(\theta_*) \right)} \\ &\overset{(b)}{\leq} \frac{1}{k_1} \sqrt{\left( X_\ell - \tilde{V}_\ell^{-1/2} h_\ell(\theta_*) \right)' \left( X_\ell - \tilde{V}_\ell^{-1/2} h_\ell(\theta_*) \right)} \\ &\overset{(c)}{\leq} \frac{C_1}{k_1} \left( \sqrt{\frac{d + \log(16/\delta_\ell)}{M}} + \alpha \sqrt{\log \frac{1}{\alpha}} \right). \end{aligned} \qquad (60)$$

For both (a) and (b) above, we used $k_1 \tilde{V}_\ell \preccurlyeq G_\ell$; for (c), we invoked the bound in Eq. (54). Now recall from Eq. (22) that

$$\|a\|_{\tilde{V}_\ell^{-1}} \leq \epsilon_\ell \sqrt{\frac{2M}{\log\left(\frac{1}{\delta_\ell}\right)}}.$$

Combining the above bound with the ones in Eq. (59) and Eq. (60), and using $\log(1/\delta_\ell) \geq 1$, we obtain

$$|\langle \hat{\theta}^{(\ell)} - \theta_*, a \rangle| \leq \frac{\sqrt{2}C_1}{k_1} \left( \sqrt{d + \frac{\log(16/\delta_\ell)}{\log(1/\delta_\ell)}} + \alpha \sqrt{M \log \frac{1}{\alpha}} \right) \epsilon_\ell.$$

---

[7]Here, we have dropped the dependence of $G_\ell$ on $\hat{\theta}^{(\ell)}$ and $\theta_*$ to lighten the notation.

Elementary calculations coupled with the fact that $\log(1/\delta_\ell) \geq 1$ yields:

$$\frac{\log(16/\delta_\ell)}{\log(1/\delta_\ell)} \leq 4.$$

Putting all the pieces together, and simplifying, we arrive at the following bound:

$$|\langle \hat{\theta}^{(\ell)} - \theta_*, a \rangle| \leq \frac{4C_1}{k_1} \left( \sqrt{d} + \alpha \sqrt{M \log \frac{1}{\alpha}} \right) \epsilon_\ell. \tag{61}$$

Using the fact that $\mu(\cdot)$ is $k_2$-Lipschitz then yields:

$$|\mu(\langle \hat{\theta}^{(\ell)}, a \rangle) - \mu(\langle \theta_*, a \rangle)| \leq k_2 |\langle \hat{\theta}^{(\ell)} - \theta_*, a \rangle| \leq 4C_1 \frac{k_2}{k_1} \left( \sqrt{d} + \alpha \sqrt{M \log \frac{1}{\alpha}} \right) \epsilon_\ell,$$

which is the desired claim. This completes the proof. $\qquad \square$

Having derived the robust confidence bounds for `RC-GLM`, we can complete the proof of Theorem 3.

*Proof.* (**Proof of Theorem 3**). Using essentially the same arguments as those used to prove Lemmas 4 and 5, we can prove that there exists a clean event, say $\mathcal{E}$, of measure at least $1 - \delta$, such that on $\mathcal{E}$, the following hold: (i) $a_* \in \mathcal{A}_\ell, \forall \ell \in [L]$, where $L$ is the total number of epochs; and (ii) for any epoch $\ell \in [L], a \in \mathcal{A}_\ell \implies \tilde{\Delta}_a \leq 8\bar{\gamma}_\ell$. Here, $\tilde{\Delta}_a = \mu(\langle \theta_*, a_* \rangle) - \mu(\langle \theta_*, a \rangle)$. Let us condition on this clean event $\mathcal{E}$. The remainder of the proof follows the same line of reasoning as that of Theorem 1. For any good agent $i \in [M] \setminus \mathcal{B}$, we can bound the regret as follows.

$$\sum_{t=1}^{T} \left( \mu(\langle \theta_*, a_* \rangle) - \mu(\langle \theta_*, a_{i,t} \rangle) \right) = \sum_{\ell=1}^{L} \sum_{a \in \text{Supp}(\pi_\ell)} m_a^{(\ell)} \left( \mu(\langle \theta_*, a_* \rangle) - \mu(\langle \theta_*, a \rangle) \right)$$

$$= \sum_{\ell=1}^{L} \sum_{a \in \text{Supp}(\pi_\ell)} \left\lceil \frac{T_a^{(\ell)}}{M} \right\rceil \tilde{\Delta}_a \tag{62}$$

$$\leq \underbrace{\sum_{\ell=1}^{L} \sum_{a \in \text{Supp}(\pi_\ell)} \frac{T_a^{(\ell)}}{M} \tilde{\Delta}_a}_{T_1} + \underbrace{\sum_{\ell=1}^{L} \sum_{a \in \text{Supp}(\pi_\ell)} \tilde{\Delta}_a}_{T_2}.$$

As in the proof of Theorem 1, we bound $T_1$ and $T_2$ separately. For bounding $T_1$, we have:

$$T_1 = \sum_{\ell=1}^{L} \sum_{a \in \text{Supp}(\pi_\ell)} \frac{T_a^{(\ell)}}{M} \tilde{\Delta}_a$$

$$= \frac{d}{M} \sum_{\ell=1}^{L} \frac{1}{\epsilon_\ell^2} \log \left( \frac{1}{\delta_\ell} \right) \sum_{a \in \text{Supp}(\pi_\ell)} \pi_\ell(a) \tilde{\Delta}_a$$

$$\leq \frac{8d}{M} \sum_{\ell=1}^{L} \frac{1}{\epsilon_\ell^2} \log \left( \frac{1}{\delta_\ell} \right) \bar{\gamma}_\ell \tag{63}$$

$$= \frac{32C_1 d \left( \sqrt{d} + \alpha \sqrt{M \log(1/\alpha)} \right)}{M} \left( \frac{k_2}{k_1} \right) \sum_{\ell=1}^{L} \frac{1}{\epsilon_\ell} \log \left( \frac{10K^2 \ell^2}{\delta} \right)$$

$$\leq \frac{32C_1 d \left( \sqrt{d} + \alpha \sqrt{M \log(1/\alpha)} \right)}{M} \left( \frac{k_2}{k_1} \right) \log \left( \frac{10K^2 L^2}{\delta} \right) \sum_{\ell=1}^{L} 2^\ell$$

$$= O \left( \left( \frac{k_2}{k_1} \right) \frac{d \left( \sqrt{d} + \alpha \sqrt{M \log(1/\alpha)} \right)}{M} \log \left( \frac{10K^2 L^2}{\delta} \right) 2^L \right).$$

Recall the following fact that we proved earlier for Theorem 1:

$$2^L \leq \sqrt{\frac{MT}{d \log\left(10K^2L^2/\delta\right)}}.$$

Plugging this bound in Eq. (63), we obtain:

$$T_1 = O\left(\left(\frac{k_2}{k_1}\right)\left(\alpha\sqrt{\log(1/\alpha)} + \sqrt{\frac{d}{M}}\right)\sqrt{\log\left(\frac{KT}{\delta}\right)dT}\right).$$

One can upper-bound $T_2$ using the same bound as above using exactly the same steps as in the proof of Theorem 1. Combining the bounds on $T_1$ and $T_2$ leads to the desired claim. $\qquad\square$

• **Comments on solving for $\hat{\theta}^{(\ell)}$ in RC-GLM.** Recall that line 9 of Algorithm RC-GLM requires the server to solve for $\hat{\theta}^{(\ell)}$ based on the following equation:

$$h_\ell(\hat{\theta}^{(\ell)}) = \tilde{V}_\ell^{1/2} X_\ell. \tag{64}$$

Based on Lemma 9, we have that for any $\theta_1, \theta_2 \in \Theta$ such that $\theta_1 \neq \theta_2$:

$$(\theta_1 - \theta_2)'\left(h_\ell(\theta_1) - h_\ell(\theta_2)\right) = (\theta_1 - \theta_2)' G_\ell(\theta_1; \theta_2)(\theta_1 - \theta_2) > 0,$$

since $G_\ell(\theta_1; \theta_2)$ is positive-definite. Thus, the map $h_\ell : \mathbb{R}^d \to \mathbb{R}^d$ is injective, and $h_\ell^{-1}$ is well-defined. This, in turn, implies that Eq. (64) has a unique solution.

### E.2 The Iteratively Reweighted Mean Estimation Algorithm

In this section, we briefly explain the main idea behind the Iteratively Reweighted Mean Estimation Algorithm in [22]. Suppose we are given an $\alpha$-corrupted set $\mathcal{X}$ of samples, namely, $M$ $d$-dimensional samples $x_1, \ldots, x_M$, such that $(1 - \alpha)M$ of these samples are drawn i.i.d. from $\mathcal{N}(v, \Sigma)$, and the remaining $\alpha M$ samples are arbitrarily corrupted by an adversary. The goal is to recover the unknown mean vector $v \in \mathbb{R}^d$, given knowledge of the samples $\mathcal{X}$, the corruption fraction $\alpha$, and the covariance matrix $\Sigma$. To see how this is done, let us define a couple of quantities for any pair of vectors $w \in [0, 1]^d$ and $\mu \in \mathbb{R}^d$:

$$\bar{x}_w = \sum_{i=1}^M w_i x_i; \quad G(w, \mu) = \lambda_{\max}\left(\sum_{i=1}^M w_i (x_i - \mu)(x_i - \mu)' - \Sigma\right). \tag{65}$$

The basic idea is to find a weight vector $\hat{w}$ within the $d$-dimensional probability simplex such that the weighted average $\bar{x}_{\hat{w}}$ is close to the true mean $v$. Intuitively, a "small" value of $G(\hat{w}, \bar{x}_{\hat{w}})$ is an indicator of a good candidate for such a weight vector. This is essentially the strategy pursued in Algorithm 5 where one iteratively updates the weight vectors, and the associated weighted averages, so as to minimize the function $G(\cdot, \cdot)$ defined in Eq. (65). At the termination of this algorithm, say after $N$ iterations, the goal is to output a weight vector $\hat{w}_N$ that mimics the ideal weight vector $w^*$ defined by: $w_j^* = \mathbf{1}(j \in \mathcal{I})/|\mathcal{I}|$, where $\mathcal{I}$ is the set of good samples (inliers). The steps of the Iteratively Reweighted Mean Estimator are outlined in Algorithm 5.

---
**Algorithm 5** Iteratively Reweighted Mean Estimator (`ITW`)
---
1: **Input:** $\alpha$-corrupted set of $M$ samples $\mathcal{X}$, corruption fraction $\alpha$, and covariance matrix $\Sigma$.

2: **Output:** Robust estimate of the mean $\hat{v}$.

3: **Initialize:** Compute $\hat{v}_0$ as a minimizer of $\underset{\mu}{\mathrm{argmin}} \sum\limits_{i=1}^{M} \|x_i - \mu\|$.

4: Let $N = 0 \vee \left\lceil \frac{\log(4r_\Sigma) - 2\log(\alpha(1-2\alpha))}{2\log((1-2\alpha)) - \log(\alpha) - \log(1-\alpha)} \right\rceil$. Here, $r_\Sigma = \mathrm{Trace}(\Sigma)/\|\Sigma\|_2$.

5:

6: **for** $k = 1$ to $N$ **do**

7:     Compute current weights:

$$w \in \underset{(M-M\epsilon)\|w\|_\infty \leq 1}{\mathrm{argmin}} \lambda_{\max} \left( \sum_{i=1}^{M} w_i(x_i - \hat{v}_{k-1})(x_i - \hat{v}_{k-1})' - \Sigma \right) \vee 0.$$

8:     Update the estimator:

$$\hat{v}_k = \sum_{i=1}^{M} w_i x_i.$$

9: **end for**

10: **Return** $\hat{v} = \hat{v}_K$.
---

# F  Analysis for the Contextual Bandit Setting: Proof of Theorem 4

The proof of Theorem 4 proceeds in multiple steps. We start with an analysis of the Robust BaseLinUCB subroutine, namely Algorithm 2.

**Lemma 11.** *(**Bounds for Robust BaseLinUCB**) Suppose the input index set $\Psi_t$ is constructed so that for fixed $x_{\tau,a_\tau}, \tau \in \Psi_t$, the rewards $\{r_{i,\tau}\}_{\tau \in \Psi_t}$ are independent random variables for each good agent $i \in [M] \setminus \mathcal{B}$. Then, for each $a \in [K]$, with probability at least $1 - \bar{\delta}$, it holds that:*

$$|\hat{r}_{t,a} - \langle \theta_*, x_{t,a} \rangle| \leq \left( \alpha + 2C\sqrt{\frac{\log\left(\frac{1}{\bar{\delta}}\right)}{M}} \right) \|x_{t,a}\|_{A_t^{-1}}, \tag{66}$$

*where $C$ is as in Lemma 2.*

*Proof.* Fix any agent $i \in [M] \setminus \mathcal{B}$. Now from the expression for $\hat{\theta}_{i,t}$ in Algorithm 2, observe that:

$$
\begin{aligned}
\hat{\theta}_{i,t} &= A_t^{-1} b_{i,t} \\
&= A_t^{-1} \left( \sum_{\tau \in \Psi_t} r_{i,\tau} x_{\tau,a_\tau} \right) \\
&= A_t^{-1} \left( \sum_{\tau \in \Psi_t} \left( \langle \theta_*, x_{\tau,a_\tau} \rangle + \eta_{i,\tau} \right) x_{\tau,a_\tau} \right) \\
&= A_t^{-1} \left( \sum_{\tau \in \Psi_t} x_{\tau,a_\tau} x'_{\tau,a_\tau} \right) \theta_* + A_t^{-1} \left( \sum_{\tau \in \Psi_t} \eta_{i,\tau} x_{\tau,a_\tau} \right) \\
&= \left( I_d - \frac{A_t^{-1}}{M} \right) \theta_* + A_t^{-1} \left( \sum_{\tau \in \Psi_t} \eta_{i,\tau} x_{\tau,a_\tau} \right),
\end{aligned}
\tag{67}
$$

where we used $\eta_{i,\tau}$ as a shorthand for $\eta_{i,\tau}(a_\tau)$. Now consider any $a \in [K]$. Using the above expression, we will now decompose the error in estimation of $\langle \theta_*, x_{t,a} \rangle$ into a bias term and a variance term:

$$\langle \hat{\theta}_{i,t} - \theta_*, x_{t,a} \rangle = \underbrace{-\frac{1}{M} \langle A_t^{-1} \theta_*, x_{t,a} \rangle}_{\text{bias term}} + \underbrace{\sum_{\tau \in \Psi_t} \langle A_t^{-1} x_{\tau,a_\tau}, x_{t,a} \rangle \eta_{i,\tau}}_{\text{variance term}}. \tag{68}$$

For a fixed set of feature vectors, under our assumption that $\{r_{i,\tau}\}_{\tau \in \Psi_t}$ are independent random variables, the variance term is a sum of independent zero-mean Gaussian noise variables. Thus,

$$\mathbb{E}\left[ \sum_{\tau \in \Psi_t} \langle A_t^{-1} x_{\tau,a_\tau}, x_{t,a} \rangle \eta_{i,\tau} \right] = \sum_{\tau \in \Psi_t} \langle A_t^{-1} x_{\tau,a_\tau}, x_{t,a} \rangle \mathbb{E}\left[ \eta_{i,\tau} \right] = 0.$$

Furthermore, we have:

$$
\begin{aligned}
\mathbb{E}\left[ \left( \sum_{\tau \in \Psi_t} \langle A_t^{-1} x_{\tau,a_\tau}, x_{t,a} \rangle \eta_{i,\tau} \right)^2 \right] &= x'_{t,a} A_t^{-1} \left( \sum_{\tau \in \Psi_t} \mathbb{E}\left[ (\eta_{i,\tau}^2) \right] x_{\tau,a_\tau} x'_{\tau,a_\tau} \right) A_t^{-1} x_{t,a} \\
&= x'_{t,a} A_t^{-1} \left( \sum_{\tau \in \Psi_t} x_{\tau,a_\tau} x'_{\tau,a_\tau} \right) A_t^{-1} x_{t,a} \\
&\leq x'_{t,a} A_t^{-1} \underbrace{\left( \frac{I_d}{M} + \sum_{\tau \in \Psi_t} x_{\tau,a_\tau} x'_{\tau,a_\tau} \right)}_{A_t} A_t^{-1} x_{t,a} \\
&= \|x_{t,a}\|_{A_t^{-1}}^2.
\end{aligned}
\tag{69}
$$

From the above arguments, we conclude that for each $i \in [M] \setminus \mathcal{B}$,

$$\langle \hat{\theta}_{i,t}, x_{t,a} \rangle \sim \mathcal{N}\left( \langle \theta_* - \frac{A_t^{-1}\theta_*}{M}, x_{t,a} \rangle, \sigma^2 \right), \quad \text{where} \quad \sigma^2 \leq \|x_{t,a}\|^2_{A_t^{-1}}.$$

Since the noise samples are independent across agents, we also know that $\{\langle \hat{\theta}_{i,t}, x_{t,a} \rangle\}_{i \in [M] \setminus \mathcal{B}}$ are independent. Recalling that $\hat{r}_{t,a} = \texttt{Median}\left( \{\langle \hat{\theta}_{i,t}, x_{t,a} \rangle, i \in [M]\} \right)$, and invoking Lemma 2, we then have that with probability at least $1 - \bar{\delta}$,

$$|\hat{r}_{t,a} - \langle \theta_* - \frac{A_t^{-1}\theta_*}{M}, x_{t,a} \rangle| \leq \left( \alpha + C\sqrt{\frac{\log\left(\frac{1}{\bar{\delta}}\right)}{M}} \right) \|x_{t,a}\|_{A_t^{-1}}. \tag{70}$$

This immediately implies:

$$|\hat{r}_{t,a} - \langle \theta_*, x_{t,a} \rangle| \leq \frac{1}{M}|\langle A_t^{-1}\theta_*, x_{t,a} \rangle| + \left( \alpha + C\sqrt{\frac{\log\left(\frac{1}{\bar{\delta}}\right)}{M}} \right) \|x_{t,a}\|_{A_t^{-1}}. \tag{71}$$

It remains to bound the first term in the above display. To that end, we proceed as follows:

$$\begin{aligned}
\frac{1}{M}|\langle A_t^{-1}\theta_*, x_{t,a} \rangle| &= \frac{1}{M}|\langle A_t^{-1}x_{t,a}, \theta_* \rangle| \\
&\leq \frac{1}{M}\|A_t^{-1}x_{t,a}\|\|\theta_*\| \\
&\overset{(a)}{\leq} \frac{1}{\sqrt{M}}\sqrt{x'_{t,a}A_t^{-1}\frac{I_d}{M}A_t^{-1}x_{t,a}} \\
&\leq \frac{1}{\sqrt{M}}\sqrt{x'_{t,a}A_t^{-1}\left(\frac{I_d}{M} + \sum_{\tau \in \Psi_t} x_{\tau,a_\tau}x'_{\tau,a_\tau}\right)A_t^{-1}x_{t,a}} \\
&= \frac{1}{\sqrt{M}}\|x_{t,a}\|_{A_t^{-1}} \\
&\overset{(b)}{\leq} C\sqrt{\frac{\log\left(\frac{1}{\bar{\delta}}\right)}{M}}\|x_{t,a}\|_{A_t^{-1}},
\end{aligned} \tag{72}$$

where for (a), we used the fact that $\|\theta_*\| \leq 1$ by assumption; and for (b), we used $C \geq 1, \log\left(\frac{1}{\bar{\delta}}\right) \geq 1$. Combining the above bound with the one in Eq.(71) leads to the desired claim. $\square$

For Lemma 11 to hold, the crucial requirement is for the rewards corresponding to indices in $\Psi_t$ to be independent. Our next result shows that this is indeed the case; the proof of this lemma essentially follows the same arguments as that of [24, Lemma 14], we reproduce these arguments here only for completeness.

**Lemma 12.** *(Independence of Samples) Fix any agent $i \in [M] \setminus \mathcal{B}$. For each $s \in [S]$ and each $t \in [T]$, given any fixed sequence of feature vectors $\{x_{\tau,a_\tau}\}_{\tau \in \Psi_t^{(s)}}$, the rewards $\{r_{i,\tau}\}_{\tau \in \Psi_t^{(s)}}$ are independent random variables.*

*Proof.* Let us start by observing that a time-step $t$ can be added to $\Psi_t^{(s)}$ only in line 6 of Algorithm 3. Thus, the event $\{t \in \Psi_t^{(s)}\}$ only depends on all prior phases $\cup_{\ell < s}\Psi_t^{(\ell)}$, and on the confidence width $w_{t,a}^{(s)}$. From the definition of $w_{t,a}^{(s)}$ in line 7 of Algorithm 2, we note that $w_{t,a}^{(s)}$ depends only on the feature vectors $x_{\tau,a_\tau}, \tau \in \Psi_t^{(s)}$, and $x_{t,a}$. Combining the above observations, it is easy to see that $\{t \in \Psi_t^{(s)}\}$ only depends on the feature vectors. Noting that the feature vector sequence is fixed, and cannot be controlled by the adversarial agents, leads to the claim of the lemma. $\square$

The next result tells us that with high-probability, at each time-step $t \in [T]$, (i) the best arm $a_t^*$ is retained in all stages of the screening process; and (ii) an active arm in phase $s$ can contribute to at most $8/(2^s\sqrt{M})$ instantaneous regret.

**Lemma 13.** *With probability at least* $1 - \bar{\delta}KST$, *for any* $t \in [T]$ *and any* $s \in [S]$, *the following hold:*

(i) $|\hat{r}_{t,a}^{(s)} - \langle \theta_*, x_{t,a} \rangle| \leq w_{t,a}^{(s)}, \forall a \in \mathcal{A}_s.$

(ii) $a_t^* \in \mathcal{A}_s.$

(iii) $\langle \theta_*, x_{t,a_t^*} \rangle - \langle \theta_*, x_{t,a} \rangle \leq 8/(2^s \sqrt{M}), \forall a \in \mathcal{A}_s.$

*Proof.* Part (i) of the result follows directly from Lemma 11, and an union bound over all time-steps, phases and arms.

For part (ii), let us condition on the clean event, say $\mathcal{E}$, on which part (i) holds. From the rules of Algorithm 3, it holds trivially that $a_t^* \in \mathcal{A}_s$ for $s = 1$. Now suppose there exists some phase $s > 1$ such that $a_t^* \in \mathcal{A}_{s-1}$, but $a_t^* \notin \mathcal{A}_s$. From line 8 in Algorithm 3, we must have $w_{t,a}^{(s-1)} \leq 2^{1-s}/\sqrt{M}, \forall a \in \mathcal{A}_{s-1}$. From the phased elimination strategy in line 8 of Algorithm 3, $a_t^* \notin \mathcal{A}_s$ implies the existence of some arm $a \in \mathcal{A}_s$ such that:

$$
\begin{aligned}
&\left( \hat{r}_{t,a}^{(s-1)} + w_{t,a}^{(s-1)} \right) - \left( \hat{r}_{t,a_t^*}^{(s-1)} + w_{t,a_t^*}^{(s-1)} \right) > \frac{2^{2-s}}{\sqrt{M}} \\
&\overset{(a)}{\Longrightarrow} \langle \theta_*, x_{t,a} \rangle + 2w_{t,a}^{(s-1)} - \left( \hat{r}_{t,a_t^*}^{(s-1)} + w_{t,a_t^*}^{(s-1)} \right) > \frac{2^{2-s}}{\sqrt{M}} \\
&\overset{(b)}{\Longrightarrow} \langle \theta_*, x_{t,a} - x_{t,a_t^*} \rangle + 2w_{t,a}^{(s-1)} > \frac{2^{2-s}}{\sqrt{M}} \\
&\overset{(c)}{\Longrightarrow} 2w_{t,a}^{(s-1)} > \frac{2^{2-s}}{\sqrt{M}},
\end{aligned}
\tag{73}
$$

which leads to a contradiction as $w_{t,a}^{(s-1)} \leq 2^{1-s}/\sqrt{M}$. In the above steps, both (a) and (b) follow from the defining property of the clean event $\mathcal{E}$; for (c), we used $\langle \theta_*, x_{t,a} - x_{t,a_t^*} \rangle \leq 0$ from the optimality of $a_t^*$. This completes the proof of part (ii).

For part (iii), let us once again condition on the clean event $\mathcal{E}$ on which part (i) holds. Now $a \in \mathcal{A}_s \implies a \in \mathcal{A}_{s-1}$. We also know from part (ii) that $a_t^* \in \mathcal{A}_{s-1}$. The retention of arm $a$ in $\mathcal{A}_s$ implies (based on line 8 of Algorithm 3),

$$
\begin{aligned}
&\left( \hat{r}_{t,a_t^*}^{(s-1)} + w_{t,a_t^*}^{(s-1)} \right) - \left( \hat{r}_{t,a}^{(s-1)} + w_{t,a}^{(s-1)} \right) \leq \frac{2^{2-s}}{\sqrt{M}} \\
&\overset{(a)}{\Longrightarrow} \langle \theta_*, x_{t,a_t^*} \rangle - \left( \hat{r}_{t,a}^{(s-1)} + w_{t,a}^{(s-1)} \right) \leq \frac{2^{2-s}}{\sqrt{M}} \\
&\overset{(b)}{\Longrightarrow} \langle \theta_*, x_{t,a_t^*} - x_{t,a} \rangle \leq \frac{2^{2-s}}{\sqrt{M}} + 2w_{t,a}^{(s-1)} \\
&\overset{(c)}{\Longrightarrow} \langle \theta_*, x_{t,a_t^*} - x_{t,a} \rangle \leq \frac{8}{2^s \sqrt{M}},
\end{aligned}
\tag{74}
$$

which is the desired claim. Here, for (a) and (b) we used the defining property of $\mathcal{E}$. As for (c), we used the fact that $w_{t,a}^{(s-1)} \leq 2^{1-s}/\sqrt{M}$. $\qquad\square$

The final piece needed in the proof of Theorem 4 is a bound on $|\Psi_T^{(s)}|$ for each $s \in [S]$.

To that end, we will make use of the elliptical potential lemma from [16].

**Lemma 14.** *(**Elliptical Potential Lemma**) Let* $\{X_t\}_{t=1}^{\infty}$ *be a sequence in* $\mathbb{R}^d$, $V$ *a* $d \times d$ *positive definite matrix, and define* $\bar{V}_t = V + \sum_{\tau=1}^{t} X_\tau X_\tau'$. *If* $\|X_t\| \leq L$ *for all* $t$, *then we have that*

$$
\sum_{t=1}^{T} \min\{1, \|X_t\|_{\bar{V}_{t-1}^{-1}}^2\} \leq 2 \log \frac{\det(\bar{V}_T)}{\det(V)} \leq 2(d \log ((trace(V) + TL^2)/d) - \log(\det V)).
$$

We have the following result.

**Lemma 15.** (***Bound on*** $|\Psi_T^{(s)}|$) *Fix any* $s \in [S]$. *The following then holds:*

$$|\Psi_T^{(s)}| \leq 2^s \sqrt{M} \left( \alpha + 2C\sqrt{\frac{\log\left(\frac{1}{\delta}\right)}{M}} \right) \sqrt{2d|\Psi_T^{(s)}|\log(2M|\Psi_T^{(s)}|)}. \qquad (75)$$

*Proof.* Fix any phase $s \in [S]$. Now consider any time-step $t \in \Psi_T^{(s)}$. Since $t \in \Psi_T^{(s)}$, based on line 6 of Algorithm 3, it must be that:

$$\frac{1}{2^s \sqrt{M}} < w_{t,a_t}^{(s)} = \left( \alpha + 2C\sqrt{\frac{\log\left(\frac{1}{\delta}\right)}{M}} \right) \|x_{t,a_t}\|_{A_{s,t}^{-1}}, \quad \text{where} \qquad (76)$$

$$A_{s,t} = \left( \frac{I_d}{M} + \sum_{\tau \in \Psi_t^{(s)}} x_{\tau,a_\tau} x'_{\tau,a_\tau} \right).$$

Also, since $s \geq 1$, we have the following trivial inequality:

$$\frac{1}{2^s \sqrt{M}} < \left( \alpha + 2C\sqrt{\frac{\log\left(\frac{1}{\delta}\right)}{M}} \right).$$

Combining the above inequality with the one in (76), we note that for each $t \in \Psi_T^{(s)}$, it holds that:

$$\frac{1}{2^s \sqrt{M}} < \left( \alpha + 2C\sqrt{\frac{\log\left(\frac{1}{\delta}\right)}{M}} \right) \min\{1, \|x_{t,a_t}\|_{A_{s,t}^{-1}}\}.$$

Summing the above display over all indices in $\Psi_T^{(s)}$ yields:

$$\begin{aligned}
\frac{|\Psi_T^{(s)}|}{2^s \sqrt{M}} &< \left( \alpha + 2C\sqrt{\frac{\log\left(\frac{1}{\delta}\right)}{M}} \right) \sum_{t \in \Psi_T^{(s)}} \min\{1, \|x_{t,a_t}\|_{A_{s,t}^{-1}}\} \\
&\stackrel{(a)}{\leq} \left( \alpha + 2C\sqrt{\frac{\log\left(\frac{1}{\delta}\right)}{M}} \right) \sqrt{|\Psi_T^{(s)}| \sum_{t \in \Psi_T^{(s)}} \min\{1, \|x_{t,a_t}\|_{A_{s,t}^{-1}}^2\}} \\
&\stackrel{(b)}{\leq} \left( \alpha + 2C\sqrt{\frac{\log\left(\frac{1}{\delta}\right)}{M}} \right) \sqrt{2d|\Psi_T^{(s)}|\log\left(1 + \frac{M}{d}|\Psi_T^{(s)}|\right)} \\
&\leq \left( \alpha + 2C\sqrt{\frac{\log\left(\frac{1}{\delta}\right)}{M}} \right) \sqrt{2d|\Psi_T^{(s)}|\log\left(2M|\Psi_T^{(s)}|\right)},
\end{aligned} \qquad (77)$$

where (a) follows from Jensen's inequality, and (b) follows from an application of Lemma 14. Reorganizing the resulting inequality above leads to the desired claim. $\qquad \square$

We are now ready to prove Theorem 4.

*Proof.* (**Proof of Theorem 4**) Since $S = \lceil \log T \rceil$, we have $1/(2^S \sqrt{M}) \leq 1/(\sqrt{MT})$. Thus, from the rules of Algorithm 3, it is apparent that at every time-step $t$, an action $a_t$ is *always* chosen, either based on line 6, or on line 7. If we use $\Xi_T$ to store those time steps in $[T]$ where an action is chosen based on line 7 of Algorithm 3, then the above reasoning implies: $[T] = \Xi_T \cup \bigcup_{s \in [S]} \Psi_T^{(s)}$.

Throughout the rest of the proof, we will condition on the clean event on which items (i)-(iii) in Lemma 13 hold. We also recall that this clean event has measure at least $1 - \bar{\delta}KST$. Now fix any

good agent $i \in [M] \setminus \mathcal{B}$, and note that the same action $a_t$ is played by every good agent at time $t$. The cumulative regret for agent $i$ can thus be decomposed as follows:

$$\sum_{t=1}^{T} \left( \langle \theta_*, x_{t,a_t^*} \rangle - \langle \theta_*, x_{t,a_{i,t}} \rangle \right) = \underbrace{\sum_{t \in \Xi_T} \left( \langle \theta_*, x_{t,a_t^*} \rangle - \langle \theta_*, x_{t,a_t} \rangle \right)}_{T_1} + \underbrace{\sum_{s=1}^{S} \sum_{t \in \Psi_T^{(s)}} \left( \langle \theta_*, x_{t,a_t^*} \rangle - \langle \theta_*, x_{t,a_t} \rangle \right)}_{T_2}.$$

$$(78)$$

Let us first bound $T_2$ as follows.

$$
\begin{aligned}
T_2 &= \sum_{s=1}^{S} \sum_{t \in \Psi_T^{(s)}} \left( \langle \theta_*, x_{t,a_t^*} \rangle - \langle \theta_*, x_{t,a_t} \rangle \right) \\
&\overset{(a)}{\leq} \sum_{s=1}^{S} \sum_{t \in \Psi_T^{(s)}} \frac{8}{2^s \sqrt{M}} \\
&= \sum_{s=1}^{S} \frac{8}{2^s \sqrt{M}} |\Psi_T^{(s)}| \\
&\overset{(b)}{\leq} \sum_{s=1}^{S} 8 \left( \alpha + 2C \sqrt{\frac{\log \left( \frac{1}{\bar{\delta}} \right)}{M}} \right) \sqrt{2d |\Psi_T^{(s)}| \log(2M |\Psi_T^{(s)}|)} \\
&\overset{(c)}{\leq} 8S \left( \alpha + 2C \sqrt{\frac{\log \left( \frac{1}{\bar{\delta}} \right)}{M}} \right) \sqrt{2dT \log(2MT)} \\
&\leq 8(1 + \log(T)) \left( \alpha + 2C \sqrt{\frac{\log \left( \frac{1}{\bar{\delta}} \right)}{M}} \right) \sqrt{2dT \log(2MT)},
\end{aligned}
$$

$$(79)$$

where for (a), we used item (iii) of Lemma 13; for (b), we invoked Lemma 15 to bound $|\Psi_T^{(s)}|$; and for (c), we used the trivial bound $|\Psi_T^{(s)}| \leq T$. With $\bar{\delta} = \delta/(KST)$, the bound on $T_2$ reads as follows:

$$T_2 = O \left( \left( \alpha + 2C \sqrt{\frac{\log \left( \frac{KT}{\delta} \right)}{M}} \right) \log(T) \sqrt{2dT \log(2MT)} \right). \quad (80)$$

Now let us turn to bounding $T_1$. Consider any time-step $t \in \Xi_T$ where the action $a_t$ is chosen based on line 7 of Algorithm 3. Since $a_t^*$ is never eliminated on the clean event (item (ii) of Lemma 13), and since $a_t$ has the highest robust upper confidence bound among all active arms, it must be that:

$$
\begin{aligned}
& \hat{r}_{t,a_t^*}^{(s)} + w_{t,a_t^*}^{(s)} \leq \hat{r}_{t,a_t}^{(s)} + w_{t,a_t}^{(s)} \\
\implies & \langle \theta_*, x_{t,a_t^*} - x_{t,a_t} \rangle \leq 2 w_{t,a_t}^{(s)} \\
\implies & \langle \theta_*, x_{t,a_t^*} - x_{t,a_t} \rangle \leq \frac{2}{\sqrt{MT}},
\end{aligned}
$$

$$(81)$$

where for the first implication, we used item (i) of Lemma 13; and for the second, we used the fact for an action to be chosen based on line 7, it's confidence width must be bounded above by $1/(\sqrt{MT})$. We conclude that

$$T_1 = \sum_{t \in \Xi_T} \left( \langle \theta_*, x_{t,a_t^*} \rangle - \langle \theta_*, x_{t,a_t} \rangle \right) = O \left( \sqrt{\frac{T}{M}} \right).$$

Combining the above bound on $T_1$ with that on $T_2$ in Eq. (80), we have that with probability at least $1 - \delta$, the following is true for each good agent $i \in [M] \setminus \mathcal{B}$:

$$
\begin{aligned}
\sum_{t=1}^{T} \left( \langle \theta_*, x_{t,a_t^*} \rangle - \langle \theta_*, x_{t,a_t} \rangle \right) &= O\left( \left( \alpha + 2C\sqrt{\frac{\log\left(\frac{KT}{\delta}\right)}{M}} \right) \log(T) \sqrt{2dT \log(2MT)} \right) \\
&= \tilde{O}\left( \left( \alpha + \sqrt{1/M} \right) \sqrt{dT} \right).
\end{aligned}
\tag{82}
$$

This completes the proof. $\qquad \square$

# G  Alternate Strategies for Robust Collaborative Phased Elimination can lead to Sub-Optimal Regret Bounds.

In Section 3, where we introduced RCLB, we alluded to the fact that certain natural candidate strategies may lead to sub-optimal regret bounds. In this section, we elaborate on this point. Note that our end goal is to come up with a phased elimination step akin to line 9 of RCLB. To achieve this, in every epoch $\ell$, we need estimates of $\langle\theta_*, a\rangle$ along with associated confidence intervals for each $a \in \mathcal{A}_\ell$. In what follows, we will consider two natural candidate strategies for the same, and demonstrate that they each lead to confidence bounds that are looser than the ones we derived in Lemma 1. As such, using such bounds in the phased elimination step will lead to regret guarantees that are sub-optimal in their dependence on the model dimension $d$.

• **Candidate Strategy 1.** Suppose in every epoch $\ell$, the server collects the local model estimates $\{\hat{\theta}_i^{(\ell)}\}_{i \in [M]}$, and aims to first construct a robust estimate $\hat{\theta}^{(\ell)}$ of $\theta_*$. Subsequently, it uses $\langle\hat{\theta}^{(\ell)}, a\rangle$ as an estimate of $\langle\theta_*, a\rangle$ for each active arm $a \in \mathcal{A}_\ell$. To extract $\hat{\theta}^{(\ell)}$ from the local estimates $\{\hat{\theta}_i^{(\ell)}\}_{i \in [M]}$, we need a high-dimensional robust mean estimator. One natural candidate for this is the Iteratively Reweighted Mean Estimator from [22] since it leads to minimax-optimal error bounds. However, there is an immediate obstacle to directly applying the estimator from [22] on the model estimates $\{\hat{\theta}_i^{(\ell)}\}_{i \in [M]}$. This stems from the observation that although $\hat{\theta}_i^{(\ell)}$ is an unbiased estimate of $\theta_*$ for each good agent $i$, the covariance matrix associated with such an estimate may be *ill-conditioned*. In particular, it is not hard to verify that for each $i \in [M] \setminus \mathcal{B}$:

$$\mathbb{E}\left[\left(\hat{\theta}_i^{(\ell)} - \theta_*\right)\left(\hat{\theta}_i^{(\ell)} - \theta_*\right)' \Big| \mathcal{F}_\ell\right] = \tilde{V}_\ell^{-1}.$$

Thus, if we were to construct $\hat{\theta}^{(\ell)}$ as

$$\hat{\theta}^{(\ell)} = \mathtt{ITW}(\{\hat{\theta}_i^{(\ell)}, \, i \in [M]\}),$$

then based on Lemma 8, the error bound $\|\hat{\theta}^{(\ell)} - \theta_*\|$ would scale with $\|\tilde{V}_\ell^{-1}\|_2^{1/2}$.[8] This is undesirable as $\|\tilde{V}_\ell^{-1}\|_2^{1/2}$ can potentially take on a large value. To bypass this problem, we can use the same trick as we did for RC-GLM, and compute $\hat{\theta}^{(\ell)}$ as follows:

$$\hat{\theta}^{(\ell)} = \tilde{V}_\ell^{-1/2}\left(\mathtt{ITW}(\{\tilde{V}_\ell^{1/2}\hat{\theta}_i^{(\ell)}, \, i \in [M]\})\right).$$

The rationale behind the above approach is that the covariance matrix associated with $\tilde{V}_\ell^{1/2}\hat{\theta}_i^{(\ell)}, i \in [M] \setminus \mathcal{B}$, is $I_d$. Using Lemma 8, and following similar arguments as used to arrive at Lemmas 1 and 10, one can show that for each $a \in \mathcal{A}_\ell$, with probability at least $1 - \delta_\ell$, it holds that:

$$|\langle\hat{\theta}^{(\ell)}, a\rangle - \langle\theta_*, a\rangle| = O\left(\left(\sqrt{d} + \alpha\sqrt{M\log(1/\alpha)}\right)\epsilon_\ell\right). \tag{83}$$

It is instructive to compare the above estimate on $\langle\theta_*, a\rangle$ with the estimate $\mu_a^{(\ell)}$ we used in RCLB. Specifically, recall from Lemma 1 that for each $a \in \mathcal{A}_\ell$, with probability at least $1 - \delta_\ell$, it holds that

$$|\mu_a^{(\ell)} - \langle\theta_*, a\rangle| = O\left(\left(1 + \alpha\sqrt{M}\right)\epsilon_\ell\right).$$

Comparing the above upper bound with the one in Eq. (83), we note that while the former is independent of the dimension $d$, the latter does exhibit a dependence via the $\sqrt{d}$ term. Now suppose we use the upper-bound from Eq. (83) to construct a robust confidence threshold - say $\tilde{\gamma}_\ell$ - and use it to devise a phased elimination step as the one in line 9 of RCLB. Then, following the reasoning as that used to prove Theorem 1, one can establish a per-agent regret bound of

$$\tilde{O}\left(\left(\alpha\sqrt{\log(1/\alpha)} + \sqrt{\frac{d}{M}}\right)\sqrt{dT}\right),$$

which is unfortunately weaker than the guarantee we have in Theorem 1.

---

[8]Recall that we use $\mathtt{ITW}(\cdot)$ to represent the output of the Iteratively Reweighted Mean Estimator from [22], namely Algorithm 5.

• **Candidate Strategy 2.** The main idea is as follows. In every epoch $\ell$, the server queries each agent $i \in [M]$ to report their aggregate observation $r_{i,a}^{(\ell)}$ for each arm $a \in \text{Supp}(\pi_\ell)$. Recall that $r_{i,a}^{(\ell)}$ is the average of the rewards for arm $a$ observed by agent $i$ during epoch $\ell$. The server next computes an aggregate "clean" observation $\tilde{r}_a^{(\ell)}$ for each $a \in \text{Supp}(\pi_\ell)$ as follows:

$$\tilde{r}_a^{(\ell)} = \texttt{Median}\left(\{r_{i,a}^{(\ell)}, i \in [M]\}\right).$$

It then uses these clean observations to compute an estimate $\hat{\theta}^{(\ell)}$ of $\theta_*$ as follows:

$$\hat{\theta}^{(\ell)} = \bar{V}_\ell^{-1} Y_\ell, \text{ where } \bar{V}_\ell = \sum_{a \in \text{Supp}(\pi_\ell)} T_a^{(\ell)} a a' \; ; \; Y_\ell = \sum_{a \in \text{Supp}(\pi_\ell)} T_a^{(\ell)} \tilde{r}_a^{(\ell)} a,$$

and

$$T_a^{(\ell)} = \left\lceil \frac{\pi_\ell(a) d}{\epsilon_\ell^2} \log\left(\frac{1}{\delta_\ell}\right) \right\rceil.$$

The quantity $\hat{\theta}^{(\ell)}$ obtained above is now used to compute $\langle \hat{\theta}^{(\ell)}, a \rangle$ as an estimate of the true mean payoff $\langle \theta_*, a \rangle$ of each arm $a \in \mathcal{A}_\ell$.[9] As before, our goal is to bound $|\langle \hat{\theta}^{(\ell)}, a \rangle - \langle \theta_*, a \rangle|$ for each $a \in \mathcal{A}_\ell$. To that end, we start by noting that for each good agent $i$, $r_{i,a}^{(\ell)} \sim \mathcal{N}\left(\langle \theta_*, a \rangle, \frac{1}{m_a^{(\ell)}}\right)$. Invoking Lemma 2 then tells us that with probability at least $1 - \delta_\ell$,

$$|\tilde{r}_a^{(\ell)} - \langle \theta_*, a \rangle| \leq C\left(\alpha + \sqrt{\frac{\log(\frac{1}{\delta_\ell})}{M}}\right) \frac{1}{\sqrt{m_a^{(\ell)}}} \leq C\left(\alpha\sqrt{M} + \sqrt{\log\left(\frac{1}{\delta_\ell}\right)}\right) \frac{1}{\sqrt{T_a^{(\ell)}}}. \quad (84)$$

For our subsequent discussion, let us condition on the event on which the above bound holds for every arm in $\mathcal{A}_\ell$. On this event, we can say that for each $a \in \mathcal{A}_\ell$, $\tilde{r}_a^{(\ell)} = \langle \theta_*, a \rangle + e_a^{(\ell)}$, where $e_a^{(\ell)}$ is an error term satisfying the bound in Eq. (84). Now fix any $b \in \mathcal{A}_\ell$. Simple calculations reveal that:

$$
\begin{aligned}
|\langle \hat{\theta}^{(\ell)} - \theta_*, b \rangle| &= \left| \sum_{a \in \text{Supp}(\pi_\ell)} T_a^{(\ell)} \langle \bar{V}_\ell^{-1} a, b \rangle e_a^{(\ell)} \right| \\
&\leq \sum_{a \in \text{Supp}(\pi_\ell)} T_a^{(\ell)} |\langle \bar{V}_\ell^{-1} a, b \rangle| |e_a^{(\ell)}| \\
&\overset{(a)}{\leq} C\left(\alpha\sqrt{M} + \sqrt{\log\left(\frac{1}{\delta_\ell}\right)}\right) \underbrace{\sum_{a \in \text{Supp}(\pi_\ell)} \sqrt{T_a^{(\ell)}} |\langle \bar{V}_\ell^{-1} a, b \rangle|}_{T_1} \\
&\overset{(b)}{\leq} C\left(\alpha\sqrt{M} + \sqrt{\log\left(\frac{1}{\delta_\ell}\right)}\right) \sqrt{|\text{Supp}(\pi_\ell)| \left(b' \bar{V}_\ell^{-1} \left(\sum_{a \in \text{Supp}(\pi_\ell)} T_a^{(\ell)} a a'\right) \bar{V}_\ell^{-1} b\right)} \\
&\leq C\left(\alpha\sqrt{M} + \sqrt{\log\left(\frac{1}{\delta_\ell}\right)}\right) \sqrt{|\text{Supp}(\pi_\ell)|} \|b\|_{\bar{V}_\ell^{-1}} \\
&\overset{(c)}{\leq} C\left(\alpha\sqrt{M} + \sqrt{\log\left(\frac{1}{\delta_\ell}\right)}\right) \sqrt{48 d \log\log d} \|b\|_{\bar{V}_\ell^{-1}} \\
&\overset{(d)}{=} \tilde{O}\left(\sqrt{d}\left(1 + \alpha\sqrt{M}\right)\epsilon_\ell\right).
\end{aligned}
$$

$$(85)$$

---

[9]Note that the observations obtained from each agent $i$, namely $r_{i,a}^{(\ell)}, a \in \text{Supp}(\pi_\ell)$, provide direct information about the mean payoffs of arms *only* in $\text{Supp}(\pi_\ell)$. However, for the phased elimination step, we need estimates of the mean payoffs of *all arms in* $\mathcal{A}_\ell$, not just the ones in $\text{Supp}(\pi_\ell) \subseteq \mathcal{A}_\ell$. This is precisely why we need to go through an intermediate regression step to first compute an estimate of $\theta_*$.

In the above steps, for (a) we used the bound from Eq. (84); for (b), we used Jensen's inequality; for (c), we used the fact that $|\text{Supp}(\pi_\ell)| \leq 48 d \log \log d$; and for (d), following a similar argument as in the proof of Lemma 1, we used that

$$\|b\|_{\bar{V}_\ell^{-1}} = O\left(\frac{\epsilon_\ell}{\sqrt{\log\left(1/\delta_\ell\right)}}\right).$$

Comparing the bound in Eq. (85) with the one in Lemma 1, we once again note that while the latter bound is $d$-independent, the former has a clear dependence on $\sqrt{d}$. At the risk of sounding repetitive, if one were to employ the bound in Eq. (85) to construct a confidence threshold for phased elimination, and run through the same arguments as in the proof of Theorem 1, one would end up with a per-agent regret bound of

$$\tilde{O}\left(\left(1 + \alpha\sqrt{M}\right) d\sqrt{T}\right).$$

Unlike the near-optimal guarantee we have in Theorem 1, the above bound is clearly off by a factor of $\sqrt{d}$ from the optimal dependence on the model dimension $d$. The looseness in the bound mainly stems from the following fact: the error terms $\{e_a^{(\ell)}\}_{a\in\text{Supp}(\pi_\ell)}$ are not necessarily sub-Gaussian random variables that are independent across arms. One can contrast this to the analysis in Lemma 1, where the noise terms $\{\bar{\eta}_{i,a}^{(\ell)}\}_{a\in\text{Supp}(\pi_\ell)}$ were in fact Gaussian, and independent across arms. It is precisely the lack of nice statistical properties for the error terms $\{e_a^{(\ell)}\}_{a\in\text{Supp}(\pi_\ell)}$ that compels us to use Jensen's inequality to bound the term $T_1$ in Eq. (85). At the moment, it is unclear to us whether one can come up with a tighter bound for this candidate strategy.

**Main Takeaway.** The main message from this section is that deriving robust confidence intervals that lead to near-optimal bounds (such as the one in Theorem 1) is non-trivial, and requires a lot of care. In particular, the above discussion serves to highlight the significance of our algorithmic approach.

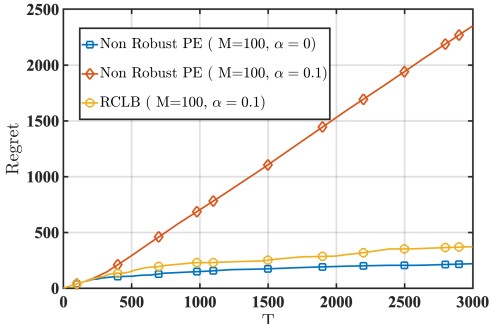

Figure 2: Performance of a vanilla non-robust distributed phased elimination algorithm vs RCLB for the attack model in Eq. (86).

# H   Experimental Results

In this section, we will provide various simulation results on synthetic data to corroborate the theory developed in our work. We start by describing the experimental setup for the linear bandit setting considered in Section 7.

## H.1   Experiments for the Linear Bandit Setting

**Linear Bandit Experimental Setup.** We generate 50 arms $a_1, a_2, \ldots, a_{50} \in \mathbb{R}^d$, where $d = 5$. Each arm $a_j, j \in [50]$, is generated by drawing each of the arm's coordinates i.i.d from the interval $[-\frac{1}{\sqrt{d}}, \frac{1}{\sqrt{d}}]$. It thus follows that $\|a_j\|_2 \leq 1, \forall j \in [50]$. The model parameter $\theta_*$ is chosen to be a 5-dimensional vector with each entry equal to $1/\sqrt{d}$. The rewards are generated based on the observation model in Eq. (1). We now describe the attack model for the linear bandit setting.

**Attack Model for Linear Bandit Setting.** The collective goal of the adversarial agents is to manipulate the server into selecting sub-optimal arms. To that end, each adversarial agent $i$ employs the simple strategy of reducing the rewards of the good arms and increasing the rewards of the bad arms. More precisely, in each epoch $\ell$, upon pulling an arm $a$ and observing the corresponding reward $y_a$, an adversarial agent $i$ does the following: if $y_a > p\langle\theta_*, a_*\rangle$, then this reward is corrupted to $\tilde{y}_a = y_a - \beta$; and if $y_a \leq p\langle\theta_*, a_*\rangle$, then the reward is corrupted to $\tilde{y}_a = y_a + \beta$. For this experiment, we fix $p = 0.6$ and $\beta = 5$. Agent $i \in \mathcal{B}$ then uses all the corrupted rewards in epoch $\ell$ to generate the local model estimate $\hat{\theta}_i^{(\ell)}$ that is transmitted to the server.

**An Alternate Attack Model.** To further test the robustness of RCLB, we consider an attack model different from the one in Section 7. In this attack, in each epoch $\ell$, every adversarial agent $i \in \mathcal{B}$ generates and transmits the following corrupted local model estimate to the server:

$$\hat{\theta}_i^{(\ell)} = -\frac{M}{|\mathcal{B}|}\theta_* - \frac{1}{|\mathcal{B}|}\sum_{j \in [M]\setminus\mathcal{B}} \hat{\theta}_j^{(\ell)}. \tag{86}$$

The idea behind the above attack is to trick the server into thinking that the true model estimate is $-\theta_*$, as opposed to $\theta_*$, by shifting the average of the agents' local model estimates towards $-\theta_*$. As we can see from Figure 2, the adversarial agents succeed in doing so when one employs a vanilla non-robust distributed phased elimination algorithm. However, our proposed approach RCLB continues to remain immune to such attacks, and guarantees sub-linear regret as suggested by our theory.

## H.2   Experiments for the Contextual Bandit Setting

The goal of this section is to validate our proposed robust collaborative algorithm for the contextual bandit setting, namely Algorithm 3.

**Contextual Bandit Experimental Setup.** As in the linear bandit experiment, we set the number of arms $K$ to be 50, the model dimension $d$ to be 5, and the true parameter $\theta_*$ to be a $d$-dimensional

vector with each entry equal to $1/\sqrt{d}$. At each time-step $t$, for each $a \in \mathcal{A}$, we generate the feature vector $x_{t,a}$ by drawing each of its entries i.i.d from the interval $[-\frac{1}{\sqrt{d}}, \frac{1}{\sqrt{d}}]$. The rewards are then generated based on the observation model in Eq. (12).

**Attack Model for Contextual Bandit Setting.** We use an attack strategy similar in spirit to the first attack model for the linear bandit setting. Specifically, at each time-step $t$, each adversarial agent $i \in \mathcal{B}$ does the following: if $r_{i,t} > p\langle \theta_*, x_{t,a_t^*}\rangle$, then the attacker sets $\tilde{r}_{i,t} = r_{i,t} - \beta$; if $r_{i,t} < p\langle \theta_*, x_{t,a_t^*}\rangle$, then the attacker sets $\tilde{r}_{i,t} = r_{i,t} + \beta$. The corrupted reward $\tilde{r}_{i,t}$ is then sent to the server. In this experiment, we fixed $p = 0.6$ and $\beta = 5$.

**Discussion of Simulation Results.** Figure 3 illustrates the results for the contextual linear bandit experiment. In Figure 3(a), we compare our proposed algorithm, namely Algorithm 3, to a naive distributed implementation of Algorithm 3 that does not account for adversarial agents. Similar to what we observed in Figure 1(a), while the non-robust algorithm incurs linear regret in the presence of adversaries, Algorithm 3 continues to guarantee sub-linear regret bounds. The plots in Figures 3(b)-(d) are analogous to the ones in Figures 1(b)-(d). In short, these plots once again indicate a clear benefit of collaboration (for small $\alpha$) in the presence of adversarial agents, thereby highlighting the importance of Algorithm 3, and validating Theorem 4.

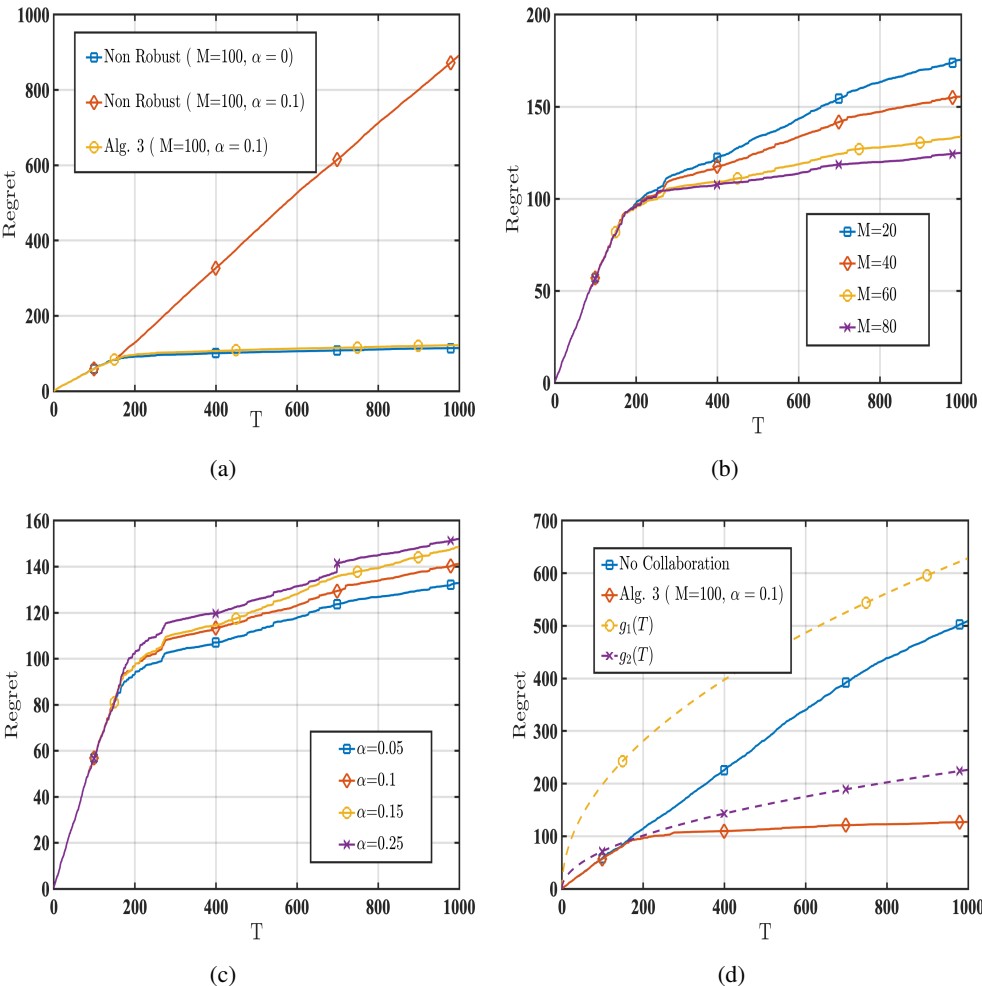

Figure 3: Plots of per-agent regret for the contextual bandit experiment. (a) Comparison between our proposed algorithm, namely Algorithm 3, and a vanilla non-robust distributed contextual bandit algorithm. (b) Performance of Algorithm 3 for varying number of agents M, with $\alpha = 0.1$. (c) Performance of Algorithm 3 for varying corruption fraction $\alpha$, with $M = 100$. (d) Comparison of Algorithm 3 to a non-robust contextual bandit algorithm where the agents do not collaborate; here, $\alpha = 0.1$ and $M = 100$. We also plotted theoretical upper bounds: $g_1(T) = 3\sqrt{dT}$ and $g_2(T) = 17(\alpha + \sqrt{\frac{1}{M}})\sqrt{dT}$.