# OpenReview forum: "Collaborative Linear Bandits with Adversarial Agents: Near-Optimal Regret Bounds"
_NeurIPS.cc/2022/Conference — NeurIPS 2022 Accept_

### Official Review · Reviewer_2PF8 · 2022-07-03

**Rating:** 7
**Confidence:** 3
**Soundness:** 3 good
**Presentation:** 4 excellent
**Contribution:** 3 good

**Summary:**

This paper considers the collaborative stochastic linear bandit problem where an $\alpha$ portion of the $M$ agents may act arbitrarily.
* The authors provided an algorithm RCLB that enjoys $\mathcal O((\alpha+1/\sqrt M)\sqrt{dT})$-style regret bound for each good agent.
* They also proved that regardless of $M$, an $\alpha\sqrt{dT}$ factor is unavoidable. So the regret bound of RCLB is nearly tight.
* They generalized the results to generalized linear bandits (GLM) and contextual linear bandits (CLB), giving near-optimal guarantees as well.

**Questions:**

* Is it possible to implement this algorithm more efficiently?
* Is this algorithm able to handle more general noise models other than independent Gaussians (e.g., conditionally 1-sub-Gaussian or $[-1,1]$-bounded noises which are common in the linear bandit literature)?

**Limitations:**

See Questions.

**Strengths And Weaknesses:**

Pros:
* This work is the first to consider the presence of misbehaving adversaries among the agents, showing that it is still possible to achieve $\mathcal O(\sqrt{MdT})$ total regret even with adversaries.
* The algorithm is not much more complicated compared to usual phase elimination algorithms (where there is no adversaries).
* The authors justified that the algorithm is worst-case optimal by showing both $\sqrt{MT}$ and $\alpha \sqrt{dT}$ factors are unavoidable.
* The algorithm is also the first to work in harder generalizations (collaborative GLMs and CLBs with adversaries) while achieving near-optimal regret.

Cons:
* The algorithm is not computationally efficient as it builds on phased elimination.
* The noise model is assumed to be independent Gaussians.

---

> ### Author Response · Authors · 2022-07-31
> **Response to Reviewer 2PF8**
>
> Thank you for your insightful review of our paper, and for your positive appraisal of our work. Below, we respond to your questions.
>
> **Reviewer 2PF8**: Is it possible to implement this algorithm more efficiently?
>
> **Response**: We would first like to mention that our proposed Algorithm 1 (RCLB) **does indeed admit an efficient implementation**. The only part of our algorithm that may appear to be *computationally expensive* is the G-optimal design. As we also explain to Reviewer nXam, to achieve minimax-optimal rates, it suffices for us to compute an *approximate* G-optimal design. Moreover, such an approximate G-optimal design can in fact be computed efficiently using the Frank-Wolfe algorithm, with run-time complexity $O(K d^2 \log \log d)$, where $K$ is the number of arms, and $d$ is the model-dimension. For more details, we refer the Reviewer to Proposition 3.17 of Reference [57] (in our paper), and also to Appendix G of the paper *Learning with Good Feature Representations in Bandits and in RL with a Generative Model*, Lattimore et al., 2020. In fact, the Frank-Wolfe implementation of an approximate G-optimal design is very standard, and has appeared in many other papers too; see, for instance, [31], and the aforementioned work by Lattimore. Finally, we note that for all our simulations in Appendix F of the supplementary material, we rely on this efficient Frank-Wolfe implementation.
>
> **Reviewer**: Is this algorithm able to handle more general noise models other than independent Gaussians (e.g., conditionally 1-sub-Gaussian or -bounded noises which are common in the linear bandit literature)?
>
> **Response**: This is an excellent question, one that was also raised by Reviewer Dj5j. In what follows, we explain in detail that **both our algorithms and analysis apply, with slight modifications, to both sub-Gaussian noise, and more generally, noise with bounded variance. The reason why we chose Gaussian noise was primarily for ease of exposition.** Suppose the noise samples are i.i.d. with zero-mean (a standard assumption) and unit variance (the argument we present next trivially extends to bounded variance $\sigma^2$). To deal with such general noise distributions, we need to make two minor changes to Algorithm 1. First, we replace the Median operation in line 8 of Algorithm 1 with the scalar univariate trimmed mean estimator recently proposed in reference [R0] below. The estimator in this paper is *easily implementable, and minimax-optimal*. Second, we set the robust threshold $\gamma_{\ell}$ in line 9 to be $\gamma_{\ell}=C(1+\sqrt{\alpha M}) \epsilon_{\ell}$, where $C$ is a suitably large universal constant; note that the only thing that has changed from before is the replacement of $\alpha$ by $\sqrt{\alpha}$ in $\gamma_{\ell}$. We will justify the choice of this threshold shortly, but before that, we mention the implications.
>
> [R0] ``*Robust multivariate mean estimation:
> the optimality of trimmed mean*", Lugosi and Mendelson, The Annals of Statistics, 2021.
>
> **Implications**.  With the two minor modifications to Algorithm 1 described above, we can establish an analogue of Theorem 1 where for each good agent, with probability at least $1-\delta$, the regret is bounded above by $\tilde{O}\left( \left(\sqrt{\alpha}+\sqrt{1/M} \right) \sqrt{dT}\right)$. Compared to the bound in Theorem 1, we note that our new bound is worsened by the replacement of $\alpha$ with $\sqrt{\alpha}$ - this is the price paid to account for general non-Gaussian distributions. For robust mean estimation with general noise distributions (with bounded fourth moments), the additive $\sqrt{\alpha}$ factor is *unavoidable*; see [R0], for instance. Thus, we conjecture that the bound on regret we mentioned above is also minimax-optimal.
>
> For sub-Gaussian distributions, we can achieve a tighter regret bound: by setting the robust confidence threshold to be $\gamma_{\ell}=C(1+\alpha\sqrt{\log(1/\alpha) M}) \epsilon_{\ell}$, we can achieve a per-agent regret bound of  $\tilde{O}\left( \left(\alpha\sqrt{\log(1/\alpha)}+\sqrt{1/M} \right) \sqrt{dT}\right)$. **Hence, with sub-Gaussian noise, we essentially recover the same bounds as with Gaussian noise (up to logarithmic factors).** It is very likely that with sub-Gaussian noise, the median will continue to yield the bound above, i.e., we may not even need the univariate trimmed mean estimator from [R0]. However, we do not have concrete proof of this fact yet.

---

> > ### Author Response · Authors · 2022-07-31
> > **Response to Reviewer 2PF8 (Continued)**
> >
> > **Changes in Analysis.** We now go over the minor changes that need to be made to the analysis of Theorem 1 in view of replacing the median operator with the trimmed mean estimator in [R0]. We first need an analog of Lemma 2 in Appendix A. This is supplied by Theorem 1 in [R0] which provides guarantees on the univariate trimmed mean estimator. Suppose we are given $M$ data samples, where all the good samples are i.i.d. with mean $\mu$ and variance $\sigma^2$. Moreover, suppose the fraction of bad samples is at most $1/2$. With probability at least $1-\delta$, we then have
> >
> > $$ |\hat{\mu}-\mu| \leq C \left(\sqrt{\alpha}+\sqrt{ \frac{\log(1/\delta)}{M}}\right) \sigma, $$
> >
> > where $\hat{\mu}$ is the output of the univariate trimmed mean estimator. Equipped with this result, one can follow the *exact same steps* as in the proof of Lemma 1 to justify the choice of the robust confidence threshold $\gamma_{\ell}=C(1+\sqrt{\alpha M}) \epsilon_{\ell}$. When the noise is additionally sub-Gaussian, the above bound can be tightened by replacing $\sqrt{\alpha}$ with $\alpha \sqrt{\log(1/\alpha)}$. The rest of the proof goes through *identically*.
> >
> > To sum up, we have argued that with minor modifications to both our algorithm and analysis, our overall approach can handle both sub-Gaussian noise, and noise with bounded variance. We will provide a detailed discussion of the above points in our revised manuscript.
> >
> > We will add a detailed discussion of the above points in our revised paper.
> >
> > We thank the Reviewer once again for their valuable review of our paper. We would be more than happy to answer any further questions.

---

> > > ### Comment · Reviewer_2PF8 · 2022-08-07
> > > **Reply to the Authors**
> > >
> > > Thank you for the detailed response! The response addresses my questions. I would like to keep my score unchanged.

---

### Official Review · Reviewer_32Q7 · 2022-07-11

**Rating:** 6
**Confidence:** 4
**Soundness:** 3 good
**Presentation:** 4 excellent
**Contribution:** 3 good

**Summary:**

The paper considers the collaborative linear bandit problem in the case of adversarial agents. In the proposed setting, a fraction $\alpha$ of agents can be adversarial. The main dilemma comes from the fact that although the collaboration reduces the number of required samples when adversarial agents are present, this can also work in the opposite direction. Besides the introduced setting, the authors propose a novel phased elimination algorithm that archives tight regret guarantees in the considered setting. Besides, the results are also provided for the two related settings: generalized linear bandit and the contextual bandit setting.

**Questions:**

Please see the previous section.

**Limitations:**

Yes, I think so.

**Strengths And Weaknesses:**

This is a well-written paper that considers an interesting twist to the collaborative linear bandit problem inspired by some recent works in the corrupted bandit literature. One of the main contributions is a novel and interesting setting that is first introduced in this paper, and that is not (to my knowledge) considered before.

When it comes to the setting, one might argue why such a clean distinction among adversarial and non-adversarial agents, i.e., perhaps, in practice, it might make sense that even “good” agents suffer some adversarial observations. On the other hand, I can see how this clean separation (coming from the problem setting) is suitable for the median of means tools used in the analysis.

Can you please comment on  $\alpha \in [0, 0.5)$, and what prevents us from $\alpha > 0.5$? What are the limits here? For example, in case each good agent plays individually the bound $(1-\alpha) M \sqrt{dT}$ should still hold, i.e., it does not depend on $\alpha$ being in  $[0, 0.5)$.

While the phased algorithm and inflated confidence bounds are used in some similar adversarial settings, I still think the theoretical results of this paper are solid, and the techniques used in the analysis are different from the previous works. The obtained results are tight, as is evident by the lower bound result. Finally, the results are also provided for the GLM and contextual settings.

The paper is missing simulations. I would argue that a natural baseline is an algorithm that ignores collaboration. Depending on $\alpha$, one can compare the proposed algorithm to that non-collaborative but completely robust one. Also, it is very easy to generate attacks in this setting by simply generating various $\hat{\theta}$’s in the case of adversarial agents.

Finally, I think that this is an interesting work with solid theoretical results that can be of interest to the community. After the discussion period, I might rethink my score and consider increasing it.

Some minor suggestions to improve:
- I’m not sure that Corollary 1 is needed. Too much space is used to multiply the result of Theorem 1 with M. Perhaps, consider removing it and using this additional space to provide more intuition on the “Construction of Robust Arm-Payoff Estimates and Confidence Intervals”.
- Problem formulation: Line 95: ”Based on all the information acquired by an agent ut to time…” This sentence was confusing because it was not clear how this is collaborative. It reads as if the agent only has access to its own observed rewards. Later on, this was explained, but a more careful introductory paragraph could avoid confusion.

---

> ### Author Response · Authors · 2022-07-31
> **Response to Reviewer 32Q7**
>
> Thank you for your insightful review of our paper, and for your encouraging comments. In what follows, we respond to each of your comments/questions.
>
> **Reviewer**: When it comes to the setting, one might argue why such a clean distinction among adversarial and non-adversarial agents.
>
> **Response**: This is an interesting point. While the Reviewer is correct that the *clean distinction* between adversarial and good agents does come with advantages, there are compelling reasons beyond this to consider our attack model. Indeed, the adversarial model where a fraction of agents in the system can act *arbitrarily*, while the others always behave as expected, has been extensively studied in the context of distributed stochastic optimization/learning [4]-[14], as well as the seminal work in robust consensus by Leslie Lamport (1982).  *Our work is the first to study this (historically well-accepted) model in the context of structured sequential-decision making problems.*
>
> Studying this attack model in the context of linear bandits entails a lot of care, both in algorithm design, and analysis. Indeed, naively applying tools from robust statistics (e.g., median of means) may not necessarily yield tight bounds: in lines 166-173, we mention two natural candidate strategies and explain in detail in Appendix E that these schemes incur a sub-optimal dependence on the dimension $d$. In contrast, our approach is **minimax optimal in all relevant parameters.** Establishing such minimax-optimality is non-trivial, as indicated by our lower bound analysis.
>
> Apart from historical reasons, the main rationale behind considering the attack model in this paper is to account for *completely unstructured worst-case behavior*. We would argue that such a model makes a lot of sense in practice, since for web-recommendation/advertisement applications, it may not be possible to *exactly* model the behavior of spammers. On top of such worst-case behavior, it may be possible, in principle, to consider more *structured* perturbations to the observations of even the good agents. The main point is that if we have prior knowledge about the nature of such perturbations, we can design our algorithms accordingly. As an example, suppose the observations of even the good agents are corrupted by noise that is not necessarily Gaussian, but rather *heavy-tailed*. If this is known a priori, one can then appeal to ideas from robust statistics that account for heavy-tailed noise. *In any case, we believe that the developments in this paper provide a solid theoretical foundation to study subsequent generalizations in future work (such as the one the Reviewer suggests).*
>
> We will add a discussion of the above points in our revised paper.
>
> **Reviewer**: Can you please comment on $\alpha \in [0, 0.5)$, and what prevents us from $\alpha > 0.5$ ?
>
> **Response**: This is an excellent point. The key question is:  *Who is the learner? The agent or the server?* As we explain shortly, the answer to this question has significant implications for whether or not we can tolerate the $\alpha > 0.5$ case. Suppose the learners are the agents, in that each good agent is capable of taking their own decisions (actions). If it is known a priori that $\alpha > 0.5$, then acting alone is the only option. This is because there is nothing to be gained by collaborating: no robust aggregation scheme can provide any guarantees when $\alpha > 0.5$; indeed, $\alpha = 0.5$ is a *fundamental breakdown point* when one considers an *arbitrary* corruption model; see [22], for instance. On the other hand, when $\alpha < 0.5$, one can significantly improve upon the trivial per-agent regret of $O(\sqrt{dT})$ by using the approach developed in our paper. This is revealed not only by our theory but the simulations that we report in Appendix F of the supplementary material. There, we explicitly compare our proposed algorithm RCLB with a non-collaborative strategy. As indicated by plots 3(d) and 3(e), the regret for RCLB can be significantly lower than that of the non-collaborative scheme, even when $\alpha > 0$.
>
> Now suppose the learner is the server, i.e., *the actions are decided by the server*.  As an example, consider a web-advertising setting where the server displays ads (actions) to a group of people (agents), obtains (potentially corrupted) feedback, and then decides upon subsequent ad displays to maximize click-through rates. Crucially, unlike the previous setting, the only way the server can acquire feedback about the environment is by interacting with the agents. *Thus, the server does not have the trivial option of acting alone*. Since in this setting, the server is forced to interact with potentially corrupted agents, $\alpha < 0.5$ is the only scenario that can lead to meaningful regret bounds. More precisely, when $\alpha > 0.5$, there is no hope of achieving sub-linear regret, let alone benefiting from collaboration.
>
> We will discuss these points in the revised paper.

---

> > ### Author Response · Authors · 2022-07-31
> > **Response to Reviewer 32Q7 (Continued)**
> >
> > **Reviewer**: The paper is missing simulations. I would argue that a natural baseline is an algorithm that ignores collaboration.
> >
> > **Response**: Thank you for bringing up the aspect of simulations, and for your suggestions along this line. The main point we want to convey here is that **in Appendix F of the supplementary material of our original submission, we did in fact have several simulation results, including each of the suggestions the Reviewer has made.** These include simulations for both the linear bandit setting, and its generalization to contextual bandits, for different types of attacks.  Notably, in figure 3(c), we study the effect of varying the corruption fraction $\alpha$ on the regret of our proposed algorithm RCLB. As expected, the regret increases with $\alpha$, but remains sub-linear, complying with Theorem 1.
> >
> > **Simulations to show improvement over non-collaborative baseline**. Let us now come to the key question: *how does our algorithm compare to a non-collaborative scheme that is always robust?* We did explore this important question and reported our results in figures 3(d) and 3(e). For this setting, we set the number of agents to be $M=100$, and the corruption fraction $\alpha=0.1$. From figures 3(d) and 3(e), we observe that **despite adversarial corruption, our proposed algorithm RCLB leads to considerably lower regret bounds as compared to the non-collaborative strategy, highlighting the importance of our approach.** In particular, these plots reveal that for small $\alpha$, one can indeed improve upon the trivial per-agent regret of $O(\sqrt{dT})$ achieved by the non-collaborative algorithm.
> >
> > **Simulations with different attack models**. We explore two different attack models. In the first model, the adversarial agents follow the simple strategy of increasing the rewards of the bad arms and decreasing the rewards of the good arms. The second model is similar to what the Reviewer suggests. The adversarial agents collude to generate incorrect estimates of $\theta_*$ based on Eq. (86) in Appendix F. The idea behind this attack strategy is to trick the server into thinking that the true model estimate is $- \theta_*$, as opposed to $ \theta_*$, by shifting the average of the agents’ local model estimates towards $- \theta_*$. Our simulations reveal the robustness of our proposed algorithm to each of the above attacks.
> >
> > We humbly request the Reviewer to go over our simulation results reported in Appendix F. We would be happy to perform any other simulations that the Reviewer deems important. Also, we will add some of these simulation results to the main body of our revised paper.
> >
> > **Response to Minor Suggestions**. Thank you for these suggestions, we will incorporate each of them in our revised manuscript.
> >
> > We thank the Reviewer once again for their valuable comments. We would be more than happy to answer any further questions. We also hope that our response will help improve the Reviewer's assessment of our work.

---

> > > ### Comment · Reviewer_32Q7 · 2022-08-08
> > > **Response**
> > >
> > > Thank you for the response. It addresses my questions/comments. I will keep my already positive rating unchanged.

---

### Official Review · Reviewer_Dj5j · 2022-07-13

**Rating:** 6
**Confidence:** 3
**Soundness:** 3 good
**Presentation:** 3 good
**Contribution:** 3 good

**Summary:**

This work focused on the collaborative linear bandit problem with adversarial agents. The author proposes novel robust algorithms for the linear bandit and general linear bandit problems, with a $(\alpha+1/\sqrt{M})\sqrt{dT}$ regret guarantee for each good agent. In addition, the author also provides a theoretical lower bound for the regret with adversarial agents and shows that the $\alpha \sqrt{T}$ regret is unavoidable.

**Questions:**

1. In the related work part, the author mentions recent advances in regret guarantees for the adversarial corruption bandit problem. More recent advanced works have focused on using the function approximation techniques. For instance, He et al. [2022] obtain a near-optimal regret guarantee with adversarial corruption. It is better if the author can mention them or discuss them.

He J, Zhou D, Zhang T, et al. Nearly Optimal Algorithms for Linear Contextual Bandits with Adversarial Corruptions

2. Since all theorems focus on regret for a single good agent, it seems unnecessary to consider total regret in equation (11).

**Limitations:**

This paper provides theoretical guarantees for learning linear bandit and linear mixture MDP. There is no negative societal impact.

**Strengths And Weaknesses:**

Strength:

1. The author provides theoretical guarantees for the algorithms and achieves a near-optimal regret guarantee without adversarial agents. In addition, the lower bound suggest that the adversarial-dependent term $\alpha \sqrt{dT}$ is near optimal with the factors $T$ and $\alpha$.

2. For this problem, the author also provides a theoretical guarantee for the lower regret bound.

Weakness:

 The algorithms require the stochastic noise to be Gaussian rather than sub-Gaussian, which is much more restrictive than previous work in linear bandit. In addition, for the contextual linear bandit and general linear bandit requires that all agents have the same feature vectors, which is far from the practice.

---

> ### Author Response · Authors · 2022-07-31
> **Response to Reviewer Dj5j**
>
> Thank you for the insightful review of our paper, and your positive comments on the work. Below, we carefully respond to each of your concerns.
>
> **Reviewer**: The algorithms require the stochastic noise to be Gaussian rather than sub-Gaussian, which is much more restrictive than previous work in linear bandit.
>
> **Response**: We thank the Reviewer for raising this interesting point. In what follows, we explain in detail that **both our algorithms and analysis apply, with slight modifications, to both sub-Gaussian noise, and more generally, noise with bounded variance. The reason why we chose Gaussian noise was primarily for ease of exposition.** Suppose the noise samples are i.i.d. with zero-mean (a standard assumption) and unit variance (the argument we present next trivially extends to bounded variance $\sigma^2$). To deal with such general noise distributions, we need to make two minor changes to Algorithm 1. First, we replace the Median operation in line 8 of Algorithm 1 with the scalar univariate trimmed mean estimator recently proposed in reference [R0] below. The estimator in this paper is *easily implementable, and minimax-optimal*. Second, we set the robust threshold $\gamma_{\ell}$ in line 9 to be $\gamma_{\ell}=C(1+\sqrt{\alpha M}) \epsilon_{\ell}$, where $C$ is a suitably large universal constant; note that the only thing that has changed from before is the replacement of $\alpha$ by $\sqrt{\alpha}$ in $\gamma_{\ell}$. We will justify the choice of this threshold shortly, but before that, we mention the implications.
>
> [R0] ``*Robust multivariate mean estimation:
> the optimality of trimmed mean*", Lugosi and Mendelson, The Annals of Statistics, 2021.
>
> **Implications**.  With the two minor modifications to Algorithm 1 described above, we can establish an analogue of Theorem 1 where for each good agent, with probability at least $1-\delta$, the regret is bounded above by $\tilde{O}\left( \left(\sqrt{\alpha}+\sqrt{1/M} \right) \sqrt{dT}\right)$. Compared to the bound in Theorem 1, we note that our new bound is worsened by the replacement of $\alpha$ with $\sqrt{\alpha}$ - this is the price paid to account for general non-Gaussian distributions. For robust mean estimation with general noise distributions (with bounded fourth moments), the additive $\sqrt{\alpha}$ factor is *unavoidable*; see [R0], for instance. Thus, we conjecture that the bound on regret we mentioned above is also minimax-optimal.
>
> For sub-Gaussian distributions, we can achieve a tighter regret bound: by setting the robust confidence threshold to be $\gamma_{\ell}=C(1+\alpha\sqrt{\log(1/\alpha) M}) \epsilon_{\ell}$, we can achieve a per-agent regret bound of  $\tilde{O}\left( \left(\alpha\sqrt{\log(1/\alpha)}+\sqrt{1/M} \right) \sqrt{dT}\right)$. **Hence, with sub-Gaussian noise, we essentially recover the same bounds as with Gaussian noise (up to logarithmic factors).** It is very likely that with sub-Gaussian noise, the median will continue to yield the bound above, i.e., we may not even need the univariate trimmed mean estimator from [R0]. However, we do not have concrete proof of this fact yet.
>
> **Changes in Analysis.** We now go over the minor changes that need to be made to the analysis of Theorem 1 in view of replacing the median operator with the trimmed mean estimator in [R0]. We first need an analog of Lemma 2 in Appendix A. This is supplied by Theorem 1 in [R0] which provides guarantees on the univariate trimmed mean estimator. Suppose we are given $M$ data samples, where all the good samples are i.i.d. with mean $\mu$ and variance $\sigma^2$. Moreover, suppose the fraction of bad samples is at most $1/2$. With probability at least $1-\delta$, we then have
>
> $$ |\hat{\mu}-\mu| \leq C \left(\sqrt{\alpha}+\sqrt{ \frac{\log(1/\delta)}{M}}\right) \sigma, $$
>
> where $\hat{\mu}$ is the output of the univariate trimmed mean estimator. Equipped with this result, one can follow the *exact same steps* as in the proof of Lemma 1 to justify the choice of the robust confidence threshold $\gamma_{\ell}=C(1+\sqrt{\alpha M}) \epsilon_{\ell}$. When the noise is additionally sub-Gaussian, the above bound can be tightened by replacing $\sqrt{\alpha}$ with $\alpha \sqrt{\log(1/\alpha)}$. The rest of the proof goes through *identically*.
>
> To sum up, we have argued that with minor modifications to both our algorithm and analysis, our overall approach can handle both sub-Gaussian noise, and noise with bounded variance. We will provide a detailed discussion of the above points in our revised manuscript.

---

> > ### Author Response · Authors · 2022-07-31
> > **Response to Reviewer Dj5j (Continued)**
> >
> > **Reviewer**: In addition, for the contextual linear bandit and general linear bandit requires that all agents have the same feature vectors, which is far from the practice.
> >
> >
> > **Response**: We completely agree with the Reviewer's viewpoint that in practice, the feature vectors of the agents may not be identical. However, if the feature vectors are non-identical, even without adversaries, it is not clear to us what to expect in terms of *benefit of collaboration*. Indeed, if the feature vectors of the agents are very different from one another, it may even be the case that collaboration does not necessarily improve per-agent regret beyond what an agent can achieve on its own. Formalizing ``similarities" among feature vectors to retain the benefits of collaboration (using, for instance, ideas from multi-task or representation learning), and then proving a result along those lines is a challenging and very interesting topic on its own - one that has not been well explored to the best of our knowledge. *Thus, the key message here is that even in the absence of adversaries, we do not have a clear baseline to compare with when agents are **heterogeneous**, with differing feature vectors.* As such, since the focus of our work is orthogonal, that of tackling worst-case adversaries, our goal was to provide a comprehensive analysis on this topic by starting out with the homogeneous setting as a conceptual first step. By providing minimax rates, and by showing that our framework can be extended to both generalized linear models and contextual bandits, our humble opinion is that we have taken a significant first step towards understanding more complex real-world settings (such as the one the Reviewer suggests) as future work. In accordance with the Reviewer's pertinent comment, we will discuss the above points in our revised manuscript.
> >
> > As an aside, we note that for the related problem of distributed stochastic optimization with adversarial agents, the initial works also assumed that all agents are homogeneous, in that they seek to minimize the same loss function. See, for instance, Ref [6] in our paper.
> >
> > **Response to Questions.** First, we thank the Reviewer for bringing to our attention the very recent work by He et al., we were unaware of it at the time of our submission. The paper studies a single-agent reward-corruption attack model and obtains some really elegant results; we will discuss it as such in our revised paper.
> >
> > The reason why we chose to speak about the group (total) regret is that it shows up in almost all the multi-agent bandit papers we are aware of. But as the learned Reviewer has observed, stating our bounds in terms of group regret and per-agent regret are essentially equivalent.
> >
> > We thank the Reviewer once again for their valuable comments and hope that our rebuttal will improve their final assessment of our work. We would be more than happy to answer any further questions.

---

### Official Review · Reviewer_nXam · 2022-07-20

**Rating:** 7
**Confidence:** 3
**Soundness:** 3 good
**Presentation:** 3 good
**Contribution:** 3 good

**Summary:**


This paper studies a variant of collaborative linear bandits where $\alpha$ fractions of M-agents act adversarially in a centralized setting. They propose a phased elimination algorithm with robust confidence intervals that achieves \tilde{O}((\alpha M+\sqrt{M})\sqrt{dT})  bound for the group regret. They also provide a lower bound on a slightly different model where each agent acts adversarially with probability $\alpha$, showing that the additive term $\alpha \sqrt{T}$ term is unavoidable. They further consider generalized linear bandit setting and contextual bandit setting.


**Questions:**

What if corruption fraction $\alpha$ is unknown to the system?


In algorithm 1, how do good agent's performance affect the resulting group regret? Indeed, it is not clear to me why G-optimal design is just solved approximately, though it can be exactly solved efficiently via convex optimization.

＝＝Post Rebuttal＝＝

Thank you for author's detailed response. I have read other reviews and its response, which even made me vote for acceptance.
Hence I will keep my score.

**Limitations:**

-

**Strengths And Weaknesses:**

Strength:
They propose a novel problem for the collaborative bandit learning with adversarial agents. This problem is a good capture of the real-world case of multiple data observation points, some of which are spammed or collapsed.

They propose a first algorithm named Robust Collaborative Phased Elimination: good agents follow the arm selection based on G-optimal design and report their estimate and adversarial agents report arbitrary vectors. The central server must consider both statistical uncertainty and adversarial one. To overcome this, they consider a robust confidence interval for each active arm and employ robust mean pay-offs. The server decides which to eliminate in each phase. The presented regret analysis and lower bound shows the optimality of the algorithm.

They further consider generalized linear bandit setting and contextual bandit setting.



Weakness:
Sometimes a sentence is too long or difficult to read (but in general, the paper is well structured)

The section “Related Work” is cluttered, citing more than 10 papers at a time. If you are proposing a new model, please provide a more careful summary of existing research and its position in the literature.

---

> ### Author Response · Authors · 2022-07-31
> **Response to Reviewer nXam**
>
> Thank you very much for your insightful review of our paper, and for your positive appraisal of the work. In accordance with your pertinent suggestions, we will improve our writing of the revised paper. Moreover, we will provide a more careful and thorough description of related work in the revised manuscript. Below, we respond to the two questions that you have raised.
>
> **Reviewer**:  What if corruption fraction $\alpha$ is unknown to the system?
>
> **Response**: This is an interesting question, thank you for bringing it up. In practice, if we do not know the corruption fraction $\alpha$, but have access to an upper bound on $\alpha$, say $\tilde{\alpha}$, we can essentially use $\tilde{\alpha}$ as a proxy for $\alpha$, both in our algorithm and its analysis. In this case, all our results go through identically by simply replacing $\alpha$ with  $\tilde{\alpha}$ in the bounds. However, if we have no idea at all about $\alpha$, then it is not clear whether one can design a robust algorithm that achieves minimax-optimal rates for the setting we consider here. The main reason is as follows. Note that although computing a median does not require knowledge of $\alpha$, the robust confidence threshold we use in line 9 of Algorithm 1 critically relies on the knowledge of the corruption fraction $\alpha$ (or an upper-bound on it). Moreover, the correctness of our overall approach, and the fact that it is minimax-optimal, relies heavily on the tightness of the robust confidence thresholds in Lemma 1. Thus, at the moment, we do not know of a way that can achieve tight regret bounds without any knowledge whatsoever of $\alpha$; investigating this aspect further is indeed an interesting topic of future research.
>
> **Reviewer**: In Algorithm 1, how do good agent's performance affect the resulting group regret?
>
> **Response**: Based on Algorithm 1, the fact that each good agent achieves a regret as given by Eq. (6) in Theorem 1 implies that the group regret is as given in Corollary 1. In particular, the group regret is simply the sum of the individual regrets of the good agents. Based on our algorithm, for small $\alpha$, the group regret in Eq. (7) improves upon the trivial group regret of $O(M\sqrt{dT})$; this highlights the fact that Algorithm 1 is able to still benefit from collaboration (among the good agents), despite worst-case adversarial corruptions.
>
> **Reviewer**:  Indeed, it is not clear to me why G-optimal design is just solved approximately, though it can be exactly solved efficiently via convex optimization.
>
> **Response**:  As it turns out, to achieve minimax-optimal rates, it suffices for us to compute an *approximate* G-optimal design. Moreover, the reason why we resort to such an approximate G-optimal design is because it be computed very efficiently using the Frank-Wolfe algorithm, with run-time complexity $O(K d^2 \log \log d)$, where $K$ is the number of arms, and $d$ is the model-dimension. For more details, we refer the Reviewer to Proposition 3.17 of Reference [57] (in our paper), and also to Appendix G of the paper *Learning with Good Feature Representations in Bandits and in RL with a Generative Model*, Lattimore et al., 2020. In fact, the Frank-Wolfe implementation of an approximate G-optimal design is very standard, and has appeared in many other papers too; see, for instance, [31], and the aforementioned work by Lattimore. Finally, we note that for all our simulations in Appendix F of the supplementary material, we rely on this efficient Frank-Wolfe implementation.
>
> We thank the Reviewer once again for their review of our paper. We would be happy to answer any further questions.

---

### Meta-Review · Area_Chair_wwZt · 2022-08-26

**Recommendation:** Accept
**Confidence:** Certain

**Metareview:**

This work studies an interesting collaborative linear bandits problem and makes solid technical contributions (efficient algorithms, optimal regret, and generalization to other settings). Clear accept. Please do address the minor issues pointed out by the reviewers in the final version.

**Award:**

No

---

### Decision · Program_Chairs · 2022-09-14

Accept